# On the Ability of Graph Neural Networks to Model Interactions Between Vertices

**Noam Razin, Tom Verbin, Nadav Cohen**

Tel Aviv University

{noamrazin,tomverbin}@mail.tau.ac.il, cohennadav@cs.tau.ac.il

## Abstract

Graph neural networks (GNNs) are widely used for modeling complex interactions between entities represented as vertices of a graph. Despite recent efforts to theoretically analyze the expressive power of GNNs, a formal characterization of their ability to model interactions is lacking. The current paper aims to address this gap. Formalizing strength of interactions through an established measure known as *separation rank*, we quantify the ability of certain GNNs to model interaction between a given subset of vertices and its complement, *i.e.* between the sides of a given partition of input vertices. Our results reveal that the ability to model interaction is primarily determined by the partition's *walk index* — a graph-theoretical characteristic defined by the number of walks originating from the boundary of the partition. Experiments with common GNN architectures corroborate this finding. As a practical application of our theory, we design an edge sparsification algorithm named *Walk Index Sparsification* (*WIS*), which preserves the ability of a GNN to model interactions when input edges are removed. WIS is simple, computationally efficient, and in our experiments has markedly outperformed alternative methods in terms of induced prediction accuracy.[1] More broadly, it showcases the potential of improving GNNs by theoretically analyzing the interactions they can model.

## 1 Introduction

*Graph neural networks* (*GNNs*) are a family of deep learning architectures, designed to model complex interactions between entities represented as vertices of a graph. In recent years, GNNs have been successfully applied across a wide range of domains, including social networks, biochemistry, and recommender systems (see, *e.g.*, [36, 59, 45, 49, 96, 104, 101, 18]). Consequently, significant interest in developing a mathematical theory behind GNNs has arisen.

One of the fundamental questions a theory of GNNs should address is *expressivity*, which concerns the class of functions a given architecture can realize. Existing studies of expressivity largely fall into three categories. First, and most prominent, are characterizations of ability to distinguish non-isomorphic graphs [103, 74, 72, 70, 6, 15, 10, 17, 43, 42, 80], as measured by equivalence to classical Weisfeiler-Leman graph isomorphism tests [99]. Second, are proofs for universal approximation of continuous permutation invariant or equivariant functions, possibly up to limitations in distinguishing some classes of graphs [73, 55, 25, 69, 3, 42]. Last, are works examining specific properties of GNNs such as frequency response [77, 5] or computability of certain graph attributes, *e.g.* moments, shortest paths, and substructure multiplicity [35, 9, 26, 39, 69, 23, 17, 105].

A major drawback of many existing approaches — in particular proofs of equivalence to Weisfeiler-Leman tests and those of universality — is that they operate in asymptotic regimes of unbounded

---

[1] An implementation of WIS is available at https://github.com/noamrazin/gnn_interactions.

37th Conference on Neural Information Processing Systems (NeurIPS 2023).

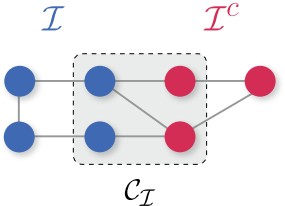
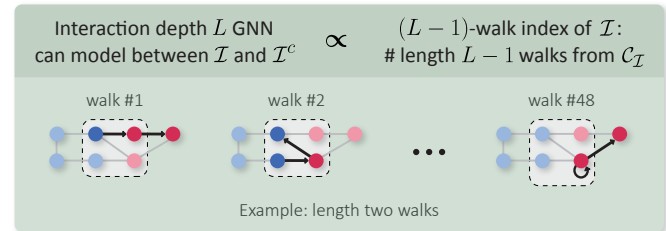

Figure 1: Illustration of our main theoretical contribution: quantifying the ability of GNNs to model interactions between vertices of an input graph. Consider a partition of vertices $(\mathcal{I}, \mathcal{I}^c)$, illustrated on the left, and a depth $L$ GNN with product aggregation (Section 3). For graph prediction, as illustrated on the right, the strength of interaction the GNN can model between $\mathcal{I}$ and $\mathcal{I}^c$, measured via separation rank (Section 2.2), is primarily determined by the partition's $(L-1)$-*walk index* — the number of length $L-1$ walks emanating from $\mathcal{C}_{\mathcal{I}}$, which is the set of vertices with an edge crossing the partition. The same holds for vertex prediction, except that there walk index is defined while only considering walks ending at the target vertex.

network width or depth. Moreover, to the best of our knowledge, none of the existing approaches formally characterize the strength of interactions GNNs can model between vertices, and how that depends on the structure of the input graph and the architecture of the neural network.

The current paper addresses the foregoing gaps. Namely, it theoretically quantifies the ability of fixed-size GNNs to model interactions between vertices, delineating the impact of the input graph structure and the neural network architecture (width and depth). Strength of modeled interactions is formalized via *separation rank* [12] — a commonly used measure for the interaction a function models between a subset of input variables and its complement (the rest of the input variables). Given a function and a partition of its input variables, the higher the separation rank, the more interaction the function models between the sides of the partition. Separation rank is prevalent in quantum mechanics, where it can be viewed as a measure of entanglement [62]. It was previously used for analyzing variants of convolutional, recurrent, and self-attention neural networks, yielding both theoretical insights and practical tools [30, 33, 61, 62, 64, 100, 65, 85]. We employ it for studying GNNs.

Key to our theory is a widely studied correspondence between neural networks with polynomial non-linearity and *tensor networks*[2] [32, 29, 30, 34, 90, 61, 62, 7, 56, 57, 63, 64, 83, 100, 84, 85, 65]. We extend this correspondence, and use it to analyze message-passing GNNs with product aggregation. We treat both graph prediction, where a single output is produced for an entire input graph, and vertex prediction, in which the network produces an output for every vertex. For graph prediction, we prove that the separation rank of a depth $L$ GNN with respect to a partition of vertices is primarily determined by the partition's $(L-1)$-*walk index* — a graph-theoretical characteristic defined to be the number of length $L-1$ walks originating from vertices with an edge crossing the partition. The same holds for vertex prediction, except that there walk index is defined while only considering walks ending at the target vertex. Our result, illustrated in Figure 1, implies that for a given input graph, the ability of GNNs to model interaction between a subset of vertices $\mathcal{I}$ and its complement $\mathcal{I}^c$, predominantly depends on the number of walks originating from the boundary between $\mathcal{I}$ and $\mathcal{I}^c$. We corroborate this proposition through experiments with standard GNN architectures, such as Graph Convolutional Network (GCN) [59] and Graph Isomorphism Network (GIN) [103].

Our theory formalizes conventional wisdom by which GNNs can model stronger interaction between regions of the input graph that are more interconnected. More importantly, we show that it facilitates an *edge sparsification* algorithm that preserves the expressive power of GNNs (in terms of ability to model interactions). Edge sparsification concerns removal of edges from a graph for reducing computational and/or memory costs, while attempting to maintain selected properties of the graph (*cf.* [11, 93, 48, 20, 86, 98, 67, 24]). In the context of GNNs, our interest lies in maintaining prediction accuracy as the number of edges removed from the input graph increases. We propose an algorithm for removing edges, guided by our separation rank characterization. The algorithm, named *Walk Index Sparsification* (*WIS*), is demonstrated to yield high predictive performance for GNNs (*e.g.* GCN and GIN) over standard benchmarks of various scales, even when removing a significant portion of

---

[2]Tensor networks form a graphical language for expressing contractions of tensors — multi-dimensional arrays. They are widely used for constructing compact representations of quantum states in areas of physics (see, *e.g.*, [97, 79]).

edges. WIS is simple, computationally efficient, and in our experiments has markedly outperformed alternative methods in terms of induced prediction accuracies across edge sparsity levels. More broadly, WIS showcases the potential of improving GNNs by theoretically analyzing the interactions they can model, and we believe its further empirical investigation is a promising direction for future research.

The remainder of the paper is organized as follows. Section 2 introduces notation and the concept of separation rank. Section 3 presents the theoretically analyzed GNN architecture. Section 4 theoretically quantifies (via separation rank) its ability to model interactions between vertices of an input graph. Section 5 proposes and evaluates WIS — an edge sparsification algorithm for arbitrary GNNs, born from our theory. Lastly, Section 6 concludes. Related work is discussed throughout, and for the reader's convenience, is recapitulated in Appendix B.

## 2 Preliminaries

### 2.1 Notation

For $N \in \mathbb{N}$, let $[N] := \{1, \ldots, N\}$. We consider an undirected input graph $\mathcal{G} = (\mathcal{V}, \mathcal{E})$ with vertices $\mathcal{V} = [|\mathcal{V}|]$ and edges $\mathcal{E} \subseteq \{\{i, j\} : i, j \in \mathcal{V}\}$. Vertices are equipped with features $\mathbf{X} := (\mathbf{x}^{(1)}, \ldots, \mathbf{x}^{(|\mathcal{V}|)}) \in \mathbb{R}^{D_x \times |\mathcal{V}|}$ — one $D_x$-dimensional feature vector per vertex ($D_x \in \mathbb{N}$). For $i \in \mathcal{V}$, we use $\mathcal{N}(i) := \{j \in \mathcal{V} : \{i, j\} \in \mathcal{E}\}$ to denote its set of neighbors, and, as customary in the context of GNNs, assume the existence of all self-loops, *i.e.* $i \in \mathcal{N}(i)$ for all $i \in \mathcal{V}$ (*cf.* [59, 50]). Furthermore, for $\mathcal{I} \subseteq \mathcal{V}$ we let $\mathcal{N}(\mathcal{I}) := \cup_{i \in \mathcal{I}} \mathcal{N}(i)$ be the neighbors of vertices in $\mathcal{I}$, and $\mathcal{I}^c := \mathcal{V} \setminus \mathcal{I}$ be the complement of $\mathcal{I}$. We use $\mathcal{C}_{\mathcal{I}}$ to denote the boundary of the partition $(\mathcal{I}, \mathcal{I}^c)$, *i.e.* the set of vertices with an edge crossing the partition, defined by $\mathcal{C}_{\mathcal{I}} := \{i \in \mathcal{I} : \mathcal{N}(i) \cap \mathcal{I}^c \neq \emptyset\} \cup \{j \in \mathcal{I}^c : \mathcal{N}(j) \cap \mathcal{I} \neq \emptyset\}$.[3] Lastly, we denote the number of length $l \in \mathbb{N}_{\geq 0}$ walks from any vertex in $\mathcal{I} \subseteq \mathcal{V}$ to any vertex in $\mathcal{J} \subseteq \mathcal{V}$ by $\rho_l(\mathcal{I}, \mathcal{J})$.[4] In particular, $\rho_l(\mathcal{I}, \mathcal{J}) = \sum_{i \in \mathcal{I}, j \in \mathcal{J}} \rho_l(\{i\}, \{j\})$.

Note that we focus on undirected graphs for simplicity of presentation. As discussed in Section 4, our results are extended to directed graphs in Appendix D.

### 2.2 Separation Rank: A Measure of Modeled Interaction

A prominent measure quantifying the interaction a multivariate function models between a subset of input variables and its complement (*i.e.* all other variables) is known as *separation rank*. The separation rank was introduced in [12], and has since been employed for various applications [51, 47, 13]. It is also a common measure of *entanglement*, a profound concept in quantum physics quantifying interaction between particles [62]. In the context of deep learning, it enabled analyses of expressivity and generalization in certain convolutional, recurrent, and self-attention neural networks, resulting in theoretical insights and practical methods (guidelines for neural architecture design, pretraining schemes, and regularizers — see [30, 33, 61, 62, 64, 100, 65, 85]).

Given a multivariate function $f : (\mathbb{R}^{D_x})^N \to \mathbb{R}$, its separation rank with respect to a subset of input variables $\mathcal{I} \subseteq [N]$ is the minimal number of summands required to express it, where each summand is a product of two functions — one that operates over variables indexed by $\mathcal{I}$, and another that operates over the remaining variables. Formally:

**Definition 1.** The *separation rank* of $f : (\mathbb{R}^{D_x})^N \to \mathbb{R}$ with respect to $\mathcal{I} \subseteq [N]$ is:

$$\mathrm{sep}(f; \mathcal{I}) := \min \left\{ R \in \mathbb{N}_{\geq 0} : \exists\, g^{(1)}, \ldots, g^{(R)} : (\mathbb{R}^{D_x})^{|\mathcal{I}|} \to \mathbb{R},\ \bar{g}^{(1)}, \ldots, \bar{g}^{(R)} : (\mathbb{R}^{D_x})^{|\mathcal{I}^c|} \to \mathbb{R} \right.$$
$$\left. \text{s.t. } f(\mathbf{X}) = \sum_{r=1}^{R} g^{(r)}(\mathbf{X}_{\mathcal{I}}) \cdot \bar{g}^{(r)}(\mathbf{X}_{\mathcal{I}^c}) \right\},$$

(1)

where $\mathbf{X} := (\mathbf{x}^{(1)}, \ldots, \mathbf{x}^{(N)})$, $\mathbf{X}_{\mathcal{I}} := (\mathbf{x}^{(i)})_{i \in \mathcal{I}}$, and $\mathbf{X}_{\mathcal{I}^c} := (\mathbf{x}^{(j)})_{j \in \mathcal{I}^c}$. By convention, if $f$ is identically zero then $\mathrm{sep}(f; \mathcal{I}) = 0$, and if the set on the right hand side of Equation (1) is empty then $\mathrm{sep}(f; \mathcal{I}) = \infty$.

---

[3]Due to the existence of self-loops, $\mathcal{C}_{\mathcal{I}}$ is exactly the shared neighbors of $\mathcal{I}$ and $\mathcal{I}^c$, *i.e.* $\mathcal{C}_{\mathcal{I}} = \mathcal{N}(\mathcal{I}) \cap \mathcal{N}(\mathcal{I}^c)$.
[4]For $l \in \mathbb{N}_{\geq 0}$, a sequence of vertices $i_0, \ldots, i_l \in \mathcal{V}$ is a length $l$ walk if $\{i_{l'-1}, i_{l'}\} \in \mathcal{E}$ for all $l' \in [l]$.

**Interpretation**  If $\text{sep}(f; \mathcal{I}) = 1$, the function is separable, meaning it does not model any interaction between $\mathbf{X}_\mathcal{I}$ and $\mathbf{X}_{\mathcal{I}^c}$, *i.e.* between the sides of the partition $(\mathcal{I}, \mathcal{I}^c)$. Specifically, it can be represented as $f(\mathbf{X}) = g(\mathbf{X}_\mathcal{I}) \cdot \bar{g}(\mathbf{X}_{\mathcal{I}^c})$ for some functions $g$ and $\bar{g}$. In a statistical setting, where $f$ is a probability density function, this would mean that $\mathbf{X}_\mathcal{I}$ and $\mathbf{X}_{\mathcal{I}^c}$ are statistically independent. The higher $\text{sep}(f; \mathcal{I})$ is, the farther $f$ is from separability, implying stronger modeling of interaction between $\mathbf{X}_\mathcal{I}$ and $\mathbf{X}_{\mathcal{I}^c}$.

## 3  Graph Neural Networks

Modern GNNs predominantly follow the message-passing paradigm [45, 50], whereby each vertex is associated with a hidden embedding that is updated according to its neighbors. The initial embedding of $i \in \mathcal{V}$ is taken to be its input features: $\mathbf{h}^{(0,i)} := \mathbf{x}^{(i)} \in \mathbb{R}^{D_x}$. Then, in a depth $L$ message-passing GNN, a common update scheme for the hidden embedding of $i \in \mathcal{V}$ at layer $l \in [L]$ is:

$$\mathbf{h}^{(l,i)} = \text{AGGREGATE}\left( \{\!\{ \mathbf{W}^{(l)} \mathbf{h}^{(l-1,j)} : j \in \mathcal{N}(i) \}\!\} \right), \tag{2}$$

where $\{\!\{\cdot\}\!\}$ denotes a multiset, $\mathbf{W}^{(1)} \in \mathbb{R}^{D_h \times D_x}, \mathbf{W}^{(2)} \in \mathbb{R}^{D_h \times D_h}, \dots, \mathbf{W}^{(L)} \in \mathbb{R}^{D_h \times D_h}$ are learnable weight matrices, with $D_h \in \mathbb{N}$ being the network's width (*i.e.* hidden dimension), and AGGREGATE is a function combining multiple input vectors into a single vector. A notable special case is GCN [59], in which AGGREGATE performs a weighted average followed by a non-linear activation function (*e.g.* ReLU).[5] Other aggregation operators are also viable, *e.g.* element-wise sum, max, or product (*cf.* [49, 53]). We note that distinguishing self-loops from other edges, and more generally, treating multiple edge types, is possible through the use of different weight matrices for different edge types [49, 88]. For conciseness, we hereinafter focus on the case of a single edge type, and treat multiple edge types in Appendix D.

After $L$ layers, the GNN generates hidden embeddings $\mathbf{h}^{(L,1)}, \dots, \mathbf{h}^{(L,|\mathcal{V}|)} \in \mathbb{R}^{D_h}$. For graph prediction, where a single output is produced for the whole graph, the hidden embeddings are usually combined into a single vector through the AGGREGATE function. A final linear layer with weights $\mathbf{W}^{(o)} \in \mathbb{R}^{1 \times D_h}$ is then applied to the resulting vector.[6] Overall, the function realized by a depth $L$ graph prediction GNN receives an input graph $\mathcal{G}$ with vertex features $\mathbf{X} := (\mathbf{x}^{(1)}, \dots, \mathbf{x}^{(|\mathcal{V}|)}) \in \mathbb{R}^{D_x \times |\mathcal{V}|}$, and returns:

$$\text{(graph prediction)} \quad f^{(\theta, \mathcal{G})}(\mathbf{X}) := \mathbf{W}^{(o)} \text{AGGREGATE}\left( \{\!\{ \mathbf{h}^{(L,i)} : i \in \mathcal{V} \}\!\} \right), \tag{3}$$

with $\theta := (\mathbf{W}^{(1)}, \dots, \mathbf{W}^{(L)}, \mathbf{W}^{(o)})$ denoting the network's learnable weights. For vertex prediction tasks, where the network produces an output for every $t \in \mathcal{V}$, the final linear layer is applied to each $\mathbf{h}^{(L,t)}$ separately. That is, for a target vertex $t \in \mathcal{V}$, the function realized by a depth $L$ vertex prediction GNN is given by:

$$\text{(vertex prediction)} \quad f^{(\theta, \mathcal{G}, t)}(\mathbf{X}) := \mathbf{W}^{(o)} \mathbf{h}^{(L,t)}. \tag{4}$$

Our aim is to investigate the ability of GNNs to model interactions between vertices. Prior studies of interactions modeled by different deep learning architectures have focused on neural networks with polynomial non-linearity, building on their representation as tensor networks [32, 30, 34, 90, 61, 62, 7, 56, 63, 64, 83, 100, 84, 85, 65]. Although neural networks with polynomial non-linearity are less common in practice, they have demonstrated competitive performance [28, 31, 91, 94, 27, 37, 53], and hold promise due to their compatibility with quantum computation [46, 14] and fully homomorphic encryption [44]. More importantly, their analyses brought forth numerous insights that were demonstrated empirically and led to development of practical tools for widespread deep learning models (with non-linearities such as ReLU).

Following the above, in our theoretical analysis (Section 4) we consider GNNs with (element-wise) product aggregation, which are polynomial functions of their inputs. Namely, the AGGREGATE operator from Equations (2) and (3) is taken to be:

$$\text{AGGREGATE}(\mathcal{X}) := \odot_{\mathbf{x} \in \mathcal{X}} \mathbf{x}, \tag{5}$$

---

[5]In GCN, AGGREGATE also has access to the degrees of vertices, which are used for computing the averaging weights. We omit the dependence on vertex degrees in our notation for conciseness.

[6]We treat the case of output dimension one merely for the sake of presentation. Extension of our theory (delivered in Section 4) to arbitrary output dimension is straightforward — the results hold as stated for each of the functions computing an output entry.

where $\odot$ stands for the Hadamard product and $\mathcal{X}$ is a multiset of vectors. The resulting architecture can be viewed as a variant of the GNN proposed in [53], where it was shown to achieve competitive performance in practice. Central to our proofs are tensor network representations of GNNs with product aggregation (formally established in Appendix E), analogous to those used for analyzing other types of neural networks. We empirically demonstrate our theoretical findings on popular GNNs (Section 4.2), such as GCN and GIN with ReLU non-linearity, and use them to derive a practical edge sparsification algorithm (Section 5).

We note that some of the aforementioned analyses of neural networks with polynomial non-linearity were extended to account for additional non-linearities, including ReLU, through constructs known as *generalized tensor networks* [29]. We thus believe our theory may be similarly extended, and regard this as an interesting direction for future work.

# 4  Theoretical Analysis: The Effect of Input Graph Structure and Neural Network Architecture on Modeled Interactions

In this section, we employ separation rank (Definition 1) to theoretically quantify how the input graph structure and network architecture (width and depth) affect the ability of a GNN with product aggregation to model interactions between input vertices. We overview the main results and their implications in Section 4.1, while deferring the formal analysis to Appendix A due to lack of space. Section 4.2 provides experiments demonstrating our theory's implications on common GNNs, such as GCN and GIN with ReLU non-linearity.

## 4.1  Overview and Implications

Consider a depth $L$ GNN with width $D_h$ and product aggregation (Section 3). Given a graph $\mathcal{G}$, any assignment to the weights of the network $\theta$ induces a multivariate function — $f^{(\theta,\mathcal{G})}$ for graph prediction (Equation (3)) and $f^{(\theta,\mathcal{G},t)}$ for prediction over a given vertex $t \in \mathcal{V}$ (Equation (4)) — whose variables correspond to feature vectors of input vertices. The separation rank of this function with respect to $\mathcal{I} \subseteq \mathcal{V}$ thus measures the interaction modeled across the partition $(\mathcal{I}, \mathcal{I}^c)$, *i.e.* between the vertices in $\mathcal{I}$ and those in $\mathcal{I}^c$. The higher the separation rank is, the stronger the modeled interaction.

Key to our analysis are the following notions of *walk index*, defined by the number of walks emanating from the boundary of the partition $(\mathcal{I}, \mathcal{I}^c)$, *i.e.* from vertices with an edge crossing the partition induced by $\mathcal{I}$ (see Figure 1 for an illustration).

**Definition 2.** Let $\mathcal{I} \subseteq \mathcal{V}$. Denote by $\mathcal{C}_{\mathcal{I}}$ the set of vertices with an edge crossing the partition $(\mathcal{I}, \mathcal{I}^c)$, *i.e.* $\mathcal{C}_{\mathcal{I}} := \{i \in \mathcal{I} : \mathcal{N}(i) \cap \mathcal{I}^c \neq \emptyset\} \cup \{j \in \mathcal{I}^c : \mathcal{N}(j) \cap \mathcal{I} \neq \emptyset\}$, and recall that $\rho_l(\mathcal{C}_{\mathcal{I}}, \mathcal{J})$ denotes the number of length $l \in \mathbb{N}_{\geq 0}$ walks from any vertex in $\mathcal{C}_{\mathcal{I}}$ to any vertex in $\mathcal{J} \subseteq \mathcal{V}$. For $L \in \mathbb{N}$:

- (graph prediction) we define the $(L-1)$-*walk index* of $\mathcal{I}$, denoted $\mathrm{WI}_{L-1}(\mathcal{I})$, to be the number of length $L-1$ walks originating from $\mathcal{C}_{\mathcal{I}}$, *i.e.* $\mathrm{WI}_{L-1}(\mathcal{I}) := \rho_{L-1}(\mathcal{C}_{\mathcal{I}}, \mathcal{V})$; and

- (vertex prediction) for $t \in \mathcal{V}$ we define the $(L-1, t)$-*walk index* of $\mathcal{I}$, denoted $\mathrm{WI}_{L-1,t}(\mathcal{I})$, to be the number of length $L-1$ walks from $\mathcal{C}_{\mathcal{I}}$ that end at $t$, *i.e.* $\mathrm{WI}_{L-1,t}(\mathcal{I}) := \rho_{L-1}(\mathcal{C}_{\mathcal{I}}, \{t\})$.

As our main theoretical contribution, we prove:

**Theorem 1** (informally stated). *For all weight assignments $\theta$ and $t \in \mathcal{V}$:*

$$\text{(graph prediction)} \quad \log\big(\mathrm{sep}\big(f^{(\theta,\mathcal{G})}; \mathcal{I}\big)\big) = \mathcal{O}\big(\log(D_h) \cdot \mathrm{WI}_{L-1}(\mathcal{I})\big),$$

$$\text{(vertex prediction)} \quad \log\big(\mathrm{sep}\big(f^{(\theta,\mathcal{G},t)}; \mathcal{I}\big)\big) = \mathcal{O}\big(\log(D_h) \cdot \mathrm{WI}_{L-1,t}(\mathcal{I})\big).$$

*Moreover, nearly matching lower bounds hold for almost all weight assignments.*[7]

The upper and lower bounds are formally established by Theorems 2 and 3 in Appendix A, respectively, and are generalized to input graphs with directed edges and multiple edge types in Appendix D. Theorem 1 implies that, the $(L-1)$-walk index of $\mathcal{I}$ in graph prediction and its $(L-1, t)$-walk index in vertex prediction control the separation rank with respect to $\mathcal{I}$, and are thus paramount for

---

[7]Almost all in the sense of all weight assignments but a set of Lebesgue measure zero.

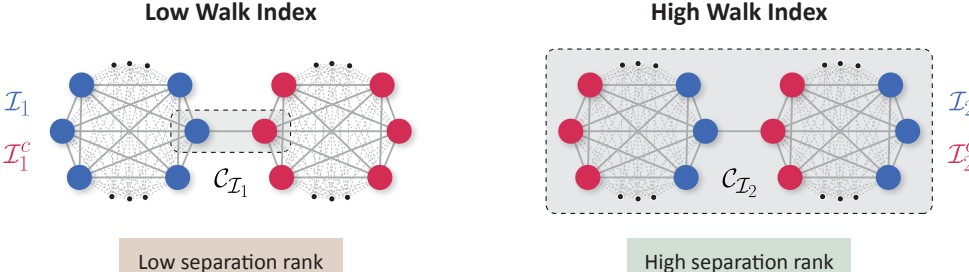

**Figure 2:** Depth $L$ GNNs can model stronger interactions between sides of partitions that have a higher walk index (Definition 2). The partition $(\mathcal{I}_1, \mathcal{I}_1^c)$ (left) divides the vertices into two separate cliques, connected by a single edge. Only two vertices reside in $\mathcal{C}_{\mathcal{I}_1}$ — the set of vertices with an edge crossing the partition. Taking for example depth $L = 3$, the 2-walk index of $\mathcal{I}_1$ is $\Theta(|\mathcal{V}|^2)$ and its $(2, t)$-walk index is $\Theta(|\mathcal{V}|)$, for $t \in \mathcal{V}$. In contrast, the partition $(\mathcal{I}_2, \mathcal{I}_2^c)$ (right) equally divides the vertices in each clique to different sides. All vertices reside in $\mathcal{C}_{\mathcal{I}_2}$, meaning the 2-walk index of $\mathcal{I}_2$ is $\Theta(|\mathcal{V}|^3)$ and its $(2, t)$-walk index is $\Theta(|\mathcal{V}|^2)$, for $t \in \mathcal{V}$. Hence, in both graph and vertex prediction scenarios, the walk index of $\mathcal{I}_1$ is relatively low compared to that of $\mathcal{I}_2$. Our analysis (Section 4.1 and Appendix A) states that a higher separation rank can be attained with respect to $\mathcal{I}_2$, meaning stronger interaction can be modeled across $(\mathcal{I}_2, \mathcal{I}_2^c)$ than across $(\mathcal{I}_1, \mathcal{I}_1^c)$. We empirically confirm this prospect in Section 4.2.

modeling interaction between $\mathcal{I}$ and $\mathcal{I}^c$ — see Figure 2 for an illustration. It thereby formalizes the conventional wisdom by which GNNs can model stronger interaction between areas of the input graph that are more interconnected. We support this finding empirically with common GNN architectures (*e.g.* GCN and GIN with ReLU non-linearity) in Section 4.2.

One may interpret Theorem 1 as encouraging addition of edges to an input graph. Indeed, the theorem states that such addition can enhance the GNN's ability to model interactions between input vertices. This accords with existing evidence by which increasing connectivity can improve the performance of GNNs in practice (see, *e.g.*, [40, 1]). However, special care needs to be taken when adding edges: it may distort the semantic meaning of the input graph, and may lead to plights known as over-smoothing and over-squashing [68, 78, 22, 1, 8]. Rather than employing Theorem 1 for adding edges, we use it to select which edges to preserve in a setting where some must be removed. That is, we employ it for designing an edge sparsification algorithm. The algorithm, named *Walk Index Sparsification* (*WIS*), is simple, computationally efficient, and in our experiments has markedly outperformed alternative methods in terms of induced prediction accuracy. We present and evaluate it in Section 5.

### 4.2 Empirical Demonstration

Our theoretical analysis establishes that, the strength of interaction GNNs can model across a partition of input vertices is primarily determined by the partition's walk index — a graph-theoretical characteristic defined by the number of walks originating from the boundary of the partition (see Definition 2). The analysis formally applies to GNNs with product aggregation (see Section 3), yet we empirically demonstrate that its conclusions carry over to various other message-passing GNN architectures, namely GCN [59], GAT [96], and GIN [103] (with ReLU non-linearity). Specifically, through controlled experiments, we show that such models perform better on tasks in which the partitions that require strong interaction are ones with higher walk index, given that all other aspects of the tasks are the same. A description of these experiments follows. For brevity, we defer some implementation details to Appendix H.2.

We constructed two graph prediction datasets, in which the vertex features of each input graph are patches of pixels from two randomly sampled Fashion-MNIST [102] images, and the goal is to predict whether the two images are of the same class.[8] In both datasets, all input graphs have the same structure: two separate cliques with 16 vertices each, connected by a single edge. The datasets differ in how the image patches are distributed among the vertices: in the first dataset each clique holds all the patches of a single image, whereas in the second dataset each clique holds half of the patches from the first image and half of the patches from the second image. Figure 2 illustrates how

---

[8]Images are sampled such that the amount of positive and negative examples are roughly balanced.

Table 1: In accordance with our theory (Section 4.1 and Appendix A), GNNs can better fit datasets in which the partitions (of input vertices) that require strong interaction are ones with higher walk index (Definition 2). Table reports means and standard deviations, taken over five runs, of train and test accuracies obtained by GNNs of depth 3 and width 16 on two datasets: one in which the essential partition — *i.e.* the main partition requiring strong interaction — has low walk index, and another in which it has high walk index (see Section 4.2 for a detailed description of the datasets). For all GNNs, the train accuracy attained over the second dataset is considerably higher than that attained over the first dataset. Moreover, the better train accuracy translates to better test accuracy. See Appendix H.2 for further implementation details.

| | | Essential Partition Walk Index | |
| --- | --- | --- | --- |
| | | Low | High |
| GCN | Train Acc. (%) | $70.4 \pm 1.7$ | $\mathbf{81.4} \pm 2.0$ |
| | Test Acc. (%) | $52.7 \pm 1.9$ | $\mathbf{66.2} \pm 1.1$ |
| GAT | Train Acc. (%) | $82.8 \pm 2.6$ | $\mathbf{88.5} \pm 1.1$ |
| | Test Acc. (%) | $69.6 \pm 0.6$ | $\mathbf{72.1} \pm 1.2$ |
| GIN | Train Acc. (%) | $83.2 \pm 0.8$ | $\mathbf{94.2} \pm 0.8$ |
| | Test Acc. (%) | $53.7 \pm 1.8$ | $\mathbf{64.8} \pm 1.4$ |

image patches are distributed in the first (left hand side of the figure) and second (right hand side of the figure) datasets, with blue and red marking assignment of vertices to images.

Each dataset requires modeling strong interaction across the partition separating the two images, referred to as the *essential partition* of the dataset. In the first dataset the essential partition separates the two cliques, thus it has low walk index. In the second dataset each side of the essential partition contains half of the vertices from the first clique and half of the vertices from the second clique, thus the partition has high walk index. For an example illustrating the gap between these walk indices see Figure 2.

Table 1 reports train and test accuracies achieved by GCN, GAT, and GIN (with ReLU non-linearity) over both datasets. In compliance with our theory, the GNNs fit the dataset whose essential partition has high walk index significantly better than they fit the dataset whose essential partition has low walk index. Furthermore, the improved train accuracy translates to improvements in test accuracy.

# 5 Practical Application: Expressivity Preserving Edge Sparsification

Section 4 theoretically characterizes the ability of a GNN to model interactions between input vertices. It reveals that this ability is controlled by a graph-theoretical property we call walk index (Definition 2). The current section derives a practical application of our theory, specifically, an *edge sparsification* algorithm named *Walk Index Sparsification* (*WIS*), which preserves the ability of a GNN to model interactions when input edges are removed. We present WIS, and show that it yields high predictive performance for GNNs over standard vertex prediction benchmarks of various scales, even when removing a significant portion of edges. In particular, we evaluate WIS using GCN [59], GIN [103], and ResGCN [66] over multiple datasets, including: Cora [89], which contains thousands of edges, DBLP [16], which contains tens of thousands of edges, and OGBN-ArXiv [52], which contains more than a million edges. WIS is simple, computationally efficient, and in our experiments has markedly outperformed alternative methods in terms of prediction accuracy across edge sparsity levels. We believe its further empirical investigation is a promising direction for future research.

## 5.1 Walk Index Sparsification (WIS)

Running GNNs over large-scale graphs can be prohibitively expensive in terms of runtime and memory. A natural way to tackle this problem is edge sparsification — removing edges from an input graph while attempting to maintain prediction accuracy (*cf.* [67, 24]).[9,10]

---

[9]As opposed to *edge rewiring* methods that add or remove only a few edges with the goal of improving prediction accuracy (*e.g.*, [106, 71, 95, 8]).

[10]An alternative approach is to remove vertices from an input graph (see, *e.g.*, [60]). However, this approach is unsuitable for vertex prediction tasks, so we limit our attention to edge sparsification.

---

**Algorithm 1** $(L-1)$-Walk Index Sparsification (WIS)
(instance of a general scheme described in Appendix F)

---

**Input:** $\mathcal{G}$ — graph , $L \in \mathbb{N}$ — GNN depth , $N \in \mathbb{N}$ — number of edges to remove
**Result:** Sparsified graph obtained by removing $N$ edges from $\mathcal{G}$

---

  **for** $n = 1, \ldots, N$ **do**
    # per edge, compute walk indices of partitions induced by $\{t\}$, for $t \in \mathcal{V}$, after its removal
    **for** $e \in \mathcal{E}$ (excluding self-loops) **do**
      initialize $\mathbf{s}^{(e)} = (0, \ldots, 0) \in \mathbb{R}^{|\mathcal{V}|}$
      remove $e$ from $\mathcal{G}$ (temporarily)
      for every $t \in \mathcal{V}$, set $\mathbf{s}^{(e)}_t = \mathrm{WI}_{L-1,t}(\{t\})$   # = number of length $L-1$ walks from $\mathcal{C}_{\{t\}}$ to $t$
      add $e$ back to $\mathcal{G}$
    **end for**
    # prune edge whose removal harms walk indices the least according to an order over $(\mathbf{s}^{(e)})_{e \in \mathcal{E}}$
    for $e \in \mathcal{E}$, sort the entries of $\mathbf{s}^{(e)}$ in ascending order
    let $e' \in \mathrm{argmax}_{e \in \mathcal{E}} \, \mathbf{s}^{(e)}$ according to lexicographic order over tuples
    **remove** $e'$ from $\mathcal{G}$ (permanently)
  **end for**

---

---

**Algorithm 2** 1-Walk Index Sparsification (WIS)   (efficient version of Algorithm 1 for $L = 2$)

---

**Input:** $\mathcal{G}$ — graph , $N \in \mathbb{N}$ — number of edges to remove
**Result:** Sparsified graph obtained by removing $N$ edges from $\mathcal{G}$

---

  **for** $n = 1, \ldots, N$ **do**
    **for** $\{i, j\} \in \mathcal{E}$ (excluding self-loops) **do**
      let $\mathrm{deg}_{min}(i, j) := \min\{|\mathcal{N}(i)|, |\mathcal{N}(j)|\}$
      let $\mathrm{deg}_{max}(i, j) := \max\{|\mathcal{N}(i)|, |\mathcal{N}(j)|\}$
    **end for**
    # prune edge $\{i, j\} \in \mathcal{E}$ with maximal $\mathrm{deg}_{min}(i, j)$, breaking ties using $\mathrm{deg}_{max}(i, j)$
    let $e' \in \mathrm{argmax}_{\{i,j\} \in \mathcal{E}} \big(\mathrm{deg}_{min}(i, j), \mathrm{deg}_{max}(i, j)\big)$ according to lexicographic order over pairs
    **remove** $e'$ from $\mathcal{G}$
  **end for**

---

Our theory (Section 4) establishes that, the strength of interaction a depth $L$ GNN can model across a partition of input vertices is determined by the partition's walk index, a quantity defined by the number of length $L-1$ walks originating from the partition's boundary. This brings forth a recipe for pruning edges. First, choose partitions across which the ability to model interactions is to be preserved. Then, for every input edge (excluding self-loops), compute a tuple holding what the walk indices of the chosen partitions will be if the edge is to be removed. Lastly, remove the edge whose tuple is maximal according to a preselected order over tuples (*e.g.* an order based on the sum, min, or max of a tuple's entries). This process repeats until the desired number of edges are removed. The idea behind the above-described recipe, which we call *General Walk Index Sparsification*, is that each iteration greedily prunes the edge whose removal takes the smallest toll in terms of ability to model interactions across chosen partitions — see Algorithm 3 in Appendix F for a formal outline. Below we describe a specific instantiation of the recipe for vertex prediction tasks, which are particularly relevant with large-scale graphs, yielding our proposed algorithm — Walk Index Sparsification (WIS). Exploration of other instantiations is regarded as a promising avenue for future work.

In vertex prediction tasks, the interaction between an input vertex and the remainder of the input graph is of central importance. Thus, it is natural to choose the partitions induced by singletons (*i.e.* the partitions $(\{t\}, \mathcal{V} \setminus \{t\})$, where $t \in \mathcal{V}$) as those across which the ability to model interactions is to be preserved. We would like to remove edges while avoiding a significant deterioration in the ability to model interaction under any of the chosen partitions. To that end, we compare walk index tuples according to their minimal entries, breaking ties using the second smallest entries, and so

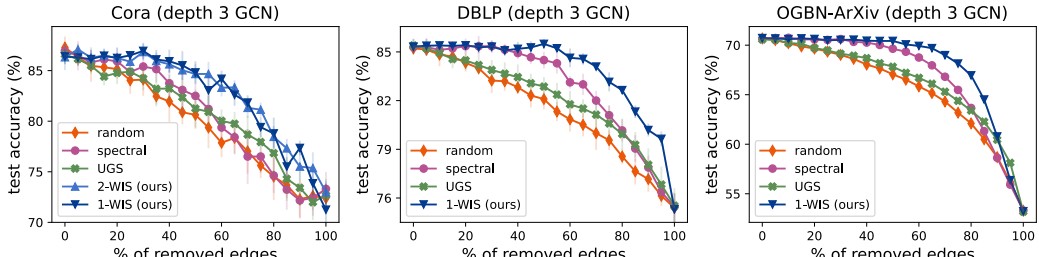

Figure 3: Comparison of GNN accuracies following sparsification of input edges — WIS, the edge sparsification algorithm brought forth by our theory (Algorithm 1), markedly outperforms alternative methods. Plots present test accuracies achieved by a depth $L = 3$ GCN of width 64 over the Cora (left), DBLP (middle), and OGBN-ArXiv (right) vertex prediction datasets, with increasing percentage of removed edges (for each combination of dataset, edge sparsification algorithm, and percentage of removed edges, a separate GCN was trained and evaluated). WIS, designed to maintain the ability of a GNN to model interactions between input vertices, is compared against: *(i)* removing edges uniformly at random; *(ii)* a spectral sparsification method [93]; and *(iii)* an adaptation of UGS [24]. For Cora, we run both 2-WIS, which is compatible with the GNN's depth, and 1-WIS, which can be viewed as an approximation that admits a particularly efficient implementation (Algorithm 2). For DBLP and OGBN-ArXiv, due to their larger scale only 1-WIS is evaluated. Markers and error bars report means and standard deviations, respectively, taken over ten runs per configuration. Note that 1-WIS achieves results similar to 2-WIS, suggesting that the efficiency it brings does not come at a significant cost in performance. Appendix H provides further implementation details and experiments with additional GNN architectures (GIN and ResGCN) and datasets (Chameleon, Squirrel, and Amazon Computers). Code for reproducing the experiment is available at https://github.com/noamrazin/gnn_interactions.

forth. This is equivalent to sorting (in ascending order) the entries of each tuple separately, and then ordering the tuples lexicographically.

Algorithm 1 provides a self-contained description of the method attained by the foregoing choices. We refer to this method as $(L-1)$-Walk Index Sparsification (WIS), where the "$(L-1)$" indicates that only walks of length $L-1$ take part in the walk indices. Since $(L-1)$-walk indices can be computed by taking the $(L-1)$'th power of the graph's adjacency matrix, $(L-1)$-WIS runs in $\mathcal{O}(N|\mathcal{E}||\mathcal{V}|^3 \log(L))$ time and requires $\mathcal{O}(|\mathcal{E}||\mathcal{V}| + |\mathcal{V}|^2)$ memory, where $N$ is the number of edges to be removed. For large graphs a runtime cubic in the number of vertices can be restrictive. Fortunately, 1-WIS, which can be viewed as an approximation for $(L-1)$-WIS with $L > 2$, facilitates a particularly simple and efficient implementation based solely on vertex degrees, requiring only linear time and memory — see Algorithm 2 (whose equivalence to 1-WIS is explained in Appendix G). Specifically, 1-WIS runs in $\mathcal{O}(N|\mathcal{E}| + |\mathcal{V}|)$ time and requires $\mathcal{O}(|\mathcal{E}| + |\mathcal{V}|)$ memory.

## 5.2 Empirical Evaluation

Below is an empirical evaluation of WIS. For brevity, we defer to Appendix H some implementation details, as well as experiments with additional GNN architectures (GIN and ResGCN) and datasets (Chameleon [82], Squirrel [82], and Amazon Computers [92]).

Using depth $L = 3$ GNNs (with ReLU non-linearity), we evaluate over the Cora dataset both 2-WIS, which is compatible with the GNNs' depth, and 1-WIS, which can be viewed as an efficient approximation. Over the DBLP and OGBN-ArXiv datasets, due to their larger scale only 1-WIS is evaluated. Figure 3 (and Figure 8 in Appendix H) shows that WIS significantly outperforms the following alternative methods in terms of induced prediction accuracy: *(i)* a baseline in which edges are removed uniformly at random; *(ii)* a well-known spectral algorithm [93] designed to preserve the spectrum of the sparsified graph's Laplacian; and *(iii)* an adaptation of UGS [24] — a recent supervised approach for learning to prune edges.[11] Both 2-WIS and 1-WIS lead to higher test accuracies, while (as opposed to UGS) avoiding the need for labels, and for training a GNN over the original (non-sparsified) graph — a procedure which in some settings is prohibitively expensive in terms of runtime and memory. Interestingly, 1-WIS performs similarly to 2-WIS, indicating that the efficiency it brings does not come at a sizable cost in performance.

---

[11]UGS [24] jointly prunes input graph edges and GNN weights. For fair comparison, we adapt it to only remove edges.

# 6 Conclusion

## 6.1 Summary

GNNs are designed to model complex interactions between entities represented as vertices of a graph. The current paper provides the first theoretical analysis for their ability to do so. We proved that, given a partition of input vertices, the strength of interaction that can be modeled between its sides is controlled by the *walk index* — a graph-theoretical characteristic defined by the number of walks originating from the boundary of the partition. Experiments with common GNN architectures, *e.g.* GCN [59] and GIN [103], corroborated this result.

Our theory formalizes conventional wisdom by which GNNs can model stronger interaction between regions of the input graph that are more interconnected. More importantly, we showed that it facilitates a novel edge sparsification algorithm which preserves the ability of a GNN to model interactions when edges are removed. Our algorithm, named *Walk Index Sparsification* (*WIS*), is simple, computationally efficient, and in our experiments has markedly outperformed alternative methods in terms of induced prediction accuracy. More broadly, WIS showcases the potential of improving GNNs by theoretically analyzing the interactions they can model.

## 6.2 Limitations and Future Work

The theoretical analysis considers GNNs with product aggregation, which are less common in practice (*cf.* Section 3). We empirically demonstrated that its conclusions apply to more popular GNNs (Section 4.2), and derived a practical edge sparsification algorithm based on the theory (Section 5). Nonetheless, extending our analysis to additional aggregation functions is a worthy avenue to explore.

Our work also raises several interesting directions concerning WIS. A naive implementation of $(L-1)$-WIS has runtime cubic in the number of vertices (*cf.* Section 5.1). Since this can be restrictive for large-scale graphs, the evaluation in Section 5.2 mostly focused on 1-WIS, which can be viewed as an efficient approximation of $(L-1)$-WIS (its runtime and memory requirements are linear — see Section 5.1). Future work can develop efficient exact implementations of $(L-1)$-WIS (*e.g.* using parallelization) and investigate regimes where it outperforms 1-WIS in terms of induced prediction accuracy. Additionally, $(L-1)$-WIS is a specific instantiation of the general WIS scheme (given in Appendix F), tailored for preserving the ability to model interactions across certain partitions. Exploring other instantiations, as well as methods for automatically choosing the partitions across which the ability to model interactions is preserved, are valuable directions for further research.

## Acknowledgments and Disclosure of Funding

We thank Eshbal Hezroni for aid in preparing illustrative figures. This work was supported by a Google Research Scholar Award, a Google Research Gift, the Yandex Initiative in Machine Learning, the Israel Science Foundation (grant 1780/21), the Tel Aviv University Center for AI and Data Science, the Adelis Research Fund for Artificial Intelligence, Len Blavatnik and the Blavatnik Family Foundation, and Amnon and Anat Shashua. NR is supported by the Apple Scholars in AI/ML PhD fellowship.

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

## A    Formal Analysis: Quantifying the Ability of Graph Neural Networks to Model Interactions

We begin by upper bounding the separation ranks a GNN can achieve.

**Theorem 2.** *For an undirected graph $\mathcal{G}$ and $t \in \mathcal{V}$, let $f^{(\theta, \mathcal{G})}$ and $f^{(\theta, \mathcal{G}, t)}$ be the functions realized by depth $L$ graph and vertex prediction GNNs, respectively, with width $D_h$, learnable weights $\theta$, and product aggregation (Equations (2) to (5)). Then, for any $\mathcal{I} \subseteq \mathcal{V}$ and assignment of weights $\theta$ it holds that:*

$$\text{(graph prediction)} \quad \log\big(\text{sep}\big(f^{(\theta, \mathcal{G})}; \mathcal{I}\big)\big) \leq \log(D_h) \cdot \big(4 \underbrace{\rho_{L-1}(\mathcal{C}_\mathcal{I}, \mathcal{V})}_{\text{WI}_{L-1}(\mathcal{I})} + 1\big), \tag{6}$$

$$\text{(vertex prediction)} \quad \log\big(\text{sep}\big(f^{(\theta, \mathcal{G}, t)}; \mathcal{I}\big)\big) \leq \log(D_h) \cdot 4 \underbrace{\rho_{L-1}(\mathcal{C}_\mathcal{I}, \{t\})}_{\text{WI}_{L-1,t}(\mathcal{I})}. \tag{7}$$

*Proof sketch (proof in Appendix I.2).* In Appendix E, we show that the computations performed by a GNN with product aggregation can be represented as a *tensor network*. In brief, a tensor network is a weighted graph that describes a sequence of arithmetic operations known as tensor contractions (see Appendices E.1 and E.2 for a self-contained introduction to tensor networks). The tensor network corresponding to a GNN with product aggregation adheres to a tree structure — its leaves are associated with input vertex features and interior nodes embody the operations performed by the GNN. Importing machinery from tensor analysis literature, we prove that $\text{sep}(f^{(\theta, \mathcal{G})}; \mathcal{I})$ is upper bounded by a minimal cut weight in the corresponding tensor network, among cuts separating leaves associated with input vertices in $\mathcal{I}$ from leaves associated with input vertices in $\mathcal{I}^c$. Equation (6) then follows by finding such a cut in the tensor network with sufficiently low weight. Equation (7) is established analogously. $\square$

A natural question is whether the upper bounds in Theorem 2 are tight, *i.e.* whether separation ranks close to them can be attained. We show that nearly matching lower bounds hold for almost all assignments of weights $\theta$. To this end, we define *admissible subsets of $\mathcal{C}_\mathcal{I}$*, based on a notion of vertex subsets with *no repeating shared neighbors*.

**Definition 3.** We say that $\mathcal{I}, \mathcal{J} \subseteq \mathcal{V}$ *have no repeating shared neighbors* if every $k \in \mathcal{N}(\mathcal{I}) \cap \mathcal{N}(\mathcal{J})$ has only a single neighbor in each of $\mathcal{I}$ and $\mathcal{J}$, *i.e.* $|\mathcal{N}(k) \cap \mathcal{I}| = |\mathcal{N}(k) \cap \mathcal{J}| = 1$.

**Definition 4.** For $\mathcal{I} \subseteq \mathcal{V}$, we refer to $\mathcal{C} \subseteq \mathcal{C}_\mathcal{I}$ as an *admissible subset of $\mathcal{C}_\mathcal{I}$* if there exist $\mathcal{I}' \subseteq \mathcal{I}, \mathcal{J}' \subseteq \mathcal{I}^c$ with no repeating shared neighbors such that $\mathcal{C} = \mathcal{N}(\mathcal{I}') \cap \mathcal{N}(\mathcal{J}')$. We use $\mathcal{S}(\mathcal{I})$ to denote the set comprising all admissible subsets of $\mathcal{C}_\mathcal{I}$:

$$\mathcal{S}(\mathcal{I}) := \big\{ \mathcal{C} \subseteq \mathcal{C}_\mathcal{I} : \mathcal{C} \text{ is an admissible subset of } \mathcal{C}_\mathcal{I} \big\}.$$

Theorem 3 below establishes that almost all possible values for the network's weights lead the upper bounds in Theorem 2 to be tight, up to logarithmic terms and to the number of walks from $\mathcal{C}_\mathcal{I}$ being replaced with the number of walks from any single $\mathcal{C} \in \mathcal{S}(\mathcal{I})$. The extent to which $\mathcal{C}_\mathcal{I}$ can be covered by an admissible subset thus determines how tight the upper bounds are. Trivially, at least the shared neighbors of any $i \in \mathcal{I}, j \in \mathcal{I}^c$ can be covered, since $\mathcal{N}(i) \cap \mathcal{N}(j) \in \mathcal{S}(\mathcal{I})$. Appendix C shows that for various canonical graphs all of $\mathcal{C}_\mathcal{I}$, or a large part of it, can be covered by an admissible subset.

**Theorem 3.** *Consider the setting and notation of Theorem 2. Given $\mathcal{I} \subseteq \mathcal{V}$, for almost all assignments of weights $\theta$, i.e. for all but a set of Lebesgue measure zero, it holds that:*

$$\text{(graph prediction)} \quad \log\big(\text{sep}\big(f^{(\theta, \mathcal{G})}; \mathcal{I}\big)\big) \geq \max_{\mathcal{C} \in \mathcal{S}(\mathcal{I})} \log(\alpha_\mathcal{C}) \cdot \rho_{L-1}(\mathcal{C}, \mathcal{V}), \tag{8}$$

$$\text{(vertex prediction)} \quad \log\big(\text{sep}\big(f^{(\theta, \mathcal{G}, t)}; \mathcal{I}\big)\big) \geq \max_{\mathcal{C} \in \mathcal{S}(\mathcal{I})} \log(\alpha_{\mathcal{C}, t}) \cdot \rho_{L-1}(\mathcal{C}, \{t\}), \tag{9}$$

*where:*

$$\alpha_\mathcal{C} := \begin{cases} D^{1/\rho_0(\mathcal{C}, \mathcal{V})} & , \text{if } L = 1 \\ (D-1) \cdot \rho_{L-1}(\mathcal{C}, \mathcal{V})^{-1} + 1 & , \text{if } L \geq 2 \end{cases},$$

$$\alpha_{\mathcal{C}, t} := \begin{cases} D & , \text{if } L = 1 \\ (D-1) \cdot \rho_{L-1}(\mathcal{C}, \{t\})^{-1} + 1 & , \text{if } L \geq 2 \end{cases},$$

with $D := \min\{D_x, D_h\}$. If $\rho_{L-1}(\mathcal{C}, \mathcal{V}) = 0$ or $\rho_{L-1}(\mathcal{C}, \{t\}) = 0$, the respective lower bound (right hand side of Equation (8) or Equation (9)) is zero by convention.

*Proof sketch (proof in Appendix I.3).* Our proof follows a line similar to that used in [64, 100, 65] for lower bounding the separation rank of self-attention neural networks. The separation rank of any $f : (\mathbb{R}^{D_x})^{|\mathcal{V}|} \to \mathbb{R}$ can be lower bounded by examining its outputs over a grid of inputs. Specifically, for $M \in \mathbb{N}$ *template vectors* $\mathbf{v}^{(1)}, \ldots, \mathbf{v}^{(M)} \in \mathbb{R}^{D_x}$, we can create a *grid tensor* for $f$ by evaluating it over each point in $\{(\mathbf{v}^{(d_1)}, \ldots, \mathbf{v}^{(d_{|\mathcal{V}|})})\}_{d_1, \ldots, d_{|\mathcal{V}|}=1}^{M}$ and storing the outcomes in a tensor with $|\mathcal{V}|$ axes of dimension $M$ each. Arranging the grid tensor as a matrix $\mathbf{B}(f)$ where rows correspond to axes indexed by $\mathcal{I}$ and columns correspond to the remaining axes, we show that $\mathrm{rank}(\mathbf{B}(f)) \leq \mathrm{sep}(f; \mathcal{I})$. The proof proceeds by establishing that for almost every assignment of $\theta$, there exist template vectors with which $\log(\mathrm{rank}(\mathbf{B}(f^{(\theta, \mathcal{G})})))$ and $\log(\mathrm{rank}(\mathbf{B}(f^{(\theta, \mathcal{G}, t)})))$ are greater than (or equal to) the right hand sides of Equations (8) and (9), respectively. $\qquad\square$

**Directed edges and multiple edge types**   Appendix D generalizes Theorems 2 and 3 to the case of graphs with directed edges and an arbitrary number of edge types.

# B   Related Work

**Expressivity of GNNs**   The expressivity of GNNs has been predominantly evaluated through ability to distinguish non-isomorphic graphs, as measured by correspondence to Weisfeiler-Leman (WL) graph isomorphism tests (see [75] for a recent survey). [103, 74] instigated this thread of research, establishing that message-passing GNNs are at most as powerful as the WL algorithm, and can match it under certain technical conditions. Subsequently, architectures surpassing WL were proposed, with expressivity measured via higher-order WL variants (see, *e.g.*, [74, 72, 25, 41, 6, 15, 10, 42, 17, 80]). Another line of inquiry regards universality among continuous permutation invariant or equivariant functions [73, 55, 69, 3, 42]. [25] showed that distinguishing non-isomorphic graphs and universality are, in some sense, equivalent. Lastly, there exist analyses of expressivity focused on the frequency response of GNNs [77, 5] and their capacity to compute specific graph functions, *e.g.* moments, shortest paths, and substructure counting [35, 9, 39, 69, 26, 23, 17].

Although a primary purpose of GNNs is to model interactions between vertices, none of the past works formally characterize their ability to do so, as our theory does.[12] The current work thus provides a novel perspective on the expressive power of GNNs. Furthermore, a major limitation of existing approaches — in particular, proofs of equivalence to WL tests and universality — is that they often operate in asymptotic regimes of unbounded network width or depth. Consequently, they fall short of addressing which type of functions can be realized by GNNs of practical size. In contrast, we characterize how the modeled interactions depend on both the input graph structure and the neural network architecture (width and depth). As shown in Section 5, this facilitates designing an efficient and effective edge sparsification algorithm.

**Measuring modeled interactions via separation rank**   Separation rank (Section 2.2) has been paramount to the study of interactions modeled by certain convolutional, recurrent, and self-attention neural networks. It enabled theoretically analyzing how different architectural parameters impact expressivity [32, 29, 30, 34, 7, 90, 62, 61, 56, 57, 64, 100, 65] and implicit regularization [83, 84, 85].[13] On the practical side, insights brought forth by separation rank led to tools for improving performance, including: guidelines for architecture design [30, 62, 64, 100], pretraining schemes [65], and regularizers for countering locality in convolutional neural networks [85]. We employ separation rank

---

[12]In [21], the mutual information between the embedding of a vertex and the embeddings of its neighbors was proposed as a measure of interaction. However, this measure is inherently local and allows reasoning only about the impact of neighboring nodes on each other in a GNN layer. In contrast, separation rank formulates the strength of interaction the whole GNN models across any partition of an input graph's vertices.

[13]We note that, over a two-dimensional grid graph, a message-passing GNN can be viewed as a convolutional neural network with overlapping convolutional windows. Similarly, over a chain graph, it can be viewed as a bidirectional recurrent neural network. Thus, for these special cases, our separation rank bounds (delivered in Section 4) extend those of [30, 62, 56, 61], which consider convolutional neural networks with non-overlapping convolutional windows and unidirectional recurrent neural networks.

for studying the interactions GNNs model between vertices, and similarly provide both theoretical insights and a practical application — edge sparsification algorithm (Section 5).

**Edge sparsification**  Computations over large-scale graphs can be prohibitively expensive in terms of runtime and memory. As a result, various methods were proposed for sparsifying graphs by removing edges while attempting to maintain structural properties, such as distances between vertices [11, 48], graph Laplacian spectrum [93, 86], and vertex degree distribution [98], or outcomes of graph analysis and clustering algorithms [87, 20]. Most relevant to our work are recent edge sparsification methods aiming to preserve the prediction accuracy of GNNs as the number of removed edges increases [67, 24]. These methods require training a GNN over the original (non-sparsified) graph, hence only inference costs are reduced. Guided by our theory, in Section 5 we propose *Walk Index Sparsification (WIS)* — an edge sparsification algorithm that preserves expressive power in terms of ability to model interactions. WIS improves efficiency for both training and inference. Moreover, comparisons with the spectral algorithm of [93] and a recent method from [24] demonstrate that WIS brings about higher prediction accuracies across edge sparsity levels.

## C  Tightness of Upper Bounds for Separation Rank

Theorem 2 upper bounds the separation rank with respect to $\mathcal{I} \subseteq \mathcal{V}$ of a depth $L$ GNN with product aggregation. According to it, under the setting of graph prediction, the separation rank is largely capped by the $(L-1)$-walk index of $\mathcal{I}$, *i.e.* the number of length $L-1$ walks from $\mathcal{C}_{\mathcal{I}}$ — the set of vertices with an edge crossing the partition $(\mathcal{I}, \mathcal{I}^c)$. Similarly, for prediction over $t \in \mathcal{V}$, separation rank is largely capped by the $(L-1, t)$-walk index of $\mathcal{I}$, which takes into account only length $L-1$ walks from $\mathcal{C}_{\mathcal{I}}$ ending at $t$. Theorem 3 provides matching lower bounds, up to logarithmic terms and to the number of walks from $\mathcal{C}_{\mathcal{I}}$ being replaced with the number of walks from any single admissible subset $\mathcal{C} \in \mathcal{S}(\mathcal{I})$ (Definition 4). Hence, the match between the upper and lower bounds is determined by the portion of $\mathcal{C}_{\mathcal{I}}$ that can be covered by an admissible subset.

In this appendix, to shed light on the tightness of the upper bounds, we present several concrete examples on which a significant portion of $\mathcal{C}_{\mathcal{I}}$ can be covered by an admissible subset.

**Complete graph**  Suppose that every two vertices are connected by an edge, *i.e.* $\mathcal{E} = \{\{i, j\} : i, j \in \mathcal{V}\}$. For any non-empty $\mathcal{I} \subsetneq \mathcal{V}$, clearly $\mathcal{C}_{\mathcal{I}} = \mathcal{N}(\mathcal{I}) \cap \mathcal{N}(\mathcal{I}^c) = \mathcal{V}$. In this case, $\mathcal{C}_{\mathcal{I}} = \mathcal{V} \in \mathcal{S}(\mathcal{I})$, meaning $\mathcal{C}_{\mathcal{I}}$ is an admissible subset of itself. To see it is so, notice that for any $i \in \mathcal{I}, j \in \mathcal{I}^c$, all vertices are neighbors of both $\mathcal{I}' := \{i\}$ and $\mathcal{J}' := \{j\}$, which trivially have no repeating shared neighbors (Definition 3). Thus, up to a logarithmic factor, the upper and lower bounds from Theorems 2 and 3 coincide.

**Chain graph**  Suppose that $\mathcal{E} = \{\{i, i+1\} : i \in [|\mathcal{V}| - 1]\} \cup \{\{i, i\} : i \in \mathcal{V}\}$. For any non-empty $\mathcal{I} \subsetneq \mathcal{V}$, at least half of the vertices in $\mathcal{C}_{\mathcal{I}}$ can be covered by an admissible subset. That is, there exists $\mathcal{C} \in \mathcal{S}(\mathcal{I})$ satisfying $|\mathcal{C}| \geq 2^{-1} \cdot |\mathcal{C}_{\mathcal{I}}|$. For example, such $\mathcal{C}$ can be constructed algorithmically as follows. Let $\mathcal{I}', \mathcal{J}' = \emptyset$. Starting from $k = 1$, if $\{k, k+1\} \subseteq \mathcal{C}_{\mathcal{I}}$ and one of $\{k, k+1\}$ is in $\mathcal{I}$ while the other is in $\mathcal{I}^c$, then assign $\mathcal{I}' \leftarrow \mathcal{I}' \cup (\{k, k+1\} \cap \mathcal{I})$, $\mathcal{J}' \leftarrow \mathcal{J}' \cup (\{k, k+1\} \cap \mathcal{I}^c)$, and $k \leftarrow k + 3$. That is, add each of $\{k, k+1\}$ to either $\mathcal{I}'$ if it is in $\mathcal{I}$ or $\mathcal{J}'$ if it is in $\mathcal{I}^c$, and skip vertex $k + 2$. Otherwise, set $k \leftarrow k + 1$. The process terminates once $k > |\mathcal{V}| - 1$. By construction, $\mathcal{I}' \subseteq \mathcal{I}$ and $\mathcal{J}' \subseteq \mathcal{I}^c$, implying that $\mathcal{N}(\mathcal{I}') \cap \mathcal{N}(\mathcal{J}') \subseteq \mathcal{C}_{\mathcal{I}}$. Due to the chain graph structure, $\mathcal{I}' \cup \mathcal{J}' \subseteq \mathcal{N}(\mathcal{I}') \cap \mathcal{N}(\mathcal{J}')$ and $\mathcal{I}'$ and $\mathcal{J}'$ have no repeating shared neighbors (Definition 3). Furthermore, for every pair of vertices from $\mathcal{C}_{\mathcal{I}}$ added to $\mathcal{I}'$ and $\mathcal{J}'$, we can miss at most two other vertices from $\mathcal{C}_{\mathcal{I}}$. Thus, $\mathcal{C} := \mathcal{N}(\mathcal{I}') \cap \mathcal{N}(\mathcal{J}')$ is an admissible subset of $\mathcal{C}_{\mathcal{I}}$ satisfying $|\mathcal{C}| \geq 2^{-1} \cdot |\mathcal{C}_{\mathcal{I}}|$.

**General graph**  For an arbitrary graph and non-empty $\mathcal{I} \subsetneq \mathcal{V}$, an admissible subset of $\mathcal{C}_{\mathcal{I}}$ can be obtained by taking any sequence of pairs $(i_1, j_1), \ldots, (i_M, j_M) \in \mathcal{I} \times \mathcal{I}^c$ with no shared neighbors, in the sense that $[\mathcal{N}(i_m) \cup \mathcal{N}(j_m)] \cap [\mathcal{N}(i_{m'}) \cup \mathcal{N}(j_{m'})] = \emptyset$ for all $m \neq m' \in [M]$. Defining $\mathcal{I}' := \{i_1, \ldots, i_M\}$ and $\mathcal{J}' := \{j_1, \ldots, j_M\}$, by construction they do not have repeating shared neighbors (Definition 3), and so $\mathcal{N}(\mathcal{I}') \cap \mathcal{N}(\mathcal{J}') \in \mathcal{S}(\mathcal{I})$. In particular, the shared neighbors of each pair are covered by $\mathcal{N}(\mathcal{I}') \cap \mathcal{N}(\mathcal{J}')$, *i.e.* $\cup_{m=1}^{M} \mathcal{N}(i_m) \cap \mathcal{N}(j_m) \subseteq \mathcal{N}(\mathcal{I}') \cap \mathcal{N}(\mathcal{J}')$.

## D  Extension of Analysis to Directed Graphs With Multiple Edge Types

In this appendix, we generalize the separation rank bounds from Theorems 2 and 3 to directed graphs with multiple edge types.

Let $\mathcal{G} = (\mathcal{V}, \mathcal{E}, \tau)$ be a directed graph with vertices $\mathcal{V} = [|\mathcal{V}|]$, edges $\mathcal{E} \subseteq \{(i,j) : i, j \in \mathcal{V}\}$, and a map $\tau : \mathcal{E} \to [Q]$ from edges to one of $Q \in \mathbb{N}$ edge types. For $i \in \mathcal{V}$, let $\mathcal{N}_{in}(i) := \{j \in \mathcal{V} : (j,i) \in \mathcal{E}\}$ be its *incoming neighbors* and $\mathcal{N}_{out}(i) := \{j \in \mathcal{V} : (i,j) \in \mathcal{E}\}$ be its *outgoing neighbors*. For $\mathcal{I} \subseteq \mathcal{V}$, we denote $\mathcal{N}_{in}(\mathcal{I}) := \cup_{i \in \mathcal{I}} \mathcal{N}_{in}(i)$ and $\mathcal{N}_{out}(\mathcal{I}) := \cup_{i \in \mathcal{I}} \mathcal{N}_{out}(i)$. As customary in the context of GNNs, we assume the existence of all self-loops (*cf.* Section 2.1).

Message-passing GNNs (Section 3) operate identically over directed and undirected graphs, except that in directed graphs the hidden embedding of a vertex is updated only according to its incoming neighbors. For handling multiple edge types, common practice is to use different weight matrices per type in the GNN's update rule (*cf.* [49, 88]). Hence, we consider the following update rule for directed graphs with multiple edge types, replacing that from Equation (2):

$$\mathbf{h}^{(l,i)} = \text{AGGREGATE}\Big(\big\{\!\!\big\{\mathbf{W}^{(l,\tau(j,i))}\mathbf{h}^{(l-1,j)} : j \in \mathcal{N}_{in}(i)\big\}\!\!\big\}\Big), \tag{10}$$

where $(\mathbf{W}^{(1,q)} \in \mathbb{R}^{D_h \times D_x})_{q \in [Q]}$ and $(\mathbf{W}^{(l,q)} \in \mathbb{R}^{D_h \times D_h})_{l \in \{2,\ldots,L\}, q \in [Q]}$ are learnable weight matrices.

In our analysis for undirected graphs (Appendix A), a central concept is $\mathcal{C}_{\mathcal{I}}$ — the set of vertices with an edge crossing the partition induced by $\mathcal{I} \subseteq \mathcal{V}$. Due to the existence of self-loops it is equal to the shared neighbors of $\mathcal{I}$ and $\mathcal{I}^c$, *i.e.* $\mathcal{C}_{\mathcal{I}} = \mathcal{N}(\mathcal{I}) \cap \mathcal{N}(\mathcal{I}^c)$. We generalize this concept to directed graphs, defining $\mathcal{C}_{\mathcal{I}}^{\rightarrow}$ to be the set of vertices with an incoming edge from the other side of the partition induced by $\mathcal{I}$, *i.e.* $\mathcal{C}_{\mathcal{I}}^{\rightarrow} := \{i \in \mathcal{I} : \mathcal{N}_{in}(i) \cap \mathcal{I}^c \neq \emptyset\} \cup \{j \in \mathcal{I}^c : \mathcal{N}_{in}(j) \cap \mathcal{I} \neq \emptyset\}$. Due to the existence of self-loops it is given by $\mathcal{C}_{\mathcal{I}}^{\rightarrow} = \mathcal{N}_{out}(\mathcal{I}) \cap \mathcal{N}_{out}(\mathcal{I}^c)$. Indeed, for undirected graphs $\mathcal{C}_{\mathcal{I}}^{\rightarrow} = \mathcal{C}_{\mathcal{I}}$.

With the definition of $\mathcal{C}_{\mathcal{I}}^{\rightarrow}$ in place, Theorem 4 upper bounds the separation ranks a GNN can achieve over directed graphs with multiple edge types. A technical subtlety is that the bounds depend on walks of lengths $l = L-1, L-2, \ldots, 0$, while those in Theorem 2 for undirected graphs depend only on walks of length $L-1$. As shown in the proof of Theorem 2, this dependence exists in undirected graphs as well. Though, in undirected graphs with self-loops, the number of length $l \in \mathbb{N}$ walks from $\mathcal{C}_{\mathcal{I}}$ decays exponentially as $l$ decreases. One can therefore replace the sum over walk lengths with walks of length $L-1$ (up to a multiplicative constant). By contrast, in directed graphs this is not true in general, *e.g.*, when $\mathcal{C}_{\mathcal{I}}^{\rightarrow}$ contains only vertices with no outgoing edges (besides self-loops).

**Theorem 4.** *For a directed graph with multiple edge types $\mathcal{G}$ and $t \in \mathcal{V}$, let $f^{(\theta,\mathcal{G})}$ and $f^{(\theta,\mathcal{G},t)}$ be the functions realized by depth $L$ graph and vertex prediction GNNs, respectively, with width $D_h$, learnable weights $\theta$, and product aggregation (Equations (3) to (5) and (10)). Then, for any $\mathcal{I} \subseteq \mathcal{V}$ and assignment of weights $\theta$ it holds that:*

$$\textit{(graph prediction)} \quad \log\big(\text{sep}\big(f^{(\theta,\mathcal{G})}; \mathcal{I}\big)\big) \leq \log(D_h) \cdot \Big(\sum_{l=1}^{L} \rho_{L-l}(\mathcal{C}_{\mathcal{I}}^{\rightarrow}, \mathcal{V}) + 1\Big), \tag{11}$$

$$\textit{(vertex prediction)} \quad \log\big(\text{sep}\big(f^{(\theta,\mathcal{G},t)}; \mathcal{I}\big)\big) \leq \log(D_h) \cdot \sum_{l=1}^{L} \rho_{L-l}(\mathcal{C}_{\mathcal{I}}^{\rightarrow}, \{t\}). \tag{12}$$

*Proof sketch (proof in Appendix I.4).* The proof follows a line identical to that of Theorem 2, only requiring adjusting definitions from undirected graphs to directed graphs with multiple edge types. □

Towards lower bounding separation ranks, we generalize the definitions of vertex subsets with no repeating shared neighbors (Definition 3) and admissible subsets of $\mathcal{C}_{\mathcal{I}}$ (Definition 4) to directed graphs.

**Definition 5.** We say that $\mathcal{I}, \mathcal{J} \subseteq \mathcal{V}$ *have no outgoing repeating shared neighbors* if every $k \in \mathcal{N}_{out}(\mathcal{I}) \cap \mathcal{N}_{out}(\mathcal{J})$ has only a single incoming neighbor in each of $\mathcal{I}$ and $\mathcal{J}$, *i.e.* $|\mathcal{N}_{in}(k) \cap \mathcal{I}| = |\mathcal{N}_{in}(k) \cap \mathcal{J}| = 1$.

**Definition 6.** For $\mathcal{I} \subseteq \mathcal{V}$, we refer to $\mathcal{C} \subseteq \mathcal{C}_{\mathcal{I}}^{\rightarrow}$ as an *admissible subset of $\mathcal{C}_{\mathcal{I}}^{\rightarrow}$* if there exist $\mathcal{I}' \subseteq \mathcal{I}, \mathcal{J}' \subseteq \mathcal{I}^c$ with no outgoing repeating shared neighbors such that $\mathcal{C} = \mathcal{N}_{out}(\mathcal{I}') \cap \mathcal{N}_{out}(\mathcal{J}')$. We use $\mathcal{S}^{\rightarrow}(\mathcal{I})$ to denote the set comprising all admissible subsets of $\mathcal{C}_{\mathcal{I}}^{\rightarrow}$:

$$\mathcal{S}^{\rightarrow}(\mathcal{I}) := \big\{\mathcal{C} \subseteq \mathcal{C}_{\mathcal{I}}^{\rightarrow} : \mathcal{C} \text{ is an admissible subset of } \mathcal{C}_{\mathcal{I}}^{\rightarrow}\big\}.$$

Theorem 5 generalizes the lower bounds from Theorem 3 to directed graphs with multiple edge types.

**Theorem 5.** *Consider the setting and notation of Theorem 4. Given $I \subseteq \mathcal{V}$, for almost all assignments of weights $\theta$, i.e. for all but a set of Lebesgue measure zero, it holds that:*

$$(\text{graph prediction}) \quad \log\big(\text{sep}\big(f^{(\theta,\mathcal{G})}; \mathcal{I}\big)\big) \geq \max_{\mathcal{C} \in \mathcal{S}^{\rightarrow}(\mathcal{I})} \log(\alpha_{\mathcal{C}}) \cdot \rho_{L-1}(\mathcal{C}, \mathcal{V}), \tag{13}$$

$$(\text{vertex prediction}) \quad \log\big(\text{sep}\big(f^{(\theta,\mathcal{G},t)}; \mathcal{I}\big)\big) \geq \max_{\mathcal{C} \in \mathcal{S}^{\rightarrow}(\mathcal{I})} \log(\alpha_{\mathcal{C},t}) \cdot \rho_{L-1}(\mathcal{C}, \{t\}), \tag{14}$$

*where:*

$$\alpha_{\mathcal{C}} := \begin{cases} D^{1/\rho_0(\mathcal{C}, \mathcal{V})} & , \text{if } L = 1 \\ (D-1) \cdot \rho_{L-1}(\mathcal{C}, \mathcal{V})^{-1} + 1 & , \text{if } L \geq 2 \end{cases},$$

$$\alpha_{\mathcal{C},t} := \begin{cases} D & , \text{if } L = 1 \\ (D-1) \cdot \rho_{L-1}(\mathcal{C}, \{t\})^{-1} + 1 & , \text{if } L \geq 2 \end{cases},$$

*with $D := \min\{D_x, D_h\}$. If $\rho_{L-1}(\mathcal{C}, \mathcal{V}) = 0$ or $\rho_{L-1}(\mathcal{C}, \{t\}) = 0$, the respective lower bound (right hand side of Equation (13) or Equation (14)) is zero by convention.*

*Proof sketch (proof in Appendix I.5).* The proof follows a line identical to that of Theorem 3, only requiring adjusting definitions from undirected graphs to directed graphs with multiple edge types. □

# E   Representing Graph Neural Networks With Product Aggregation as Tensor Networks

In this appendix, we prove that GNNs with product aggregation (Section 3) can be represented through tensor networks — a graphical language for expressing tensor contractions, widely used in quantum mechanics literature for modeling quantum states (*cf.* [97]). This representation facilitates upper bounding the separation ranks of a GNN with product aggregation (proofs for Theorem 2 and its extension in Appendix D), and is delivered in Appendix E.3. We note that analogous tensor network representations were shown for variants of recurrent and convolutional neural networks [61, 62]. For the convenience of the reader, we lay out basic concepts from the field of tensor analysis in Appendix E.1 and provide a self-contained introduction to tensor networks in Appendix E.2 (see [79] for a more in-depth treatment).

## E.1   Primer on Tensor Analysis

For our purposes, a *tensor* is simply a multi-dimensional array. The *order* of a tensor is its number of axes, which are typically called *modes* (*e.g.* a vector is an order one tensor and a matrix is an order two tensor). The *dimension* of a mode refers to its length, *i.e.* the number of values it can be indexed with. For an order $N \in \mathbb{N}$ tensor $\mathcal{A} \in \mathbb{R}^{D_1 \times \cdots \times D_N}$ with modes of dimensions $D_1, \ldots, D_N \in \mathbb{N}$, we will denote by $\mathcal{A}_{d_1,\ldots,d_N}$ its $(d_1, \ldots, d_N)$'th entry, where $(d_1, \ldots, d_N) \in [D_1] \times \cdots \times [D_N]$.

It is possible to rearrange tensors into matrices — a process known as *matricization*. The matricization of $\mathcal{A}$ with respect to $\mathcal{I} \subseteq [N]$, denoted $[\![\mathcal{A}; \mathcal{I}]\!] \in \mathbb{R}^{\prod_{i \in \mathcal{I}} D_i \times \prod_{j \in \mathcal{I}^c} D_j}$ is its arrangement as a matrix where rows correspond to modes indexed by $\mathcal{I}$ and columns correspond to the remaining modes. Specifically, denoting the elements in $\mathcal{I}$ by $i_1 < \cdots < i_{|\mathcal{I}|}$ and those in $\mathcal{I}^c$ by $j_1 < \cdots < j_{|\mathcal{I}^c|}$, the matricization $[\![\mathcal{A}; \mathcal{I}]\!]$ holds the entries of $\mathcal{A}$ such that $\mathcal{A}_{d_1,\ldots,d_N}$ is placed in row index $1 + \sum_{l=1}^{|\mathcal{I}|}(d_{i_l} - 1) \prod_{l'=l+1}^{|\mathcal{I}|} D_{i_{l'}}$ and column index $1 + \sum_{l=1}^{|\mathcal{I}^c|}(d_{j_l} - 1) \prod_{l'=l+1}^{|\mathcal{I}^c|} D_{j_{l'}}$.

Tensors with modes of the same dimension can be combined via *contraction* — a generalization of matrix multiplication. It will suffice to consider contractions where one of the modes being contracted is the last mode of its tensor.

**Definition 7.** Let $\mathcal{A} \in \mathbb{R}^{D_1 \times \cdots \times D_N}, \mathcal{B} \in \mathbb{R}^{D_1' \times \cdots \times D_{N'}'}$ for orders $N, N' \in \mathbb{N}$ and mode dimensions $D_1, \ldots, D_N, D_1', \ldots, D_{N'}' \in \mathbb{N}$ satisfying $D_n = D_{N'}'$ for some $n \in [N]$. The *mode-$n$ contraction* of $\mathcal{A}$ with $\mathcal{B}$, denoted $\mathcal{A} *_n \mathcal{B} \in \mathbb{R}^{D_1 \times \cdots \times D_{n-1} \times D_1' \times \cdots \times D_{N'-1}' \times D_{n+1} \times \cdots \times D_N}$, is given element-wise by:

$$(\mathcal{A} *_n \mathcal{B})_{d_1,\ldots,d_{n-1},d_1',\ldots,d_{N'-1}',d_{n+1},\ldots,d_N} = \sum_{d_n=1}^{D_n} \mathcal{A}_{d_1,\ldots,d_N} \cdot \mathcal{B}_{d_1',\ldots,d_{N'-1}',d_n},$$

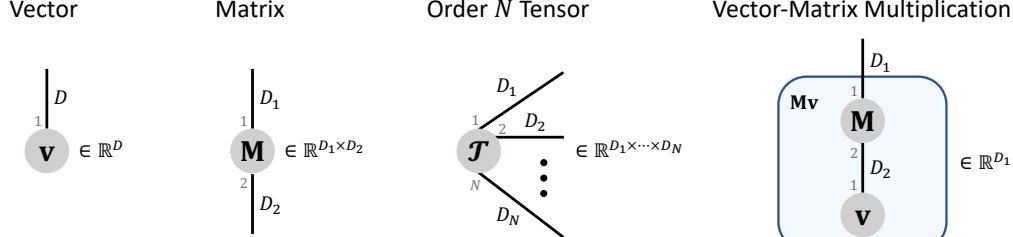

**Figure 4:** Tensor network diagrams of (from left to right): a vector $\mathbf{v} \in \mathbb{R}^D$, matrix $\mathbf{M} \in \mathbb{R}^{D_1 \times D_2}$, order $N \in \mathbb{N}$ tensor $\mathcal{T} \in \mathbb{R}^{D_1 \times \cdots \times D_N}$, and vector-matrix multiplication $\mathbf{Mv} \in \mathbb{R}^{D_1}$. The mode index associated with a leg's end point is specified in gray, and the weight of the leg, specified in black, determines the mode dimension.

for all $d_1 \in [D_1], \ldots, d_{n-1} \in [D_{n-1}], d'_1 \in [D'_1], \ldots, d'_{N'-1} \in [D'_{N'-1}], d_{n+1} \in [D_{n+1}], \ldots, d_N \in [D_N]$.

For example, the mode-2 contraction of $\mathbf{A} \in \mathbb{R}^{D_1 \times D_2}$ with $\mathbf{B} \in \mathbb{R}^{D'_1 \times D_2}$ boils down to multiplying $\mathbf{A}$ with $\mathbf{B}^\top$ from the right, *i.e.* $\mathbf{A} *_2 \mathbf{B} = \mathbf{AB}^\top$. It is oftentimes convenient to jointly contract multiple tensors. Given an order $N$ tensor $\mathcal{A}$ and $M \in \mathbb{N}_{\leq N}$ tensors $\mathcal{B}^{(1)}, \ldots, \mathcal{B}^{(M)}$, we use $\mathcal{A} *_{i \in [M]} \mathcal{B}^{(i)}$ to denote the contraction of $\mathcal{A}$ with $\mathcal{B}^{(1)}, \ldots, \mathcal{B}^{(M)}$ in modes $1, \ldots, M$, respectively (assuming mode dimensions are such that the contractions are well-defined).

## E.2 Tensor Networks

A *tensor network* is an undirected weighted graph $\mathcal{T} = (\mathcal{V}_\mathcal{T}, \mathcal{E}_\mathcal{T}, w_\mathcal{T})$ that describes a sequence of tensor contractions (Definition 7), with vertices $\mathcal{V}_\mathcal{T}$, edges $\mathcal{E}_\mathcal{T}$, and a function mapping edges to natural weights $w_\mathcal{T} : \mathcal{E}_\mathcal{T} \to \mathbb{N}$. We will only consider tensor networks that are connected. To avoid confusion with vertices and edges of a GNN's input graph, and in accordance with tensor network terminology, we refer by *nodes* and *legs* to the vertices and edges of a tensor network, respectively.

Every node in a tensor network is associated with a tensor, whose order is equal to the number of legs emanating from the node. Each end point of a leg is associated with a mode index, and the leg's weight determines the dimension of the corresponding tensor mode. That is, an end point of $e \in \mathcal{E}_\mathcal{T}$ is a pair $(\mathcal{A}, n) \in \mathcal{V}_\mathcal{T} \times \mathbb{N}$, with $n$ ranging from one to the order of $\mathcal{A}$, and $w_\mathcal{T}(e)$ is the dimension of $\mathcal{A}$ in mode $n$. A leg can either connect two nodes or be connected to a node on one end and be loose on the other end. If two nodes are connected by a leg, their associated tensors are contracted together in the modes specified by the leg. Legs with a loose end are called *open legs*. The number of open legs is exactly the order of the tensor produced by executing all contractions in the tensor network, *i.e.* by contracting the tensor network. Figure 4 presents exemplar tensor network diagrams of a vector, matrix, order $N \in \mathbb{N}$ tensor, and vector-matrix multiplication.

## E.3 Tensor Networks Corresponding to Graph Neural Networks With Product Aggregation

Fix some undirected graph $\mathcal{G}$ and learnable weights $\theta = (\mathbf{W}^{(1)}, \ldots, \mathbf{W}^{(L)}, \mathbf{W}^{(o)})$. Let $f^{(\theta, \mathcal{G})}$ and $f^{(\theta, \mathcal{G}, t)}$, for $t \in \mathcal{V}$, be the functions realized by depth $L$ graph and vertex prediction GNNs, respectively, with width $D_h$ and product aggregation (Equations (2) to (5)). For $\mathbf{X} = (\mathbf{x}^{(1)}, \ldots, \mathbf{x}^{(|\mathcal{V}|)}) \in \mathbb{R}^{D_x \times |\mathcal{V}|}$, we construct tensor networks $\mathcal{T}(\mathbf{X})$ and $\mathcal{T}^{(t)}(\mathbf{X})$ whose contraction yields $f^{(\theta, \mathcal{G})}(\mathbf{X})$ and $f^{(\theta, \mathcal{G}, t)}(\mathbf{X})$, respectively. Both $\mathcal{T}(\mathbf{X})$ and $\mathcal{T}^{(t)}(\mathbf{X})$ adhere to a tree structure, where each leaf node is associated with a vertex feature vector, *i.e.* one of $\mathbf{x}^{(1)}, \ldots, \mathbf{x}^{(|\mathcal{V}|)}$, and each interior node is associated with a weight matrix from $\mathbf{W}^{(1)}, \ldots, \mathbf{W}^{(L)}, \mathbf{W}^{(o)}$ or a $\delta$-*tensor* with modes of dimension $D_h$, holding ones on its hyper-diagonal and zeros elsewhere. We denote an order $N \in \mathbb{N}$ tensor of the latter type by $\delta^{(N)} \in \mathbb{R}^{D_h \times \cdots \times D_h}$, *i.e.* $\delta^{(N)}_{d_1, \ldots, d_N} = 1$ if $d_1 = \cdots = d_N$ and $\delta^{(N)}_{d_1, \ldots, d_N} = 0$ otherwise for all $d_1, \ldots, d_N \in [D_h]$.

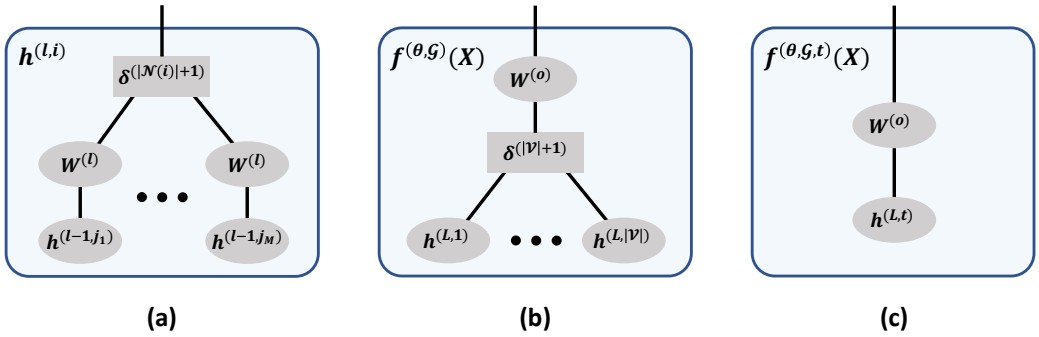

Figure 5: Tensor network diagrams of the operations performed by GNNs with product aggregation (Section 3). **(a)** Hidden embedding update (*cf.* Equations (2) and (5)): $\mathbf{h}^{(l,i)} = (\mathbf{W}^{(l)}\mathbf{h}^{(l-1,j_1)}) \odot \cdots \odot (\mathbf{W}^{(l)}\mathbf{h}^{(l-1,j_M)})$, where $\mathcal{N}(i) = \{j_1, \ldots, j_M\}$, for $l \in [L], i \in \mathcal{V}$. **(b)** Output layer for graph prediction (*cf.* Equations (3) and (5)): $f^{(\theta,\mathcal{G})}(\mathbf{X}) = \mathbf{W}^{(o)}(\mathbf{h}^{(L,1)} \odot \cdots \odot \mathbf{h}^{(L,|\mathcal{V}|)})$. **(c)** Output layer for vertex prediction over $t \in \mathcal{V}$ (*cf.* Equation (4)): $f^{(\theta,\mathcal{G},t)}(\mathbf{X}) = \mathbf{W}^{(o)}\mathbf{h}^{(L,t)}$. We draw nodes associated with $\delta$-tensors as rectangles to signify their special (hyper-diagonal) structure, and omit leg weights to avoid clutter (legs connected to $\mathbf{h}^{(0,i)} = \mathbf{x}^{(i)}$, for $i \in \mathcal{V}$, have weight $D_x$ while all other legs have weight $D_h$).

Intuitively, $\mathcal{T}(\mathbf{X})$ and $\mathcal{T}^{(t)}(\mathbf{X})$ embody unrolled computation trees, describing the operations performed by the respective GNNs through tensor contractions. Let $\mathbf{h}^{(l,i)} = \odot_{j \in \mathcal{N}(i)}(\mathbf{W}^{(l)}\mathbf{h}^{(l-1,j)})$ be the hidden embedding of $i \in \mathcal{V}$ at layer $l \in [L]$ (recall $\mathbf{h}^{(0,j)} = \mathbf{x}^{(j)}$ for $j \in \mathcal{V}$), and denote $\mathcal{N}(i) = \{j_1, \ldots, j_M\}$. We can describe $\mathbf{h}^{(l,i)}$ as the outcome of contracting each $\mathbf{h}^{(l-1,j_1)}, \ldots, \mathbf{h}^{(l-1,j_M)}$ with $\mathbf{W}^{(l)}$, *i.e.* computing $\mathbf{W}^{(l)}\mathbf{h}^{(l-1,j_1)}, \ldots, \mathbf{W}^{(l)}\mathbf{h}^{(l-1,j_M)}$, followed by contracting the resulting vectors with $\delta^{(|\mathcal{N}(i)|+1)}$, which induces product aggregation (see Figure 5(a)). Furthermore, in graph prediction, the output layer producing $f^{(\theta,\mathcal{G})}(\mathbf{X}) = \mathbf{W}^{(o)}(\odot_{i \in \mathcal{V}}\mathbf{h}^{(L,i)})$ amounts to contracting $\mathbf{h}^{(L,1)}, \ldots, \mathbf{h}^{(L,|\mathcal{V}|)}$ with $\delta^{(|\mathcal{V}|+1)}$, and subsequently contracting the resulting vector with $\mathbf{W}^{(o)}$ (see Figure 5(b)); while for vertex prediction, $f^{(\theta,\mathcal{G},t)}(\mathbf{X}) = \mathbf{W}^{(o)}\mathbf{h}^{(L,t)}$ is a contraction of $\mathbf{h}^{(L,t)}$ with $\mathbf{W}^{(o)}$ (see Figure 5(c)).

Overall, every layer in a GNN with product aggregation admits a tensor network formulation given the outputs of the previous layer. Thus, we can construct a tree tensor network for the whole GNN by starting from the output layer — Figure 5(b) for graph prediction or Figure 5(c) for vertex prediction — and recursively expanding nodes associated with $\mathbf{h}^{(l,i)}$ according to Figure 5(a), for $l = L, \ldots, 1$ and $i \in \mathcal{V}$. A technical subtlety is that each $\mathbf{h}^{(l,i)}$ can appear multiple times during this procedure. In the language of tensor networks this translate to duplication of nodes. Namely, there are multiple copies of the sub-tree representing $\mathbf{h}^{(l,i)}$ in the tensor network — one copy per appearance when unraveling the recursion. Figure 6 displays examples for tensor network diagrams of $\mathcal{T}(\mathbf{X})$ and $\mathcal{T}^{(t)}(\mathbf{X})$.

We note that, due to the node duplication mentioned above, the explicit definitions of $\mathcal{T}(\mathbf{X})$ and $\mathcal{T}^{(t)}(\mathbf{X})$ entail cumbersome notation. Nevertheless, we provide them in Appendix E.3.1 for the interested reader.

### E.3.1 Explicit Tensor Network Definitions

The tree tensor network representing $f^{(\theta,\mathcal{G})}(\mathbf{X})$ consists of an initial input level — the leaves of the tree — comprising $\rho_L(\{i\}, \mathcal{V})$ copies of $\mathbf{x}^{(i)}$ for each $i \in \mathcal{V}$. We will use $\mathbf{x}^{(i,\gamma)}$ to denote the copies of $\mathbf{x}^{(i)}$ for $i \in \mathcal{V}$ and $\gamma \in [\rho_L(\{i\}, \mathcal{V})]$. In accordance with the GNN inducing $f^{(\theta,\mathcal{G})}$, following the initial input level are $L + 1$ *layers*. Each layer $l \in [L]$ includes two levels: one comprising $\rho_{L-l+1}(\mathcal{V}, \mathcal{V})$ nodes standing for copies of $\mathbf{W}^{(l)}$, and another containing $\delta$-tensors — $\rho_{L-l}(\{i\}, \mathcal{V})$ copies of $\delta^{(|\mathcal{N}(i)|+1)}$ per $i \in \mathcal{V}$. We associate each node in these layers with its layer index and a vertex of the input graph $i \in \mathcal{V}$. Specifically, we will use $\mathbf{W}^{(l,i,\gamma)}$ to denote copies of $\mathbf{W}^{(l)}$ and $\delta^{(l,i,\gamma)}$ to denote copies of $\delta^{(|\mathcal{N}(i)|+1)}$, for $l \in [L], i \in \mathcal{V}$, and $\gamma \in \mathbb{N}$. In terms of connectivity, every leaf $\mathbf{x}^{(i,\gamma)}$ has a leg to $\mathbf{W}^{(1,i,\gamma)}$. The rest of the connections between nodes are such that each sub-tree whose root is $\delta^{(l,i,\gamma)}$ represents $\mathbf{h}^{(l,i)}$, *i.e.* contracting the sub-tree results in the hidden

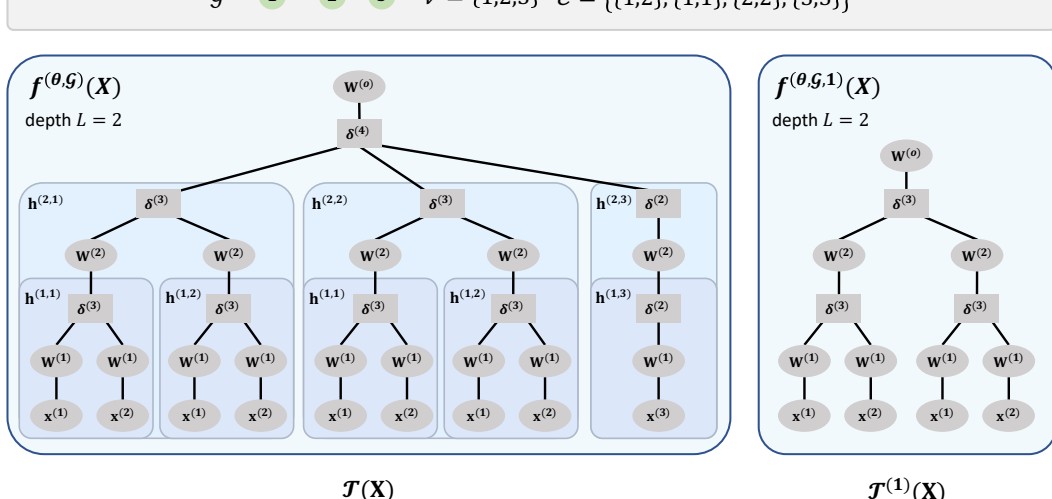

Figure 6: Tensor network diagrams of $\mathcal{T}(\mathbf{X})$ (left) and $\mathcal{T}^{(t)}(\mathbf{X})$ (right) representing $f^{(\theta,\mathcal{G})}(\mathbf{X})$ and $f^{(\theta,\mathcal{G},t)}(\mathbf{X})$, respectively, for $t = 1 \in \mathcal{V}$, vertex features $\mathbf{X} = (\mathbf{x}^{(1)}, \ldots, \mathbf{x}^{(|\mathcal{V}|)})$, and depth $L = 2$ GNNs with product aggregation (Section 3). The underlying input graph $\mathcal{G}$, over which the GNNs operate, is depicted at the top. We draw nodes associated with $\delta$-tensors as rectangles to signify their special (hyper-diagonal) structure, and omit leg weights to avoid clutter (legs connected to $\mathbf{x}^{(1)}, \mathbf{x}^{(2)}, \mathbf{x}^{(3)}$ have weight $D_x$ while all other legs have weight $D_h$). See Appendix E.3 for further details on the construction of $\mathcal{T}(\mathbf{X})$ and $\mathcal{T}^{(t)}(\mathbf{X})$, and Appendix E.3.1 for explicit formulations.

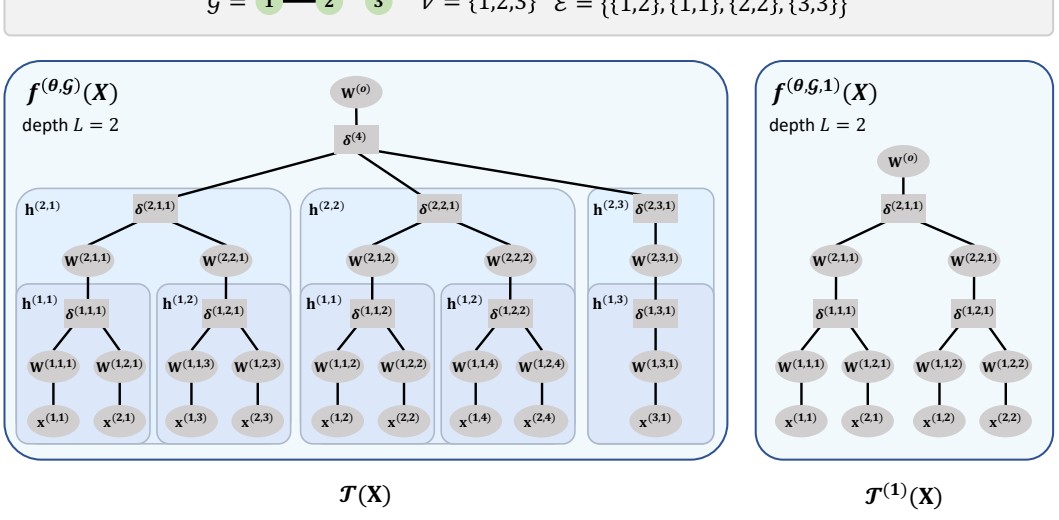

Figure 7: Tensor network diagrams (with explicit node duplication notation) of $\mathcal{T}(\mathbf{X})$ (left) and $\mathcal{T}^{(t)}(\mathbf{X})$ (right) representing $f^{(\theta,\mathcal{G})}(\mathbf{X})$ and $f^{(\theta,\mathcal{G},t)}(\mathbf{X})$, respectively, for $t = 1 \in \mathcal{V}$, vertex features $\mathbf{X} = (\mathbf{x}^{(1)}, \ldots, \mathbf{x}^{(|\mathcal{V}|)})$, and depth $L = 2$ GNNs with product aggregation (Section 3). This figure is identical to Figure 6, except that it uses the explicit notation for node duplication detailed in Appendix E.3.1. Specifically, each feature vector, weight matrix, and $\delta$-tensor is attached with an index specifying which copy it is (rightmost index in the superscript). Additionally, weight matrices and $\delta$-tensors are associated with a layer index and vertex in $\mathcal{V}$ (except for the output layer $\delta$-tensor in $\mathcal{T}(\mathbf{X})$ and $\mathbf{W}^{(o)}$). See Equations (15) to (20) for the explicit definitions of these tensor networks.

embedding for $i \in \mathcal{V}$ at layer $l \in [L]$ of the GNN inducing $f^{(\theta,\mathcal{G})}$. Last, is an output layer consisting of two connected nodes: a $\boldsymbol{\delta}^{(|\mathcal{V}|+1)}$ node, which has a leg to every $\delta$-tensor from layer $L$, and a $\mathbf{W}^{(o)}$ node. See Figure 7 (left) for an example of a tensor network diagram representing $f^{(\theta,\mathcal{G})}(\mathbf{X})$ with this notation.

The tensor network construction for $f^{(\theta,\mathcal{G},t)}(\mathbf{X})$ is analogous to that for $f^{(\theta,\mathcal{G})}(\mathbf{X})$, comprising an initial input level followed by $L + 1$ layers. Its input level and first $L$ layers are structured the same, up to differences in the number of copies for each node. Specifically, the number of copies of $\mathbf{x}^{(i)}$ is $\rho_L(\{i\}, \{t\})$ instead of $\rho_L(\{i\}, \mathcal{V})$, the number of copies of $\mathbf{W}^{(l)}$ is $\rho_{L-l+1}(\mathcal{V}, \{t\})$ instead of $\rho_{L-l+1}(\mathcal{V}, \mathcal{V})$, and the number of copies of $\boldsymbol{\delta}^{(|\mathcal{N}(i)|+1)}$ is $\rho_{L-l}(\{i\}, \{t\})$ instead of $\rho_{L-l}(\{i\}, \mathcal{V})$, for $i \in \mathcal{V}$ and $l \in [L]$. The output layer consists only of a $\mathbf{W}^{(o)}$ node, which is connected to the $\delta$-tensor in layer $L$ corresponding to vertex $t$. See Figure 7 (right) for an example of a tensor network diagram representing $f^{(\theta,\mathcal{G},t)}(\mathbf{X})$ with this notation.

Formally, the tensor network producing $f^{(\theta,\mathcal{G})}(\mathbf{X})$, denoted $\mathcal{T}(\mathbf{X}) = (\mathcal{V}_{\mathcal{T}(\mathbf{X})}, \mathcal{E}_{\mathcal{T}(\mathbf{X})}, w_{\mathcal{T}(\mathbf{X})})$, is defined by:

$$
\begin{aligned}
\mathcal{V}_{\mathcal{T}(\mathbf{X})} := &\left\{ \mathbf{x}^{(i,\gamma)} : i \in \mathcal{V}, \gamma \in [\rho_L(\{i\}, \mathcal{V})] \right\} \cup \\
&\left\{ \mathbf{W}^{(l,i,\gamma)} : l \in [L], i \in \mathcal{V}, \gamma \in [\rho_{L-l+1}(\{i\}, \mathcal{V})] \right\} \cup \\
&\left\{ \boldsymbol{\delta}^{(l,i,\gamma)} : l \in [L], i \in \mathcal{V}, \gamma \in [\rho_{L-l}(\{i\}, \mathcal{V})] \right\} \cup \\
&\left\{ \boldsymbol{\delta}^{(|\mathcal{V}|+1)}, \mathbf{W}^{(o)} \right\},
\end{aligned}
\tag{15}
$$

$$
\begin{aligned}
\mathcal{E}_{\mathcal{T}(\mathbf{X})} := &\left\{ \{ (\mathbf{x}^{(i,\gamma)}, 1), (\mathbf{W}^{(1,i,\gamma)}, 2) \} : i \in \mathcal{V}, \gamma \in [\rho_L(\{i\}, \mathcal{V})] \right\} \cup \\
&\left\{ \{ (\boldsymbol{\delta}^{(l,i,\gamma)}, j), (\mathbf{W}^{(l,\mathcal{N}(i)_j,\phi_{l,i,j}(\gamma))}, 1) \} : l \in [L], i \in \mathcal{V}, j \in [|\mathcal{N}(i)|], \gamma \in [\rho_{L-l}(\{i\}, \mathcal{V})] \right\} \cup \\
&\left\{ \{ (\boldsymbol{\delta}^{(l,i,\gamma)}, |\mathcal{N}(i)| + 1), (\mathbf{W}^{(l+1,i,\gamma)}, 2) \} : l \in [L-1], i \in \mathcal{V}, \gamma \in [\rho_{L-l}(\{i\}, \mathcal{V})] \right\} \cup \\
&\left\{ \{ (\boldsymbol{\delta}^{(|\mathcal{V}|+1)}, i), (\boldsymbol{\delta}^{(L,i,1)}, |\mathcal{N}(i)| + 1) \} : i \in \mathcal{V} \right\} \cup \left\{ \{ (\boldsymbol{\delta}^{(|\mathcal{V}|+1)}, |\mathcal{V}| + 1), (\mathbf{W}^{(o)}, 2) \} \right\},
\end{aligned}
\tag{16}
$$

$$
w_{\mathcal{T}(\mathbf{X})}(e) := \begin{cases} D_x & \text{, if } (\mathbf{x}^{(i,\gamma)}, 1) \text{ is an endpoint of } e \in \mathcal{E}_{\mathcal{T}} \text{ for some } i \in \mathcal{V}, \gamma \in [\rho_L(\{i\}, \mathcal{V})] \\ D_h & \text{, otherwise} \end{cases},
\tag{17}
$$

where $\phi_{l,i,j}(\gamma) := \gamma + \sum_{k < i \text{ s.t. } k \in \mathcal{N}(j)} \rho_{L-l}(\{k\}, \mathcal{V})$, for $l \in [L], i \in \mathcal{V}$, and $\gamma \in [\rho_{L-l}(\{i\}, \mathcal{V})]$, is used to map a $\delta$-tensor copy corresponding to $i$ in layer $l$ to a $\mathbf{W}^{(l)}$ copy, and $\mathcal{N}(i)_j$, for $i \in \mathcal{V}$ and $j \in [|\mathcal{N}(i)|]$, denotes the $j$'th neighbor of $i$ according to an ascending order (recall vertices are represented by indices from $1$ to $|\mathcal{V}|$).

Similarly, the tensor network $\mathcal{T}^{(t)}(\mathbf{X}) = (\mathcal{V}_{\mathcal{T}^{(t)}(\mathbf{X})}, \mathcal{E}_{\mathcal{T}^{(t)}(\mathbf{X})}, w_{\mathcal{T}^{(t)}(\mathbf{X})})$, producing $f^{(\theta,\mathcal{G},t)}(\mathbf{X})$, is defined by:

$$
\begin{aligned}
\mathcal{V}_{\mathcal{T}^{(t)}(\mathbf{X})} := &\left\{ \mathbf{x}^{(i,\gamma)} : i \in \mathcal{V}, \gamma \in [\rho_L(\{i\}, \{t\})] \right\} \cup \\
&\left\{ \mathbf{W}^{(l,i,\gamma)} : l \in [L], i \in \mathcal{V}, \gamma \in [\rho_{L-l+1}(\{i\}, \{t\})] \right\} \cup \\
&\left\{ \boldsymbol{\delta}^{(l,i,\gamma)} : l \in [L], i \in \mathcal{V}, \gamma \in [\rho_{L-l}(\{i\}, \{t\})] \right\} \cup \\
&\left\{ \mathbf{W}^{(o)} \right\},
\end{aligned}
\tag{18}
$$

$$\mathcal{E}_{\mathcal{T}^{(t)}(\mathbf{X})} := \left\{ \left\{ (\mathbf{x}^{(i,\gamma)}, 1), (\mathbf{W}^{(1,i,\gamma)}, 2) \right\} : i \in \mathcal{V}, \gamma \in [\rho_L(\{i\}, \{t\})] \right\} \cup$$

$$\left\{ \left\{ (\boldsymbol{\delta}^{(l,i,\gamma)}, j), (\mathbf{W}^{(l,\mathcal{N}(i)_j, \phi_{l,i,j}^{(t)}(\gamma))}, 1) \right\} : l \in [L], i \in \mathcal{V}, j \in [|\mathcal{N}(i)|], \gamma \in [\rho_{L-l}(\{i\}, \{t\})] \right\} \cup$$

$$\left\{ \left\{ (\boldsymbol{\delta}^{(l,i,\gamma)}, |\mathcal{N}(i)| + 1), (\mathbf{W}^{(l+1,i,\gamma)}, 2) \right\} : l \in [L-1], i \in \mathcal{V}, \gamma \in [\rho_{L-l}(\{i\}, \{t\})] \right\} \cup$$

$$\left\{ \left\{ (\boldsymbol{\delta}^{(L,t,1)}, |\mathcal{N}(t)| + 1), (\mathbf{W}^{(o)}, 2) \right\} \right\},$$

$$(19)$$

$$w_{\mathcal{T}^{(t)}(\mathbf{X})}(e) := \begin{cases} D_x & , \text{ if } (\mathbf{x}^{(i,\gamma)}, 1) \text{ is an endpoint of } e \in \mathcal{E}_{\mathcal{T}} \text{ for some } i \in \mathcal{V}, \gamma \in [\rho_L(\{i\}, \{t\})] \\ D_h & , \text{ otherwise} \end{cases},$$

$$(20)$$

where $\phi_{l,i,j}^{(t)}(\gamma) := \gamma + \sum_{k < i \text{ s.t. } k \in \mathcal{N}(j)} \rho_{L-l}(\{k\}, \{t\})$, for $l \in [L], i \in \mathcal{V}$, and $\gamma \in [\rho_{L-l}(\{i\}, \{t\})]$, is used to map a $\delta$-tensor copy corresponding to $i$ in layer $l$ to a $\mathbf{W}^{(l)}$ copy.

Proposition 1 verifies that contracting $\mathcal{T}(\mathbf{X})$ and $\mathcal{T}^{(t)}(\mathbf{X})$ yields $f^{(\theta,\mathcal{G})}(\mathbf{X})$ and $f^{(\theta,\mathcal{G},t)}(\mathbf{X})$, respectively.

**Proposition 1.** *For an undirected graph $\mathcal{G}$ and $t \in \mathcal{V}$, let $f^{(\theta,\mathcal{G})}$ and $f^{(\theta,\mathcal{G},t)}$ be the functions realized by depth $L$ graph and vertex prediction GNNs, respectively, with width $D_h$, learnable weights $\theta$, and product aggregation (Equations (2) to (5)). For vertex features $\mathbf{X} = (\mathbf{x}^{(1)}, \ldots, \mathbf{x}^{(|\mathcal{V}|)}) \in \mathbb{R}^{D_x \times |\mathcal{V}|}$, let the tensor networks $\mathcal{T}(\mathbf{X}) = (\mathcal{V}_{\mathcal{T}(\mathbf{X})}, \mathcal{E}_{\mathcal{T}(\mathbf{X})}, w_{\mathcal{T}(\mathbf{X})})$ and $\mathcal{T}^{(t)}(\mathbf{X}) = (\mathcal{V}_{\mathcal{T}^{(t)}(\mathbf{X})}, \mathcal{E}_{\mathcal{T}^{(t)}(\mathbf{X})}, w_{\mathcal{T}^{(t)}(\mathbf{X})})$ be as defined in Equations (15) to (20), respectively. Then, performing the contractions described by $\mathcal{T}(\mathbf{X})$ produces $f^{(\theta,\mathcal{G})}(\mathbf{X})$, and performing the contractions described by $\mathcal{T}^{(t)}(\mathbf{X})$ produces $f^{(\theta,\mathcal{G},t)}(\mathbf{X})$.*

*Proof sketch (proof in Appendix I.6).* For both $\mathcal{T}(\mathbf{X})$ and $\mathcal{T}^{(t)}(\mathbf{X})$, a straightforward induction over the layer $l \in [L]$ establishes that contracting the sub-tree whose root is $\boldsymbol{\delta}^{(l,i,\gamma)}$ results in $\mathbf{h}^{(l,i)}$ for all $i \in \mathcal{V}$ and $\gamma$, where $\mathbf{h}^{(l,i)}$ is the hidden embedding for $i$ at layer $l$ of the GNNs inducing $f^{(\theta,\mathcal{G})}$ and $f^{(\theta,\mathcal{G},t)}$, given vertex features $\mathbf{x}^{(1)}, \ldots, \mathbf{x}^{(|\mathcal{V}|)}$. The proof concludes by showing that the contractions in the output layer of $\mathcal{T}(\mathbf{X})$ and $\mathcal{T}^{(t)}(\mathbf{X})$ reproduce the operations defining $f^{(\theta,\mathcal{G})}(\mathbf{X})$ and $f^{(\theta,\mathcal{G},t)}(\mathbf{X})$ in Equations (3) and (4), respectively. $\qquad\square$

## F  General Walk Index Sparsification

Our edge sparsification algorithm — Walk Index Sparsification (WIS) — was obtained as an instance of the General Walk Index Sparsification (GWIS) scheme described in Section 5. Algorithm 3 formally outlines this general scheme.

## G  Efficient Implementation of 1-Walk Index Sparsification

Algorithm 2 (Section 5) provides an efficient implementation for 1-WIS, *i.e.* Algorithm 1 with $L = 2$. In this appendix, we formalize the equivalence between the two algorithms, meaning, we establish that Algorithm 2 indeed implements 1-WIS.

Examining some iteration $n \in [N]$ of 1-WIS, let $\mathbf{s} \in \mathbb{R}^{|\mathcal{V}|}$ be the tuple defined by $s_t = \text{WI}_{1,t}(\{t\}) = \rho_1(\mathcal{C}_{\{t\}}, \{t\})$ for $t \in \mathcal{V}$. Recall that $\mathcal{C}_{\{t\}}$ is the set of vertices with an edge crossing the partition induced by $\{t\}$. Thus, if $t$ is not isolated, then $\mathcal{C}_{\{t\}} = \mathcal{N}(t)$ and $s_t = \text{WI}_{1,t}(\{t\}) = |\mathcal{N}(t)|$. Otherwise, if $t$ is isolated, then $\mathcal{C}_{\{t\}} = \emptyset$ and $s_t = \text{WI}_{1,t}(\{t\}) = 0$. 1-WIS computes for each $e \in \mathcal{E}$ (excluding self-loops) a tuple $\mathbf{s}^{(e)} \in \mathbb{R}^{|\mathcal{V}|}$ holding in its $t$'th entry what the value of $\text{WI}_{1,t}(\{t\})$ would be if $e$ is to be removed, for all $t \in \mathcal{V}$. Notice that $\mathbf{s}^{(e)}$ and $\mathbf{s}$ agree on all entries except for $i, j \in e$, since removing $e$ from the graph only affects the degrees of $i$ and $j$. Specifically, for $i \in e$,

**Algorithm 3** $(L-1)$-General Walk Index Sparsification (GWIS)

**Input:**
- $\mathcal{G}$ — graph
- $L \in \mathbb{N}$ — GNN depth
- $N \in \mathbb{N}$ — number of edges to remove
- $\mathcal{I}_1, \ldots, \mathcal{I}_M \subseteq \mathcal{V}$ — vertex subsets specifying walk indices to maintain for graph prediction
- $\mathcal{J}_1, \ldots, \mathcal{J}_{M'} \subseteq \mathcal{V}$ and $t_1, \ldots, t_{M'} \in \mathcal{V}$ — vertex subsets specifying walk indices to maintain with respect to target vertices, for vertex prediction
- ARGMAX — operator over tuples $(\mathbf{s}^{(e)} \in \mathbb{R}^{M+M'})_{e \in \mathcal{E}}$ that returns the edge whose tuple is maximal according to some order

**Result:** Sparsified graph obtained by removing $N$ edges from $\mathcal{G}$

---

**for** $n = 1, \ldots, N$ **do**
    # for every edge, compute walk indices of partitions after the edge's removal
    **for** $e \in \mathcal{E}$ (excluding self-loops) **do**
      initialize $\mathbf{s}^{(e)} = (0, \ldots, 0) \in \mathbb{R}^{M+M'}$
      remove $e$ from $\mathcal{G}$ (temporarily)
      for every $m \in [M]$, set $\mathbf{s}_m^{(e)} = \mathrm{WI}_{L-1}(\mathcal{I}_m)$  # $= \rho_{L-1}(\mathcal{C}_{\mathcal{I}_m}, \mathcal{V})$
      for every $m \in [M']$, set $\mathbf{s}_{M+m}^{(e)} = \mathrm{WI}_{L-1,t_m}(\mathcal{J}_m)$  # $= \rho_{L-1}(\mathcal{C}_{\mathcal{J}_m}, \{t_m\})$
      add $e$ back to $\mathcal{G}$
    **end for**
    # prune edge whose removal harms walk indices the least according to the ARGMAX operator
    let $e' \in \mathrm{ARGMAX}_{e \in \mathcal{E}} \mathbf{s}^{(e)}$
    **remove** $e'$ from $\mathcal{G}$ (permanently)
**end for**

---

either $\mathbf{s}_i^{(e)} = \mathbf{s}_i - 1 = |\mathcal{N}(i)| - 1$ if the removal of $e$ did not isolate $i$, or $\mathbf{s}_i^{(e)} = \mathbf{s}_i - 2 = 0$ if it did (due to self-loops, if a vertex has a single edge to another then $|\mathcal{N}(i)| = 2$, so removing that edge changes $\mathrm{WI}_{1,i}(\{i\})$ from two to zero). As a result, for any $e = \{i,j\}, e' = \{i',j'\} \in \mathcal{E}$, after sorting the entries of $\mathbf{s}^{(e)}$ and $\mathbf{s}^{(e')}$ in ascending order we have that $\mathbf{s}^{(e')}$ is greater in lexicographic order than $\mathbf{s}^{(e)}$ if and only if the pair $(\min\{|\mathcal{N}(i')|, |\mathcal{N}(j')|\}, \max\{|\mathcal{N}(i')|, |\mathcal{N}(j')|\})$ is greater in lexicographic order than $(\min\{|\mathcal{N}(i)|, |\mathcal{N}(j)|\}, \max\{|\mathcal{N}(i)|, |\mathcal{N}(j)|\})$. Therefore, at every iteration $n \in [N]$ Algorithm 2 and 1-WIS (Algorithm 1 with $L = 2$) remove the same edge.

# H  Further Experiments and Implementation Details

## H.1  Further Experiments

Figure 8 supplements Figure 3 from Section 5.2 by including experiments with additional: *(i)* GNN architectures — GIN and ResGCN; and *(ii)* datasets — Chameleon, Squirrel, and Amazon Computers. Overall, our evaluation includes six standard vertex prediction datasets in which we observed the graph structure to be crucial for accurate prediction, as measured by the difference between the test accuracy of a GCN trained and evaluated over the original graph and its test accuracy when trained and evaluated over the graph after all of the graph's edges were removed. We also considered, but excluded, the following datasets in which the accuracy difference was insignificant (less than five percentage points): Citeseer [89], PubMed [76], Coauthor CS and Physics [92], and Amazon Photo [92].

## H.2  Further Implementation Details

We provide implementation details omitted from our experimental reports (Section 4.2, Section 5, and Appendix H.1). Source code for reproducing our results and figures, based on the PyTorch [81] and PyTorch Geometric [38] frameworks, can be found at https://github.com/noamrazin/

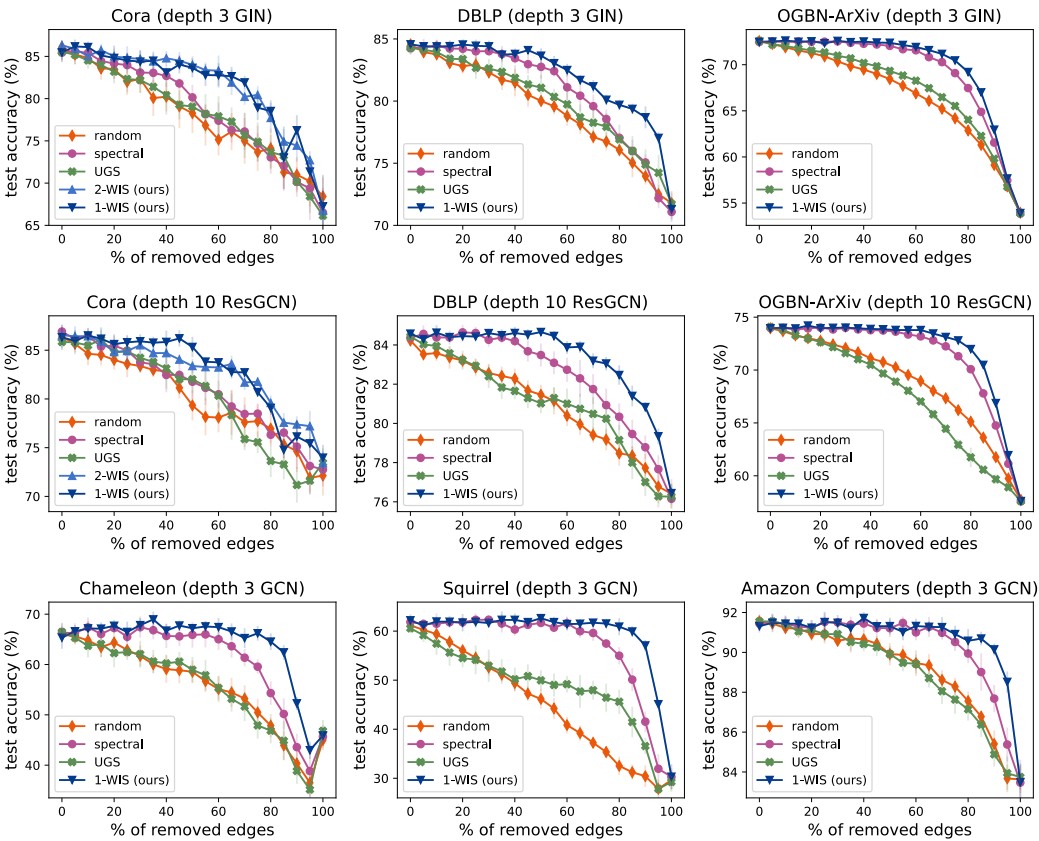

Figure 8: Comparison of GNN accuracies following sparsification of input edges — WIS, the edge sparsification algorithm brought forth by our theory (Algorithm 1), markedly outperforms alternative methods. This figure supplements Figure 3 from Section 5.2 by including experiments with: *(i)* a depth $L = 3$ GIN over the Cora, DBLP, and OGBN-ArXiv datasets; *(ii)* a depth $L = 10$ ResGCN over the Cora, DBLP, and OGBN-ArXiv datasets; and *(iii)* a depth $L = 3$ GCN over the Chameleon, Squirrel, and Amazon Computers datasets. Markers and error bars report means and standard deviations, respectively, taken over ten runs per configuration for GCN and GIN, and over five runs per configuration for ResGCN (we use fewer runs due to the larger size of ResGCN). For further details see caption of Figure 3 as well as Appendix H.2.

`gnn_interactions`. All experiments were run either on a single Nvidia RTX 2080 Ti GPU or a single Nvidia RTX A6000 GPU.

### H.2.1 Empirical Demonstration of Theoretical Analysis (Table 1)

**Models** All models used, *i.e.* GCN, GAT, and GIN, had three layers of width 16 with ReLU non-linearity. To ease optimization, we added layer normalization [4] after each one. Mean aggregation and a linear output layer were applied over the last hidden embeddings for prediction. As in the synthetic experiments of [1], each GAT layer consisted of four attention heads. Each GIN layer had its $\epsilon$ parameter fixed to zero and contained a two-layer feed-forward network, whose layers comprised a linear layer, batch normalization [54], and ReLU non-linearity.

**Data** The datasets consisted of 10000 train and 2000 test graphs. For every graph, we drew uniformly at random a label from $\{0, 1\}$ and an image from Fashion-MNIST. Then, depending on the chosen label, another image was sampled either from the same class (for label 1) or from all other classes (for label 0). We extracted patches of pixels from each image by flattening it into a vector and splitting the vector to 16 equally sized segments.

**Optimization** The binary cross-entropy loss was minimized via the Adam optimizer [58] with default $\beta_1, \beta_2$ coefficients and full-batches (*i.e.* every batch contained the whole training set). Optimiza-

tion proceeded until the train accuracy did not improve by at least 0.01 over 1000 consecutive epochs or 10000 epochs elapsed. The learning rates used for GCN, GAT, and GIN were $5 \cdot 10^{-3}, 5 \cdot 10^{-3}$, and $10^{-2}$, respectively.

**Hyperparameter tuning**  For each model separately, to tune the learning rate we carried out five runs (differing in random seed) with every value in the range $\{10^{-1}, 5 \cdot 10^{-2}, 10^{-2}, 5 \cdot 10^{-3}, 10^{-3}\}$ over the dataset whose essential partition has low walk index. Since our interest resides in expressivity, which manifests in ability to fit the training set, for every model we chose the learning rate that led to the highest mean train accuracy.

### H.2.2   Edge Sparsification (Figures 3 and 8)

**Adaptations to UGS [24]**  [24] proposed UGS as a framework for jointly pruning input graph edges and weights of a GNN. At a high-level, UGS trains two differentiable masks, $\boldsymbol{m}_g$ and $\boldsymbol{m}_\theta$, that are multiplied with the graph adjacency matrix and the GNN's weights, respectively. Then, after a certain number of optimization steps, a predefined percentage $p_g$ of graph edges are removed according to the magnitudes of entries in $\boldsymbol{m}_g$, and similarly, $p_\theta$ percent of the GNN's weights are fixed to zero according to the magnitudes of entries in $\boldsymbol{m}_\theta$. This procedure continues in iterations, where each time the remaining GNN weights are rewinded to their initial values, until the desired sparsity levels are attained — see Algorithms 1 and 2 in [24]. To facilitate a fair comparison of our $(L-1)$-WIS edge sparsification algorithm with UGS, we make the following adaptations to UGS.

- We adapt UGS to only remove edges, which is equivalent to fixing the entries in the weight mask $\boldsymbol{m}_\theta$ to one and setting $p_\theta = 0$ in Algorithm 1 of [24].

- For comparing performance across a wider range of sparsity levels, the number of edges removed at each iteration is changed from 5% of the current number of edges to 5% of the original number of edges.

- Since our evaluation focuses on undirected graphs, we enforce the adjacency matrix mask $\boldsymbol{m}_g$ to be symmetric.

**Spectral sparsification [93]**  For Cora and DBLP, we used a Python implementation of the spectral sparsification algorithm from [93], based on the PyGSP library implementation.[14] To enable more efficient experimentation over the larger scale OGBN-ArXiv dataset, we used a Julia implementation based on that from the Laplacians library.[15]

**Models**  The GCN and GIN models had three layers of width 64 with ReLU non-linearity. As in the experiments of Section 4.2, we added layer normalization [4] after each one. Every GIN layer had a trainable $\epsilon$ parameter and contained a two-layer feed-forward network, whose layers comprised a linear layer, batch normalization [54], and ReLU non-linearity. For ResGCN, we used the implementation from [24] with ten layers of width 64. In all models, a linear output layer was applied over the last hidden embeddings for prediction.

**Data**  All datasets in our evaluation are multi-class vertex prediction tasks, each consisting of a single graph. In Cora, DBLP, and OGBN-ArXiv, vertices represent scientific publications and edges stand for citation links. In Chameleon and Squirrel, vertices represent web pages on Wikipedia and edges stand for mutual links between pages. In Amazon Computers, vertices represent products and edges indicate that two products are frequently bought together. For simplicity, we treat all graphs as undirected. Table 2 reports the number of vertices and undirected edges in each dataset. For all datasets, except OGBN-ArXiv, we randomly split the labels of vertices into train, validation, and test sets comprising $80\%, 10\%$, and $10\%$ of all labels, respectively. For OGBN-ArXiv, we used the default split from [52].

**Optimization**  The cross-entropy loss was minimized via the Adam optimizer [58] with default $\beta_1, \beta_2$ coefficients and full-batches (*i.e.* every batch contained the whole training set). Optimization proceeded until the validation accuracy did not improve by at least 0.01 over 1000 consecutive epochs or 10000 epochs elapsed. The test accuracies reported in Figure 3 are those achieved during the epochs with highest validation accuracies. Table 3 specifies additional optimization hyperparameters.

---

[14]See https://github.com/epfl-lts2/pygsp/.

[15]See https://github.com/danspielman/Laplacians.jl.

Table 2: Graph size of each dataset used for comparing edge sparsification algorithms in Figures 3 and 8.

|  | # of Vertices | # of Undirected Edges |
|---|---|---|
| Cora | 2,708 | 5,278 |
| DBLP | 17,716 | 52,867 |
| OGBN-ArXiv | 169,343 | 1,157,799 |
| Chameleon | 2,277 | 31,396 |
| Squirrel | 5,201 | 198,423 |
| Amazon Computers | 13,381 | 245,861 |

Table 3: Optimization hyperparameters used in the experiments of Figures 3 and 8 per model and dataset.

|  |  | Learning Rate | Weight Decay | Edge Mask $\ell_1$ Regularization of UGS |
|---|---|---|---|---|
| GCN | Cora | $5 \cdot 10^{-4}$ | $10^{-3}$ | $10^{-2}$ |
|  | DBLP | $10^{-3}$ | $10^{-4}$ | $10^{-2}$ |
|  | OGBN-ArXiv | $10^{-3}$ | $0$ | $10^{-2}$ |
|  | Chameleon | $10^{-3}$ | $10^{-4}$ | $10^{-2}$ |
|  | Squirrel | $5 \cdot 10^{-4}$ | $0$ | $10^{-4}$ |
|  | Amazon Computers | $10^{-3}$ | $10^{-4}$ | $10^{-2}$ |
| GIN | Cora | $10^{-3}$ | $10^{-3}$ | $10^{-2}$ |
|  | DBLP | $10^{-3}$ | $10^{-3}$ | $10^{-2}$ |
|  | OGBN-ArXiv | $10^{-4}$ | $0$ | $10^{-2}$ |
| ResGCN | Cora | $5 \cdot 10^{-4}$ | $10^{-3}$ | $10^{-4}$ |
|  | DBLP | $5 \cdot 10^{-4}$ | $10^{-4}$ | $10^{-4}$ |
|  | OGBN-ArXiv | $10^{-3}$ | $0$ | $10^{-2}$ |

**Hyperparameter tuning** For each combination of model and dataset separately, we tuned the learning rate, weight decay coefficient, and edge mask $\ell_1$ regularization coefficient for UGS, and applied the chosen values for evaluating all methods without further tuning (note that the edge mask $\ell_1$ regularization coefficient is relevant only for UGS). In particular, we carried out a grid search over learning rates $\{10^{-3}, 5 \cdot 10^{-4}, 10^{-4}\}$, weight decay coefficients $\{10^{-3}, 10^{-4}, 0\}$, and edge mask $\ell_1$ regularization coefficients $\{10^{-2}, 10^{-3}, 10^{-4}\}$. Per hyperparameter configuration, we ran ten repetitions of UGS (differing in random seed), each until all of the input graph's edges were removed. At every edge sparsity level $(0\%, 5\%, 10\%, \ldots, 100\%)$, in accordance with [24], we trained a new model with identical hyperparameters, but a fixed edge mask, over each of the ten graphs. We chose the hyperparameters that led to the highest mean validation accuracy, taken over the sparsity levels and ten runs.

Due to the size of the ResGCN model, tuning its hyperparameters entails significant computational costs. Thus, over the Cora and DBLP datasets, per hyperparameter configuration we ran five repetitions of UGS with ResGCN instead of ten. For the large-scale OGBN-ArXiv dataset, we adopted the same hyperparameters used for GCN.

**Other** To allow more efficient experimentation, we compute the edge removal order of 2-WIS (Algorithm 1) in batches of size 100. Specifically, at each iteration of 2-WIS, instead of removing the edge $e'$ with maximal walk index tuple $\mathbf{s}^{(e')}$, the 100 edges with largest walk index tuples are removed. For randomized edge sparsification algorithms — random pruning, the spectral sparsification method of [93], and the adaptation of UGS [24] — the evaluation runs for a given dataset and percentage of removed edges were carried over sparsified graphs obtained using different random seeds.

# I Deferred Proofs

## I.1 Additional Notation

For vectors, matrices, or tensors, parenthesized superscripts denote elements in a collection, e.g. $(\mathbf{a}^{(i)} \in \mathbb{R}^D)_{n=1}^N$, while subscripts refer to entries, e.g. $\mathbf{A}_{d_1, d_2} \in \mathbb{R}$ is the $(d_1, d_2)$'th entry of $\mathbf{A} \in \mathbb{R}^{D_1 \times D_2}$. A colon is used to indicate a range of entries, e.g. $\mathbf{a}_{:d}$ is the first $d$ entries of

$\mathbf{a} \in \mathbb{R}^D$. We use $*$ to denote tensor contractions (Definition 7), $\circ$ to denote the Kronecker product, and $\odot$ to denote the Hadamard product. For $P \in \mathbb{N}_{\geq 0}$, the $P$'th Hadamard power operator is denoted by $\odot^P$, *i.e.* $[\odot^P \mathbf{A}]_{d_1,d_2} = \mathbf{A}_{d_1,d_2}^P$ for $\mathbf{A} \in \mathbb{R}^{D_1 \times D_2}$. Lastly, when enumerating over sets of indices an ascending order is assumed.

## I.2 Proof of Theorem 2

We assume familiarity with the basic concepts from tensor analysis introduced in Appendix E.1, and rely on the tensor network representations established for GNNs with product aggregation in Appendix E. Specifically, we use the fact that for any $\mathbf{X} = (\mathbf{x}^{(1)}, \ldots, \mathbf{x}^{(|\mathcal{V}|)}) \in \mathbb{R}^{D_x \times |\mathcal{V}|}$ there exist tree tensor networks $\mathcal{T}(\mathbf{X})$ and $\mathcal{T}^{(t)}(\mathbf{X})$ (described in Appendix E.3 and formally defined in Equations (15) to (20)) such that: *(i)* their contraction yields $f^{(\theta,\mathcal{G})}(\mathbf{X})$ and $f^{(\theta,\mathcal{G},t)}(\mathbf{X})$, respectively (Proposition 1); and *(ii)* each of their leaves is associated with a vertex feature vector, *i.e.* one of $\mathbf{x}^{(1)}, \ldots, \mathbf{x}^{(|\mathcal{V}|)}$, whereas all other aspects of the tensor networks do not depend on $\mathbf{x}^{(1)}, \ldots, \mathbf{x}^{(|\mathcal{V}|)}$.

The proof proceeds as follows. In Appendix I.2.1, by importing machinery from tensor analysis literature (in particular, adapting Claim 7 from [62]), we show that the separation ranks of $f^{(\theta,\mathcal{G})}$ and $f^{(\theta,\mathcal{G},t)}$ can be upper bounded via cuts in their corresponding tensor networks. Namely, $\mathrm{sep}(f^{(\theta,\mathcal{G})}; \mathcal{I})$ is at most the minimal multiplicative cut weight in $\mathcal{T}(\mathbf{X})$, among cuts separating leaves associated with vertices of the input graph in $\mathcal{I}$ from leaves associated with vertices of the input graph in $\mathcal{I}^c$, where multiplicative cut weight refers to the product of weights belonging to legs crossing the cut. Similarly, $\mathrm{sep}(f^{(\theta,\mathcal{G},t)}; \mathcal{I})$ is at most the minimal multiplicative cut weight in $\mathcal{T}^{(t)}(\mathbf{X})$, among cuts of the same form. We conclude in Appendices I.2.2 and I.2.3 by applying this technique for upper bounding $\mathrm{sep}(f^{(\theta,\mathcal{G})}; \mathcal{I})$ and $\mathrm{sep}(f^{(\theta,\mathcal{G},t)}; \mathcal{I})$, respectively, *i.e.* by finding cuts in the respective tensor networks with sufficiently low multiplicative weights.

### I.2.1 Upper Bounding Separation Rank via Multiplicative Cut Weight in Tensor Network

In a tensor network $\mathcal{T} = (\mathcal{V}_\mathcal{T}, \mathcal{E}_\mathcal{T}, w_\mathcal{T})$, every $\mathcal{J}_\mathcal{T} \subseteq \mathcal{V}_\mathcal{T}$ induces a *cut* $(\mathcal{J}_\mathcal{T}, \mathcal{J}_\mathcal{T}^c)$, *i.e.* a partition of the nodes into two sets. We denote by $\mathcal{E}_\mathcal{T}(\mathcal{J}_\mathcal{T}) := \{\{u, v\} \in \mathcal{E}_\mathcal{T} : u \in \mathcal{J}_\mathcal{T}, v \in \mathcal{J}_\mathcal{T}^c\}$ the set of legs crossing the cut, and define the *multiplicative cut weight* of $\mathcal{J}_\mathcal{T}$ to be the product of weights belonging to legs in $\mathcal{E}_\mathcal{T}(\mathcal{J}_\mathcal{T})$, *i.e.*:

$$w_\mathcal{T}^\Pi(\mathcal{J}_\mathcal{T}) := \prod\nolimits_{e \in \mathcal{E}_\mathcal{T}(\mathcal{J}_\mathcal{T})} w_\mathcal{T}(e).$$

For $\mathbf{X} = (\mathbf{x}^{(1)}, \ldots, \mathbf{x}^{(|\mathcal{V}|)}) \in \mathbb{R}^{D_x \times |\mathcal{V}|}$, let $\mathcal{T}(\mathbf{X})$ and $\mathcal{T}^{(t)}(\mathbf{X})$ be the tensor networks corresponding to $f^{(\theta,\mathcal{G})}(\mathbf{X})$ and $f^{(\theta,\mathcal{G},t)}(\mathbf{X})$ (detailed in Appendix E.3), respectively. Both $\mathcal{T}(\mathbf{X})$ and $\mathcal{T}^{(t)}(\mathbf{X})$ adhere to a tree structure. Each leaf node is associated with a vertex feature vector (*i.e.* one of $\mathbf{x}^{(1)}, \ldots, \mathbf{x}^{(|\mathcal{V}|)}$), while interior nodes are associated with weight matrices or $\delta$-tensors. The latter are tensors with modes of equal dimension holding ones on their hyper-diagonal and zeros elsewhere. The restrictions imposed by $\delta$-tensors induce a modified notion of multiplicative cut weight, where legs incident to the same $\delta$-tensor only contribute once to the weight product (note that weights of legs connected to the same $\delta$-tensor are equal since they stand for mode dimensions).

**Definition 8.** For a tensor network $\mathcal{T} = (\mathcal{V}_\mathcal{T}, \mathcal{E}_\mathcal{T}, w_\mathcal{T})$ and subset of nodes $\mathcal{J}_\mathcal{T} \subseteq \mathcal{V}_\mathcal{T}$, let $\mathcal{E}_\mathcal{T}(\mathcal{J}_\mathcal{T})$ be the set of edges crossing the cut $(\mathcal{J}_\mathcal{T}, \mathcal{J}_\mathcal{T}^c)$. Denote by $\widetilde{\mathcal{E}}_\mathcal{T}(\mathcal{J}_\mathcal{T}) \subseteq \mathcal{E}_\mathcal{T}(\mathcal{J}_\mathcal{T})$ a subset of legs containing for each $\delta$-tensor in $\mathcal{V}_\mathcal{T}$ only a single leg from $\mathcal{E}_\mathcal{T}(\mathcal{J}_\mathcal{T})$ incident to it, along with all legs in $\mathcal{E}_\mathcal{T}(\mathcal{J}_\mathcal{T})$ not connected to $\delta$-tensors. Then, the *modified multiplicative cut weight* of $\mathcal{J}_\mathcal{T}$ is:

$$\widetilde{w}_\mathcal{T}^\Pi(\mathcal{J}_\mathcal{T}) := \prod\nolimits_{e \in \widetilde{\mathcal{E}}_\mathcal{T}(\mathcal{J}_\mathcal{T})} w_\mathcal{T}(e).$$

Lemma 1 establishes that $\mathrm{sep}(f^{(\theta,\mathcal{G})}; \mathcal{I})$ and $\mathrm{sep}(f^{(\theta,\mathcal{G},t)}; \mathcal{I})$ are upper bounded by the minimal modified multiplicative cut weights in $\mathcal{T}(\mathbf{X})$ and $\mathcal{T}^{(t)}(\mathbf{X})$, respectively, among cuts separating leaves associated with vertices in $\mathcal{I}$ from leaves associated vertices in $\mathcal{I}^c$.

**Lemma 1.** *For any* $\mathbf{X} = (\mathbf{x}^{(1)}, \ldots, \mathbf{x}^{(|\mathcal{V}|)}) \in \mathbb{R}^{D_x \times |\mathcal{V}|}$, *let* $\mathcal{T}(\mathbf{X}) = (\mathcal{V}_{\mathcal{T}(\mathbf{X})}, \mathcal{E}_{\mathcal{T}(\mathbf{X})}, w_{\mathcal{T}(\mathbf{X})})$ *and* $\mathcal{T}^{(t)}(\mathbf{X}) = (\mathcal{V}_{\mathcal{T}^{(t)}(\mathbf{X})}, \mathcal{E}_{\mathcal{T}^{(t)}(\mathbf{X})}, w_{\mathcal{T}^{(t)}(\mathbf{X})})$ *be the tensor network representations of* $f^{(\theta,\mathcal{G})}(\mathbf{X})$ *and* $f^{(\theta,\mathcal{G},t)}(\mathbf{X})$ *(described in Appendix E.3 and formally defined in Equations (15) to (20)), respectively.*

Denote by $\mathcal{V}_{\mathcal{T}(\mathbf{X})}[\mathcal{I}] \subseteq \mathcal{V}_{\mathcal{T}(\mathbf{X})}$ and $\mathcal{V}_{\mathcal{T}^{(t)}(\mathbf{X})}[\mathcal{I}] \subseteq \mathcal{V}_{\mathcal{T}^{(t)}(\mathbf{X})}$ *the sets of leaf nodes in* $\mathcal{T}(\mathbf{X})$ *and* $\mathcal{T}^{(t)}(\mathbf{X})$, *respectively, associated with vertices in* $\mathcal{I}$ *from the input graph* $\mathcal{G}$. *Formally:*

$$\mathcal{V}_{\mathcal{T}(\mathbf{X})}[\mathcal{I}] := \left\{ \mathbf{x}^{(i,\gamma)} \in \mathcal{V}_{\mathcal{T}(\mathbf{X})} : i \in \mathcal{I}, \gamma \in [\rho_L(\{i\}, \mathcal{V})] \right\},$$

$$\mathcal{V}_{\mathcal{T}^{(t)}(\mathbf{X})}[\mathcal{I}] := \left\{ \mathbf{x}^{(i,\gamma)} \in \mathcal{V}_{\mathcal{T}^{(t)}(\mathbf{X})} : i \in \mathcal{I}, \gamma \in [\rho_L(\{i\}, \{t\})] \right\}.$$

*Similarly, denote by* $\mathcal{V}_{\mathcal{T}(\mathbf{X})}[\mathcal{I}^c] \subseteq \mathcal{V}_{\mathcal{T}(\mathbf{X})}$ *and* $\mathcal{V}_{\mathcal{T}^{(t)}(\mathbf{X})}[\mathcal{I}^c] \subseteq \mathcal{V}_{\mathcal{T}^{(t)}(\mathbf{X})}$ *the sets of leaf nodes in* $\mathcal{T}(\mathbf{X})$ *and* $\mathcal{T}^{(t)}(\mathbf{X})$, *respectively, associated with vertices in* $\mathcal{I}^c$. *Then, the following hold:*

*(graph prediction)* $\quad \mathrm{sep}\big(f^{(\theta,\mathcal{G})}; \mathcal{I}\big) \leq \min_{\substack{\mathcal{J}_{\mathcal{T}(\mathbf{X})} \subseteq \mathcal{V}_{\mathcal{T}(\mathbf{X})} \\ \text{s.t. } \mathcal{V}_{\mathcal{T}(\mathbf{X})}[\mathcal{I}] \subseteq \mathcal{J}_{\mathcal{T}(\mathbf{X})} \text{ and } \mathcal{V}_{\mathcal{T}(\mathbf{X})}[\mathcal{I}^c] \subseteq \mathcal{J}^c_{\mathcal{T}(\mathbf{X})}}} \widetilde{w}^{\Pi}_{\mathcal{T}(\mathbf{X})}(\mathcal{J}_{\mathcal{T}(\mathbf{X})}),$ $\qquad (21)$

*(vertex prediction)* $\quad \mathrm{sep}\big(f^{(\theta,\mathcal{G},t)}; \mathcal{I}\big) \leq \min_{\substack{\mathcal{J}_{\mathcal{T}^{(t)}(\mathbf{X})} \subseteq \mathcal{V}_{\mathcal{T}^{(t)}(\mathbf{X})} \\ \text{s.t. } \mathcal{V}_{\mathcal{T}^{(t)}(\mathbf{X})}[\mathcal{I}] \subseteq \mathcal{J}_{\mathcal{T}^{(t)}(\mathbf{X})} \text{ and } \mathcal{V}_{\mathcal{T}^{(t)}(\mathbf{X})}[\mathcal{I}^c] \subseteq \mathcal{J}^c_{\mathcal{T}^{(t)}(\mathbf{X})}}} \widetilde{w}^{\Pi}_{\mathcal{T}^{(t)}(\mathbf{X})}(\mathcal{J}_{\mathcal{T}^{(t)}(\mathbf{X})}),$

$$(22)$$

*where* $\widetilde{w}^{\Pi}_{\mathcal{T}(\mathbf{X})}(\mathcal{J}_{\mathcal{T}(\mathbf{X})})$ *is the modified multiplicative cut weight of* $\mathcal{J}_{\mathcal{T}(\mathbf{X})}$ *in* $\mathcal{T}(\mathbf{X})$ *and* $\widetilde{w}^{\Pi}_{\mathcal{T}^{(t)}(\mathbf{X})}(\mathcal{J}_{\mathcal{T}^{(t)}(\mathbf{X})})$ *is the modified multiplicative cut weight of* $\mathcal{J}_{\mathcal{T}^{(t)}(\mathbf{X})}$ *in* $\mathcal{T}^{(t)}(\mathbf{X})$ *(Definition 8).*

*Proof.* We first prove Equation (21). Examining $\mathcal{T}(\mathbf{X})$, notice that: *(i)* by Proposition 1 its contraction yields $f^{(\theta,\mathcal{G})}(\mathbf{X})$; *(ii)* it has a tree structure; and *(iii)* each of its leaves is associated with a vertex feature vector, *i.e.* one of $\mathbf{x}^{(1)}, \ldots, \mathbf{x}^{(|\mathcal{V}|)}$, whereas all other aspects of the tensor network do not depend on $\mathbf{x}^{(1)}, \ldots, \mathbf{x}^{(|\mathcal{V}|)}$. Specifically, for any $\mathbf{X}$ and $\mathbf{X}'$ the nodes, legs, and leg weights of $\mathcal{T}(\mathbf{X})$ and $\mathcal{T}(\mathbf{X}')$ are identical, up to the assignment of features in the leaf nodes. Let $\boldsymbol{\mathcal{F}} \in \mathbb{R}^{D_x \times \cdots \times D_x}$ be the order $\rho_L(\mathcal{V}, \mathcal{V})$ tensor obtained by contracting all interior nodes in $\mathcal{T}(\mathbf{X})$. The above implies that we may write $f^{(\theta,\mathcal{G})}(\mathbf{X})$ as a contraction of $\boldsymbol{\mathcal{F}}$ with $\mathbf{x}^{(1)}, \ldots, \mathbf{x}^{(|\mathcal{V}|)}$. Specifically, it holds that:

$$f^{(\theta,\mathcal{G})}(\mathbf{X}) = \boldsymbol{\mathcal{F}} *_{n \in [\rho_L(\mathcal{V}, \mathcal{V})]} \mathbf{x}^{(\mu(n))}, \qquad (23)$$

for any $\mathbf{X} = (\mathbf{x}^{(1)}, \ldots, \mathbf{x}^{(|\mathcal{V}|)}) \in \mathbb{R}^{D_x \times |\mathcal{V}|}$, where $\mu : [\rho_L(\mathcal{V}, \mathcal{V})] \to \mathcal{V}$ maps a mode index of $\boldsymbol{\mathcal{F}}$ to the appropriate vertex of $\mathcal{G}$ according to $\mathcal{T}(\mathbf{X})$. Let $\mu^{-1}(\mathcal{I}) := \{ n \in [\rho_L(\mathcal{V}, \mathcal{V})] : \mu(n) \in \mathcal{I} \}$ be the mode indices of $\boldsymbol{\mathcal{F}}$ corresponding to vertices in $\mathcal{I}$. Invoking Lemma 2 leads to the following matricized form of Equation (23):

$$f^{(\theta,\mathcal{G})}(\mathbf{X}) = \big( \circ_{n \in \mu^{-1}(\mathcal{I})} \mathbf{x}^{(\mu(n))} \big)^{\top} [\![ \boldsymbol{\mathcal{F}}; \mu^{-1}(\mathcal{I}) ]\!] \big( \circ_{n \in \mu^{-1}(\mathcal{I}^c)} \mathbf{x}^{(\mu(n))} \big),$$

where $\circ$ denotes the Kronecker product.

We claim that $\mathrm{sep}(f^{(\theta,\mathcal{G})}; \mathcal{I}) \leq \mathrm{rank}[\![ \boldsymbol{\mathcal{F}}; \mu^{-1}(\mathcal{I}) ]\!]$. To see it is so, denote $R := \mathrm{rank}[\![ \boldsymbol{\mathcal{F}}; \mu^{-1}(\mathcal{I}) ]\!]$ and let $\mathbf{u}^{(1)}, \ldots, \mathbf{u}^{(R)} \in \mathbb{R}^{D_x^{\rho_L(\mathcal{I}, \mathcal{V})}}$ and $\bar{\mathbf{u}}^{(1)}, \ldots, \bar{\mathbf{u}}^{(R)} \in \mathbb{R}^{D_x^{\rho_L(\mathcal{I}^c, \mathcal{V})}}$ be such that $[\![ \boldsymbol{\mathcal{F}}; \mu^{-1}(\mathcal{I}) ]\!] = \sum_{r=1}^{R} \mathbf{u}^{(r)} (\bar{\mathbf{u}}^{(r)})^{\top}$. Then, defining $g^{(r)} : (\mathbb{R}^{D_x})^{|\mathcal{I}|} \to \mathbb{R}$ and $\bar{g}^{(r)} : (\mathbb{R}^{D_x})^{|\mathcal{I}^c|} \to \mathbb{R}$, for $r \in [R]$, as:

$$g^{(r)}(\mathbf{X}_{\mathcal{I}}) := \left\langle \circ_{n \in \mu^{-1}(\mathcal{I})} \mathbf{x}^{(\mu(n))}, \mathbf{u}^{(r)} \right\rangle \quad, \quad \bar{g}^{(r)}(\mathbf{X}_{\mathcal{I}^c}) := \left\langle \circ_{n \in \mu^{-1}(\mathcal{I}^c)} \mathbf{x}^{(\mu(n))}, \bar{\mathbf{u}}^{(r)} \right\rangle,$$

where $\mathbf{X}_{\mathcal{I}} := (\mathbf{x}^{(i)})_{i \in \mathcal{I}}$ and $\mathbf{X}_{\mathcal{I}^c} := (\mathbf{x}^{(j)})_{j \in \mathcal{I}^c}$, we have that:

$$\begin{aligned} f^{(\theta,\mathcal{G})}(\mathbf{X}) &= \big( \circ_{n \in \mu^{-1}(\mathcal{I})} \mathbf{x}^{(\mu(n))} \big)^{\top} \Big( \sum_{r=1}^{R} \mathbf{u}^{(r)} (\bar{\mathbf{u}}^{(r)})^{\top} \Big) \big( \circ_{n \in \mu^{-1}(\mathcal{I}^c)} \mathbf{x}^{(\mu(n))} \big) \\ &= \sum_{r=1}^{R} \left\langle \circ_{n \in \mu^{-1}(\mathcal{I})} \mathbf{x}^{(\mu(n))}, \mathbf{u}^{(r)} \right\rangle \cdot \left\langle \circ_{n \in \mu^{-1}(\mathcal{I}^c)} \mathbf{x}^{(\mu(n))}, \bar{\mathbf{u}}^{(r)} \right\rangle \\ &= \sum_{r=1}^{R} g^{(r)}(\mathbf{X}_{\mathcal{I}}) \cdot \bar{g}^{(r)}(\mathbf{X}_{\mathcal{I}^c}). \end{aligned}$$

Since $\mathrm{sep}(f^{(\theta,\mathcal{G})}; \mathcal{I})$ is the minimal number of summands in a representation of this form of $f^{(\theta,\mathcal{G})}$, indeed, $\mathrm{sep}(f^{(\theta,\mathcal{G})}; \mathcal{I}) \leq R = \mathrm{rank}[\![ \boldsymbol{\mathcal{F}}; \mu^{-1}(\mathcal{I}) ]\!]$.

What remains is to apply Claim 7 from [62], which upper bounds the rank of a tensor's matricization with multiplicative cut weights in a tree tensor network. In particular, consider an order $N \in \mathbb{N}$

tensor $\mathcal{A}$ produced by contracting a tree tensor network $\mathcal{T}$. Then, for any $\mathcal{K} \subseteq [N]$ we have that $\text{rank}[\![\mathcal{A}; \mathcal{K}]\!]$ is at most the minimal modified multiplicative cut weight in $\mathcal{T}$, among cuts separating leaves corresponding to modes $\mathcal{K}$ from leaves corresponding to modes $\mathcal{K}^c$. Thus, invoking Claim 7 from [62] establishes Equation (21):

$$\text{sep}\big(f^{(\theta,\mathcal{G})}; \mathcal{I}\big) \leq \text{rank}[\![\mathcal{F}; \mu^{-1}(\mathcal{I})]\!] \leq \min_{\substack{\mathcal{J}_{\mathcal{T}(\mathbf{X})} \subseteq \mathcal{V}_{\mathcal{T}(\mathbf{X})} \\ \text{s.t. } \mathcal{V}_{\mathcal{T}(\mathbf{X})}[\mathcal{I}] \subseteq \mathcal{J}_{\mathcal{T}(\mathbf{X})} \text{ and } \mathcal{V}_{\mathcal{T}(\mathbf{X})}[\mathcal{I}^c] \subseteq \mathcal{J}^c_{\mathcal{T}(\mathbf{X})}}} \widetilde{w}^{\Pi}_{\mathcal{T}(\mathbf{X})}(\mathcal{J}_{\mathcal{T}(\mathbf{X})}) \,.$$

Equation (22) readily follows by steps analogous to those used above for proving Equation (21). $\quad \square$

### I.2.2    Cut in Tensor Network for Graph Prediction (Proof of Equation (6))

For $\mathbf{X} = (\mathbf{x}^{(1)}, \ldots, \mathbf{x}^{(|\mathcal{V}|)}) \in \mathbb{R}^{D_x \times |\mathcal{V}|}$, let $\mathcal{T}(\mathbf{X}) = (\mathcal{V}_{\mathcal{T}(\mathbf{X})}, \mathcal{E}_{\mathcal{T}(\mathbf{X})}, w_{\mathcal{T}(\mathbf{X})})$ be the tensor network corresponding to $f^{(\theta,\mathcal{G})}(\mathbf{X})$ (detailed in Appendix E.3 and formally defined in Equations (15) to (17)). By Lemma 1, to prove that

$$\text{sep}\big(f^{(\theta,\mathcal{G})}; \mathcal{I}\big) \leq D_h^{4\rho_{L-1}(\mathcal{C}_{\mathcal{I}}, \mathcal{V})+1} \,,$$

it suffices to find $\mathcal{J}_{\mathcal{T}(\mathbf{X})} \subseteq \mathcal{V}_{\mathcal{T}(\mathbf{X})}$ satisfying: *(i)* leaves of $\mathcal{T}(\mathbf{X})$ associated with vertices in $\mathcal{I}$ are in $\mathcal{J}_{\mathcal{T}(\mathbf{X})}$, whereas leaves associated with vertices in $\mathcal{I}^c$ are not in $\mathcal{J}_{\mathcal{T}(\mathbf{X})}$; and *(ii)* $\widetilde{w}^{\Pi}_{\mathcal{T}(\mathbf{X})}(\mathcal{J}_{\mathcal{T}(\mathbf{X})}) \leq D_h^{4\rho_{L-1}(\mathcal{C}_{\mathcal{I}}, \mathcal{V})+1}$, where $\widetilde{w}^{\Pi}_{\mathcal{T}(\mathbf{X})}(\mathcal{J}_{\mathcal{T}(\mathbf{X})})$ is the modified multiplicative cut weight of $\mathcal{J}_{\mathcal{T}(\mathbf{X})}$ (Definition 8). To this end, define $\mathcal{J}_{\mathcal{T}(\mathbf{X})}$ to hold all nodes in $\mathcal{V}_{\mathcal{T}(\mathbf{X})}$ corresponding to vertices in $\mathcal{I}$. Formally:

$$\begin{aligned}
\mathcal{J}_{\mathcal{T}(\mathbf{X})} := &\Big\{ \mathbf{x}^{(i,\gamma)} : i \in \mathcal{I}, \gamma \in [\rho_L(\{i\}, \mathcal{V})] \Big\} \cup \\
&\Big\{ \mathbf{W}^{(l,i,\gamma)} : l \in [L], i \in \mathcal{I}, \gamma \in [\rho_{L-l+1}(\{i\}, \mathcal{V})] \Big\} \cup \\
&\Big\{ \boldsymbol{\delta}^{(l,i,\gamma)} : l \in [L], i \in \mathcal{I}, \gamma \in [\rho_{L-l}(\{i\}, \mathcal{V})] \Big\} \,.
\end{aligned}$$

Clearly, $\mathcal{J}_{\mathcal{T}(\mathbf{X})}$ upholds *(i)*.

As for *(ii)*, there are two types of legs crossing the cut induced by $\mathcal{J}_{\mathcal{T}(\mathbf{X})}$ in $\mathcal{T}(\mathbf{X})$. First, are those connecting a $\delta$-tensor with a weight matrix in the same layer, where one is associated with a vertex in $\mathcal{I}$ and the other with a vertex in $\mathcal{I}^c$. That is, legs connecting $\boldsymbol{\delta}^{(l,i,\gamma)}$ with $\mathbf{W}^{(l,\mathcal{N}(i)_j, \phi_{l,i,j}(\gamma))}$, where $i \in \mathcal{V}$ and $\mathcal{N}(i)_j \in \mathcal{V}$ are on different sides of the partition $(\mathcal{I}, \mathcal{I}^c)$ in the input graph, for $j \in [|\mathcal{N}(i)|], l \in [L], \gamma \in [\rho_{L-l}(\{i\}, \mathcal{V})]$. The $\delta$-tensors participating in these legs are exactly those associated with some $i \in \mathcal{C}_{\mathcal{I}}$ (recall $\mathcal{C}_{\mathcal{I}}$ is the set of vertices with an edge crossing the partition $(\mathcal{I}, \mathcal{I}^c)$). So, for every $l \in [L]$ and $i \in \mathcal{C}_{\mathcal{I}}$ there are $\rho_{L-l}(\{i\}, \mathcal{V})$ such $\delta$-tensors. Second, are legs from $\delta$-tensors associated with $i \in \mathcal{I}$ in the $L$'th layer to the $\delta$-tensor in the output layer of $\mathcal{T}(\mathbf{X})$. That is, legs connecting $\boldsymbol{\delta}^{(L,i,1)}$ with $\boldsymbol{\delta}^{(|\mathcal{V}|+1)}$, for $i \in \mathcal{I}$. Legs incident to the same $\delta$-tensor only contribute once to $\widetilde{w}^{\Pi}_{\mathcal{T}(\mathbf{X})}(\mathcal{J}_{\mathcal{T}(\mathbf{X})})$. Thus, since the weights of all legs connected to $\delta$-tensors are equal to $D_h$, we have that:

$$\widetilde{w}^{\Pi}_{\mathcal{T}(\mathbf{X})}(\mathcal{J}_{\mathcal{T}(\mathbf{X})}) \leq D_h^{1 + \sum_{l=1}^{L} \sum_{i \in \mathcal{C}_{\mathcal{I}}} \rho_{L-l}(\{i\}, \mathcal{V})} = D_h^{1 + \sum_{l=1}^{L} \rho_{L-l}(\mathcal{C}_{\mathcal{I}}, \mathcal{V})} \,.$$

Lastly, it remains to show that $\sum_{l=1}^{L} \rho_{L-l}(\mathcal{C}_{\mathcal{I}}, \mathcal{V}) \leq 4\rho_{L-1}(\mathcal{C}_{\mathcal{I}}, \mathcal{V})$, since in that case Lemma 1 implies:

$$\text{sep}\big(f^{(\theta,\mathcal{G})}; \mathcal{I}\big) \leq \widetilde{w}^{\Pi}_{\mathcal{T}(\mathbf{X})}(\mathcal{J}_{\mathcal{T}(\mathbf{X})}) \leq D_h^{4\rho_{L-1}(\mathcal{C}_{\mathcal{I}}, \mathcal{V})+1} \,,$$

which yields Equation (6) by taking the log of both sides.

The main idea is that, in an undirected graph with self-loops, the number of length $l \in \mathbb{N}$ walks from vertices with at least one neighbor decays exponentially when $l$ decreases. Observe that $\rho_l(\mathcal{C}_{\mathcal{I}}, \mathcal{V}) \leq \rho_{l+1}(\mathcal{C}_{\mathcal{I}}, \mathcal{V})$ for all $l \in \mathbb{N}$. Hence:

$$\sum_{l=1}^{L} \rho_{L-l}(\mathcal{C}_{\mathcal{I}}, \mathcal{V}) \leq 2 \sum_{l \in \{1,3,\ldots,L-1\}} \rho_{L-l}(\mathcal{C}_{\mathcal{I}}, \mathcal{V}) \,. \tag{24}$$

Furthermore, any length $l \in \mathbb{N}_{\geq 0}$ walk $i_0, i_1, \ldots, i_l \in \mathcal{V}$ from $\mathcal{C}_{\mathcal{I}}$ induces at least two walks of length $l+2$ from $\mathcal{C}_{\mathcal{I}}$, distinct from those induced by other length $l$ walks — one which goes twice through the

self-loop of $i_0$ and then proceeds according to the length $l$ walk, *i.e.* $i_0, i_0, i_0, i_1, \ldots, i_l$, and another that goes to a neighboring vertex (exists since $i_0 \in \mathcal{C}_\mathcal{I}$), returns to $i_0$, and then proceeds according to the length $l$ walk. This means that $\rho_{L-l}(\mathcal{C}_\mathcal{I}, \mathcal{V}) \leq 2^{-1} \cdot \rho_{L-l+2}(\mathcal{C}_\mathcal{I}, \mathcal{V}) \leq \cdots \leq 2^{-\lfloor l/2 \rfloor} \cdot \rho_{L-1}(\mathcal{C}_\mathcal{I}, \mathcal{V})$ for all $l \in \{3, 5, \ldots, L-1\}$. Going back to Equation (24), this leads to:

$$\sum\nolimits_{l=1}^{L} \rho_{L-l}(\mathcal{C}_\mathcal{I}, \mathcal{V}) \leq 2 \sum\nolimits_{l \in \{1,3,\ldots,L-1\}} 2^{\lfloor l/2 \rfloor} \cdot \rho_{L-1}(\mathcal{C}_\mathcal{I}, \mathcal{V})$$
$$\leq 2 \sum\nolimits_{l=0}^{\infty} 2^{-l} \cdot \rho_{L-1}(\mathcal{C}_\mathcal{I}, \mathcal{V})$$
$$= 4\rho_{L-1}(\mathcal{C}_\mathcal{I}, \mathcal{V}),$$

completing the proof of Equation (6).

### I.2.3 Cut in Tensor Network for Vertex Prediction (Proof of Equation (7))

This part of the proof follows a line similar to that of Appendix I.2.2, with differences stemming from the distinction between the operation of a GNN over graph and vertex prediction tasks.

For $\mathbf{X} = (\mathbf{x}^{(1)}, \ldots, \mathbf{x}^{(|\mathcal{V}|)}) \in \mathbb{R}^{D_x \times |\mathcal{V}|}$, let $\mathcal{T}^{(t)}(\mathbf{X}) = (\mathcal{V}_{\mathcal{T}^{(t)}(\mathbf{X})}, \mathcal{E}_{\mathcal{T}^{(t)}(\mathbf{X})}, w_{\mathcal{T}^{(t)}(\mathbf{X})})$ be the tensor network corresponding to $f^{(\theta, \mathcal{G}, t)}(\mathbf{X})$ (detailed in Appendix E.3 and formally defined in Equations (18) to (20)). By Lemma 1, to prove that

$$\text{sep}\big(f^{(\theta, \mathcal{G}, t)}; \mathcal{I}\big) \leq D_h^{4\rho_{L-1}(\mathcal{C}_\mathcal{I}, \{t\})},$$

it suffices to find $\mathcal{J}_{\mathcal{T}^{(t)}(\mathbf{X})} \subseteq \mathcal{V}_{\mathcal{T}^{(t)}(\mathbf{X})}$ satisfying: *(i)* leaves of $\mathcal{T}^{(t)}(\mathbf{X})$ associated with vertices in $\mathcal{I}$ are in $\mathcal{J}_{\mathcal{T}^{(t)}(\mathbf{X})}$, whereas leaves associated with vertices in $\mathcal{I}^c$ are not in $\mathcal{J}_{\mathcal{T}^{(t)}(\mathbf{X})}$; and *(ii)* $\widetilde{w}^\Pi_{\mathcal{T}^{(t)}(\mathbf{X})}(\mathcal{J}_{\mathcal{T}^{(t)}(\mathbf{X})}) \leq D_h^{4\rho_{L-1}(\mathcal{C}_\mathcal{I}, \{t\})}$, where $\widetilde{w}^\Pi_{\mathcal{T}^{(t)}(\mathbf{X})}(\mathcal{J}_{\mathcal{T}^{(t)}(\mathbf{X})})$ is the modified multiplicative cut weight of $\mathcal{J}_{\mathcal{T}^{(t)}(\mathbf{X})}$ (Definition 8). To this end, define $\mathcal{J}_{\mathcal{T}^{(t)}(\mathbf{X})}$ to hold all nodes in $\mathcal{V}_{\mathcal{T}^{(t)}(\mathbf{X})}$ corresponding to vertices in $\mathcal{I}$. Formally:

$$\mathcal{J}_{\mathcal{T}^{(t)}(\mathbf{X})} := \Big\{\mathbf{x}^{(i,\gamma)} : i \in \mathcal{I}, \gamma \in [\rho_L(\{i\}, \{t\})]\Big\} \cup$$
$$\Big\{\mathbf{W}^{(l,i,\gamma)} : l \in [L], i \in \mathcal{I}, \gamma \in [\rho_{L-l+1}(\{i\}, \{t\})]\Big\} \cup$$
$$\Big\{\boldsymbol{\delta}^{(l,i,\gamma)} : l \in [L], i \in \mathcal{I}, \gamma \in [\rho_{L-l}(\{i\}, \{t\})]\Big\} \cup$$
$$\mathcal{W}^{(o)},$$

where $\mathcal{W}^{(o)} := \{\mathbf{W}^{(o)}\}$ if $t \in \mathcal{I}$ and $\mathcal{W}^{(o)} := \emptyset$ otherwise. Clearly, $\mathcal{J}_{\mathcal{T}^{(t)}(\mathbf{X})}$ upholds *(i)*.

As for *(ii)*, the legs crossing the cut induced by $\mathcal{J}_{\mathcal{T}^{(t)}(\mathbf{X})}$ in $\mathcal{T}^{(t)}(\mathbf{X})$ are those connecting a $\delta$-tensor with a weight matrix in the same layer, where one is associated with a vertex in $\mathcal{I}$ and the other with a vertex in $\mathcal{I}^c$. That is, legs connecting $\boldsymbol{\delta}^{(l,i,\gamma)}$ with $\mathbf{W}^{(l,\mathcal{N}(i)_j,\phi^{(t)}_{l,i,j}(\gamma))}$, where $i \in \mathcal{V}$ and $\mathcal{N}(i)_j \in \mathcal{V}$ are on different sides of the partition $(\mathcal{I}, \mathcal{I}^c)$ in the input graph, for $j \in [|\mathcal{N}(i)|], l \in [L], \gamma \in [\rho_{L-l}(\{i\}, \{t\})]$. The $\delta$-tensors participating in these legs are exactly those associated with some $i \in \mathcal{C}_\mathcal{I}$ (recall $\mathcal{C}_\mathcal{I}$ is the set of vertices with an edge crossing the partition $(\mathcal{I}, \mathcal{I}^c)$). Hence, for every $l \in [L]$ and $i \in \mathcal{C}_\mathcal{I}$ there are $\rho_{L-l}(\{i\}, \{t\})$ such $\delta$-tensors. Legs connected to the same $\delta$-tensor only contribute once to $\widetilde{w}^\Pi_{\mathcal{T}^{(t)}(\mathbf{X})}(\mathcal{J}_{\mathcal{T}^{(t)}(\mathbf{X})})$. Thus, since the weights of all legs connected to $\delta$-tensors are equal to $D_h$, we have that:

$$\widetilde{w}^\Pi_{\mathcal{T}^{(t)}(\mathbf{X})}(\mathcal{J}_{\mathcal{T}^{(t)}(\mathbf{X})}) = D_h^{\sum_{l=1}^{L} \sum_{i \in \mathcal{C}_\mathcal{I}} \rho_{L-l}(\{i\}, \{t\})} = D_h^{\sum_{l=1}^{L} \rho_{L-l}(\mathcal{C}_\mathcal{I}, \{t\})}.$$

Lastly, it remains to show that $\sum_{l=1}^{L} \rho_{L-l}(\mathcal{C}_\mathcal{I}, \{t\}) \leq 4\rho_{L-1}(\mathcal{C}_\mathcal{I}, \{t\})$, as in that case Lemma 1 implies:

$$\text{sep}\big(f^{(\theta, \mathcal{G}, t)}; \mathcal{I}\big) \leq \widetilde{w}^\Pi_{\mathcal{T}^{(t)}(\mathbf{X})}(\mathcal{J}_{\mathcal{T}^{(t)}(\mathbf{X})}) \leq D_h^{4\rho_{L-1}(\mathcal{C}_\mathcal{I}, \{t\})},$$

which leads to Equation (7) by taking the log of both sides.

The main idea is that, in an undirected graph with self-loops, the number of length $l \in \mathbb{N}$ walks ending at $t$ that originate from vertices with at least one neighbor decays exponentially when $l$ decreases. First, clearly $\rho_l(\mathcal{C}_\mathcal{I}, \{t\}) \leq \rho_{l+1}(\mathcal{C}_\mathcal{I}, \{t\})$ for all $l \in \mathbb{N}$. Therefore:

$$\sum\nolimits_{l=1}^{L} \rho_{L-l}(\mathcal{C}_\mathcal{I}, \{t\}) \leq 2 \sum\nolimits_{l \in \{1,3,\ldots,L-1\}} \rho_{L-l}(\mathcal{C}_\mathcal{I}, \{t\}). \tag{25}$$

Furthermore, any length $l \in \mathbb{N}_{\geq 0}$ walk $i_0, i_1, \ldots, i_{l-1}, t \in \mathcal{V}$ from $\mathcal{C}_\mathcal{I}$ to $t$ induces at least two walks of length $l + 2$ from $\mathcal{C}_\mathcal{I}$ to $t$, distinct from those induced by other length $l$ walks — one which goes twice through the self-loop of $i_0$ and then proceeds according to the length $l$ walk, *i.e.* $i_0, i_0, i_0, i_1, \ldots, i_{l-1}, t$, and another that goes to a neighboring vertex (exists since $i_0 \in \mathcal{C}_\mathcal{I}$), returns to $i_0$, and then proceeds according to the length $l$ walk. This means that $\rho_{L-l}(\mathcal{C}_\mathcal{I}, \{t\}) \leq 2^{-1} \cdot \rho_{L-l+2}(\mathcal{C}_\mathcal{I}, \{t\}) \leq \cdots \leq 2^{-\lfloor l/2 \rfloor} \cdot \rho_{L-1}(\mathcal{C}_\mathcal{I}, \{t\})$ for all $l \in \{3, 5, \ldots, L-1\}$. Going back to Equation (25), we have that:

$$
\begin{aligned}
\sum\nolimits_{l=1}^{L} \rho_{L-l}(\mathcal{C}_\mathcal{I}, \{t\}) &\leq 2 \sum\nolimits_{l \in \{1,3,\ldots,L-1\}} 2^{\lfloor l/2 \rfloor} \cdot \rho_{L-1}(\mathcal{C}_\mathcal{I}, \{t\}) \\
&\leq 2 \sum\nolimits_{l=0}^{\infty} 2^{-l} \cdot \rho_{L-1}(\mathcal{C}_\mathcal{I}, \{t\}) \\
&= 4\rho_{L-1}(\mathcal{C}_\mathcal{I}, \{t\}),
\end{aligned}
$$

concluding the proof of Equation (7). $\qquad \square$

### I.2.4   Technical Lemma

**Lemma 2.** *For any order $N \in \mathbb{N}$ tensor $\boldsymbol{\mathcal{A}} \in \mathbb{R}^{D \times \cdots \times D}$, vectors $\mathbf{x}^{(1)}, \ldots, \mathbf{x}^{(N)} \in \mathbb{R}^D$, and subset of mode indices $\mathcal{I} \subseteq [N]$, it holds that $\boldsymbol{\mathcal{A}} *_{i \in [N]} \mathbf{x}^{(i)} = \left( \circ_{i \in I} \mathbf{x}^{(i)} \right)^\top [\![\boldsymbol{\mathcal{A}}; \mathcal{I}]\!] \left( \circ_{j \in \mathcal{I}^c} \mathbf{x}^{(j)} \right) \in \mathbb{R}$.*

*Proof.* The identity follows directly from the definitions of tensor contraction, matricization, and Kronecker product (Appendix I.1):

$$
\boldsymbol{\mathcal{A}} *_{i \in [N]} \mathbf{x}^{(i)} = \sum\nolimits_{d_1, \ldots, d_N = 1}^{D} \boldsymbol{\mathcal{A}}_{d_1, \ldots, d_N} \cdot \prod\nolimits_{i \in [N]} \mathbf{x}^{(i)}_{d_i} = \left( \circ_{i \in I} \mathbf{x}^{(i)} \right)^\top [\![\boldsymbol{\mathcal{A}}; \mathcal{I}]\!] \left( \circ_{j \in \mathcal{I}^c} \mathbf{x}^{(j)} \right).
$$

$\qquad \square$

### I.3   Proof of Theorem 3

We assume familiarity with the basic concepts from tensor analysis introduced in Appendix E.1.

We begin by establishing a general technique for lower bounding the separation rank of a function through *grid tensors*, also used in [64, 100, 65, 85]. For any $f : (\mathbb{R}^{D_x})^N \to \mathbb{R}$ and $M \in \mathbb{N}$ *template vectors* $\mathbf{v}^{(1)}, \ldots, \mathbf{v}^{(M)} \in \mathbb{R}^{D_x}$, we can create a grid tensor of $f$, which is a form of function discretization, by evaluating it over each point in $\{(\mathbf{v}^{(d_1)}, \ldots, \mathbf{v}^{(d_N)})\}_{d_1, \ldots, d_N = 1}^{M}$ and storing the outcomes in an order $N$ tensor with modes of dimension $M$. That is, the grid tensor of $f$ for templates $\mathbf{v}^{(1)}, \ldots, \mathbf{v}^{(M)}$, denoted $\boldsymbol{\mathcal{B}}(f) \in \mathbb{R}^{M \times \cdots \times M}$, is defined by $\boldsymbol{\mathcal{B}}(f)_{d_1, \ldots, d_N} = f(\mathbf{v}^{(d_1)}, \ldots, \mathbf{v}^{(d_N)})$ for all $d_1, \ldots, d_N \in [M]$.[16] Lemma 3 shows that $\mathrm{sep}(f; \mathcal{I})$ is lower bounded by the rank of $\boldsymbol{\mathcal{B}}(f)$'s matricization with respect to $\mathcal{I}$.

**Lemma 3.** *For $f : (\mathbb{R}^{D_x})^N \to \mathbb{R}$ and $M \in \mathbb{N}$ template vectors $\mathbf{v}^{(1)}, \ldots, \mathbf{v}^{(M)} \in \mathbb{R}^{D_x}$, let $\boldsymbol{\mathcal{B}}(f) \in \mathbb{R}^{M \times \cdots \times M}$ be the corresponding order $N$ grid tensor of $f$. Then, for any $\mathcal{I} \subseteq [N]$:*

$$
\mathrm{rank}[\![\boldsymbol{\mathcal{B}}(f); \mathcal{I}]\!] \leq \mathrm{sep}(f; \mathcal{I}).
$$

*Proof.* If $\mathrm{sep}(f; \mathcal{I})$ is $\infty$ or zero, *i.e.* $f$ cannot be represented as a finite sum of separable functions (with respect to $\mathcal{I}$) or is identically zero, then the claim is trivial. Otherwise, denote $R := \mathrm{sep}(f; \mathcal{I})$, and let $g^{(1)}, \ldots, g^{(R)} : (\mathbb{R}^{D_x})^{|\mathcal{I}|} \to \mathbb{R}$ and $\bar{g}^{(1)}, \ldots, \bar{g}^{(R)} : (\mathbb{R}^{D_x})^{|\mathcal{I}^c|} \to \mathbb{R}$ such that:

$$
f(\mathbf{X}) = \sum\nolimits_{r=1}^{R} g^{(r)}(\mathbf{X}_\mathcal{I}) \cdot \bar{g}^{(r)}(\mathbf{X}_{\mathcal{I}^c}), \tag{26}
$$

where $\mathbf{X} := (\mathbf{x}^{(1)}, \ldots, \mathbf{x}^{(N)})$, $\mathbf{X}_\mathcal{I} := (\mathbf{x}^{(i)})_{i \in \mathcal{I}}$, and $\mathbf{X}_{\mathcal{I}^c} := (\mathbf{x}^{(j)})_{j \in \mathcal{I}^c}$. For $r \in [R]$, let $\boldsymbol{\mathcal{B}}(g^{(r)})$ and $\boldsymbol{\mathcal{B}}(\bar{g}^{(r)})$ be the grid tensors of $g^{(r)}$ and $\bar{g}^{(r)}$ over templates $\mathbf{v}^{(1)}, \ldots, \mathbf{v}^{(M)}$, respectively. That is, $\boldsymbol{\mathcal{B}}(g^{(r)})_{d_i : i \in \mathcal{I}} = g^{(r)}((\mathbf{v}^{(d_i)})_{i \in \mathcal{I}})$ and $\boldsymbol{\mathcal{B}}(\bar{g}^{(r)})_{d_j : j \in \mathcal{I}^c} = \bar{g}^{(r)}((\mathbf{v}^{(d_j)})_{j \in \mathcal{I}^c})$ for all $d_1, \ldots, d_N \in [M]$.

---

[16]The template vectors of a grid tensor $\boldsymbol{\mathcal{B}}(f)$ will be clear from context, thus we omit them from the notation.

By Equation (26) we have that for any $d_1, \ldots, d_N \in [M]$:

$$\boldsymbol{\mathcal{B}}(f)_{d_1,\ldots,d_N} = f\big(\mathbf{v}^{(d_1)}, \ldots, \mathbf{v}^{(d_N)}\big)$$

$$= \sum_{r=1}^{R} g^{(r)}\big((\mathbf{v}^{(d_i)})_{i \in \mathcal{I}}\big) \cdot \bar{g}^{(r)}\big((\mathbf{v}^{(d_j)})_{j \in \mathcal{I}^c}\big)$$

$$= \sum_{r=1}^{R} \boldsymbol{\mathcal{B}}\big(g^{(r)}\big)_{d_i : i \in \mathcal{I}} \cdot \boldsymbol{\mathcal{B}}\big(\bar{g}^{(r)}\big)_{d_j : j \in \mathcal{I}^c} .$$

Denoting by $\mathbf{u}^{(r)} \in \mathbb{R}^{M^{|\mathcal{I}|}}$ and $\bar{\mathbf{u}}^{(r)} \in \mathbb{R}^{M^{|\mathcal{I}^c|}}$ the arrangements of $\boldsymbol{\mathcal{B}}(g^{(r)})$ and $\boldsymbol{\mathcal{B}}(\bar{g}^{(r)})$ as vectors, respectively for $r \in [R]$, this implies that the matricization of $\boldsymbol{\mathcal{B}}(f)$ with respect to $\mathcal{I}$ can be written as:

$$[\![\boldsymbol{\mathcal{B}}(f); \mathcal{I}]\!] = \sum_{r=1}^{R} \mathbf{u}^{(r)}\big(\bar{\mathbf{u}}^{(r)}\big)^\top .$$

We have arrived at a representation of $[\![\boldsymbol{\mathcal{B}}(f); \mathcal{I}]\!]$ as a sum of $R$ outer products between two vectors. An outer product of two vectors is a matrix of rank at most one. Consequently, by sub-additivity of rank we conclude: $\mathrm{rank}[\![\boldsymbol{\mathcal{B}}(f); \mathcal{I}]\!] \leq R = \mathrm{sep}(f; \mathcal{I})$. $\qquad\square$

In the context of graph prediction, let $\mathcal{C}^* \in \mathrm{argmax}_{\mathcal{C} \in \mathcal{S}(\mathcal{I})} \log(\alpha_{\mathcal{C}}) \cdot \rho_{L-1}(\mathcal{C}, \mathcal{V})$. By Lemma 3, to prove that Equation (8) holds for weights $\theta$, it suffices to find template vectors for which $\log(\mathrm{rank}[\![\boldsymbol{\mathcal{B}}(f^{(\theta,\mathcal{G})}); \mathcal{I}]\!]) \geq \log(\alpha_{\mathcal{C}^*}) \cdot \rho_{L-1}(\mathcal{C}^*, \mathcal{V})$. Notice that, since the outputs of $f^{(\theta,\mathcal{G})}$ vary polynomially with the weights $\theta$, so do the entries of $[\![\boldsymbol{\mathcal{B}}(f^{(\theta,\mathcal{G})}); \mathcal{I}]\!]$ for any choice of template vectors. Thus, according to Lemma 9, by constructing weights $\theta$ and template vectors satisfying $\log(\mathrm{rank}[\![\boldsymbol{\mathcal{B}}(f^{(\theta,\mathcal{G})}); \mathcal{I}]\!]) \geq \log(\alpha_{\mathcal{C}^*}) \cdot \rho_{L-1}(\mathcal{C}^*, \mathcal{V})$, we may conclude that this is the case for almost all assignments of weights, meaning Equation (8) holds for almost all assignments of weights. In Appendix I.3.1 we construct such weights and template vectors.

In the context of vertex prediction, let $\mathcal{C}_t^* \in \mathrm{argmax}_{\mathcal{C} \in \mathcal{S}(\mathcal{I})} \log(\alpha_{\mathcal{C},t}) \cdot \rho_{L-1}(\mathcal{C}, \{t\})$. Due to arguments analogous to those above, to prove that Equation (9) holds for almost all assignments of weights, we need only find weights $\theta$ and template vectors satisfying $\log(\mathrm{rank}[\![\boldsymbol{\mathcal{B}}(f^{(\theta,\mathcal{G},t)}); \mathcal{I}]\!]) \geq \log(\alpha_{\mathcal{C}_t^*,t}) \cdot \rho_{L-1}(\mathcal{C}_t^*, \{t\})$. In Appendix I.3.2 we do so.

Lastly, recalling that a finite union of measure zero sets has measure zero as well establishes that Equations (8) and (9) jointly hold for almost all assignments of weights. $\qquad\square$

### I.3.1 Weights and Template Vectors Assignment for Graph Prediction (Proof of Equation (8))

We construct weights $\theta$ and template vectors satisfying $\log(\mathrm{rank}[\![\boldsymbol{\mathcal{B}}(f^{(\theta,\mathcal{G})}); \mathcal{I}]\!]) \geq \log(\alpha_{\mathcal{C}^*}) \cdot \rho_{L-1}(\mathcal{C}^*, \mathcal{V})$, where $\mathcal{C}^* \in \mathrm{argmax}_{\mathcal{C} \in \mathcal{S}(\mathcal{I})} \log(\alpha_{\mathcal{C}}) \cdot \rho_{L-1}(\mathcal{C}, \mathcal{V})$.

If $\rho_{L-1}(\mathcal{C}^*, \mathcal{V}) = 0$, then the claim is trivial since there exist weights and template vectors for which $[\![\boldsymbol{\mathcal{B}}(f^{(\theta,\mathcal{G})}); \mathcal{I}]\!]$ is not the zero matrix (e.g. taking all weight matrices to be zero-padded identity matrices and choosing a single template vector holding one in its first entry and zeros elsewhere).

Now, assuming that $\rho_{L-1}(\mathcal{C}^*, \mathcal{V}) > 0$, which in particular implies that $\mathcal{I} \neq \emptyset, \mathcal{I} \neq \mathcal{V}$, and $\mathcal{C}^* \neq \emptyset$, we begin with the case of GNN depth $L = 1$, after which we treat the more general $L \geq 2$ case.

**Case of $L = 1$:** Consider the weights $\theta = (\mathbf{W}^{(1)}, \mathbf{W}^{(o)})$ given by $\mathbf{W}^{(1)} := \mathbf{I} \in \mathbb{R}^{D_h \times D_x}$ and $\mathbf{W}^{(o)} := (1, \ldots, 1) \in \mathbb{R}^{1 \times D_h}$, where $\mathbf{I}$ is a zero padded identity matrix, i.e. it holds ones on its diagonal and zeros elsewhere. We choose template vectors $\mathbf{v}^{(1)}, \ldots, \mathbf{v}^{(D)} \in \mathbb{R}^{D_x}$ such that $\mathbf{v}^{(m)}$ holds the $m$'th standard basis vector of $\mathbb{R}^D$ in its first $D$ coordinates and zeros in the remaining entries, for $m \in [D]$ (recall $D := \min\{D_x, D_h\}$). Namely, denote by $\mathbf{e}^{(1)}, \ldots, \mathbf{e}^{(D)} \in \mathbb{R}^D$ the standard basis vectors of $\mathbb{R}^D$, i.e. $\mathbf{e}_d^{(m)} = 1$ if $d = m$ and $\mathbf{e}_d^{(m)} = 0$ otherwise for all $m, d \in [D]$. We let $\mathbf{v}_{:D}^{(m)} := \mathbf{e}^{(m)}$ and $\mathbf{v}_{D+1:}^{(m)} := 0$ for all $m \in [D]$.

We prove that for this choice of weights and template vectors, for all $d_1, \ldots, d_{|\mathcal{V}|} \in [D]$:

$$f^{(\theta,\mathcal{G})}\big(\mathbf{v}^{(d_1)}, \ldots, \mathbf{v}^{(d_{|\mathcal{V}|})}\big) = \begin{cases} 1 & \text{, if } d_1 = \cdots = d_{|\mathcal{V}|} \\ 0 & \text{, otherwise} \end{cases} . \tag{27}$$

To see it is so, notice that:

$$f^{(\theta,\mathcal{G})}\big(\mathbf{v}^{(d_1)},\ldots,\mathbf{v}^{(d_{|\mathcal{V}|})}\big) = \mathbf{W}^{(o)}\big(\odot_{i\in\mathcal{V}}\mathbf{h}^{(1,i)}\big) = \sum_{d=1}^{D_h}\prod_{i\in\mathcal{V}}\mathbf{h}_d^{(1,i)}\,,$$

with $\mathbf{h}^{(1,i)} = \odot_{j\in\mathcal{N}(i)}(\mathbf{W}^{(1)}\mathbf{v}^{(d_j)}) = \odot_{j\in\mathcal{N}(i)}(\mathbf{I}\mathbf{v}^{(d_j)})$ for all $i\in\mathcal{V}$. Since $\mathbf{v}_{:D}^{(d_j)} = \mathbf{e}^{(d_j)}$ for all $j\in\mathcal{N}(i)$ and $\mathbf{I}$ is a zero-padded $D\times D$ identity matrix, it holds that:

$$f^{(\theta,\mathcal{G})}\big(\mathbf{v}^{(d_1)},\ldots,\mathbf{v}^{(d_{|\mathcal{V}|})}\big) = \sum_{d=1}^{D}\prod_{i\in\mathcal{V},j\in\mathcal{N}(i)}\mathbf{e}_d^{(d_j)}\,.$$

Due to the existence of self-loops (*i.e.* $i\in\mathcal{N}(i)$ for all $i\in\mathcal{V}$), for every $d\in[D]$ the product $\prod_{i\in\mathcal{V},j\in\mathcal{N}(i)}\mathbf{e}_d^{(d_j)}$ includes each of $\mathbf{e}_d^{(d_1)},\ldots,\mathbf{e}_d^{(d_{|\mathcal{V}|})}$ at least once. Consequently, $\prod_{i\in\mathcal{V},j\in\mathcal{N}(i)}\mathbf{e}_d^{(d_j)} = 1$ if $d_1 = \cdots = d_{|\mathcal{V}|} = d$ and $\prod_{i\in\mathcal{V},j\in\mathcal{N}(i)}\mathbf{e}_d^{(d_j)} = 0$ otherwise. This implies that $f^{(\theta,\mathcal{G})}(\mathbf{v}^{(d_1)},\ldots,\mathbf{v}^{(d_{|\mathcal{V}|})}) = 1$ if $d_1 = \cdots = d_{|\mathcal{V}|}$ and $f^{(\theta,\mathcal{G})}(\mathbf{v}^{(d_1)},\ldots,\mathbf{v}^{(d_{|\mathcal{V}|})}) = 0$ otherwise, for all $d_1,\ldots,d_{|\mathcal{V}|}\in[D]$.

Equation (27) implies that $\llbracket\mathcal{B}(f^{(\theta,\mathcal{G})});\mathcal{I}\rrbracket$ has exactly $D$ non-zero entries, each in a different row and column. Thus, $\mathrm{rank}\llbracket\mathcal{B}(f^{(\theta,\mathcal{G})});\mathcal{I}\rrbracket = D$. Recalling that $\alpha_{\mathcal{C}^*} := D^{1/\rho_0(\mathcal{C}^*,\mathcal{V})}$ for $L = 1$, we conclude:

$$\log\Big(\mathrm{rank}\Big[\!\Big[\mathcal{B}\Big(f^{(\theta,\mathcal{G})}\Big);\mathcal{I}\Big]\!\Big]\Big) = \log(D) = \log(\alpha_{\mathcal{C}^*})\cdot\rho_0(\mathcal{C}^*,\mathcal{V})\,.$$

**Case of $L\geq 2$:** Let $M := \big(\!\big(\begin{smallmatrix}D\\\rho_{L-1}(\mathcal{C}^*,\mathcal{V})\end{smallmatrix}\big)\!\big) = \binom{D+\rho_{L-1}(\mathcal{C}^*,\mathcal{V})-1}{\rho_{L-1}(\mathcal{C}^*,\mathcal{V})}$ be the multiset coefficient of $D$ and $\rho_{L-1}(\mathcal{C}^*,\mathcal{V})$ (recall $D := \min\{D_x,D_h\}$). By Lemma 7, there exists $\mathbf{Z}\in\mathbb{R}_{>0}^{M\times D}$ for which

$$\mathrm{rank}\Big(\odot^{\rho_{L-1}(\mathcal{C}^*,\mathcal{V})}(\mathbf{Z}\mathbf{Z}^\top)\Big) = \left(\!\!\left(\begin{matrix}D\\\rho_{L-1}(\mathcal{C}^*,\mathcal{V})\end{matrix}\right)\!\!\right),$$

with $\odot^{\rho_{L-1}(\mathcal{C}^*,\mathcal{V})}(\mathbf{Z}\mathbf{Z}^\top)$ standing for the $\rho_{L-1}(\mathcal{C}^*,\mathcal{V})$'th Hadamard power of $\mathbf{Z}\mathbf{Z}^\top$. For this $\mathbf{Z}$, by Lemma 4 below we know that there exist weights $\theta$ and template vectors such that $\llbracket\mathcal{B}(f^{(\theta,\mathcal{G})});\mathcal{I}\rrbracket$ has an $M\times M$ sub-matrix of the form $\mathbf{S}(\odot^{\rho_{L-1}(\mathcal{C}^*,\mathcal{V})}(\mathbf{Z}\mathbf{Z}^\top))\mathbf{Q}$, where $\mathbf{S},\mathbf{Q}\in\mathbb{R}^{M\times M}$ are full-rank diagonal matrices. Since the rank of a matrix is at least the rank of any of its sub-matrices:

$$\mathrm{rank}\Big(\Big[\!\Big[\mathcal{B}(f^{(\theta,\mathcal{G})});\mathcal{I}\Big]\!\Big]\Big) \geq \mathrm{rank}\Big(\mathbf{S}\Big(\odot^{\rho_{L-1}(\mathcal{C}^*,\mathcal{V})}(\mathbf{Z}\mathbf{Z}^\top)\Big)\mathbf{Q}\Big)$$
$$= \mathrm{rank}\Big(\odot^{\rho_{L-1}(\mathcal{C}^*,\mathcal{V})}(\mathbf{Z}\mathbf{Z}^\top)\Big)$$
$$= \left(\!\!\left(\begin{matrix}D\\\rho_{L-1}(\mathcal{C}^*,\mathcal{V})\end{matrix}\right)\!\!\right),$$

where the second transition stems from $\mathbf{S}$ and $\mathbf{Q}$ being full-rank. Applying Lemma 8 to lower bound the multiset coefficient, we have that:

$$\mathrm{rank}\Big(\Big[\!\Big[\mathcal{B}(f^{(\theta,\mathcal{G})});\mathcal{I}\Big]\!\Big]\Big) \geq \left(\!\!\left(\begin{matrix}D\\\rho_{L-1}(\mathcal{C}^*,\mathcal{V})\end{matrix}\right)\!\!\right) \geq \left(\frac{D-1}{\rho_{L-1}(\mathcal{C}^*,\mathcal{V})}+1\right)^{\rho_{L-1}(\mathcal{C}^*,\mathcal{V})}.$$

Taking the log of both sides while recalling that $\alpha_{\mathcal{C}^*} := (D-1)\cdot\rho_{L-1}(\mathcal{C}^*,\mathcal{V})^{-1}+1$, we conclude that:

$$\log(\mathrm{rank}\Big[\!\Big[\mathcal{B}\Big(f^{(\theta,\mathcal{G})}\Big);\mathcal{I}\Big]\!\Big]) \geq \log(\alpha_{\mathcal{C}^*})\cdot\rho_{L-1}(\mathcal{C}^*,\mathcal{V})\,.$$

**Lemma 4.** *Suppose that the GNN inducing $f^{(\theta,\mathcal{G})}$ is of depth $L\geq 2$ and that $\rho_{L-1}(\mathcal{C}^*,\mathcal{V}) > 0$. For any $M\in\mathbb{N}$ and matrix with positive entries $\mathbf{Z}\in\mathbb{R}_{>0}^{M\times D}$, there exist weights $\theta$ and $M+1$ template vectors $\mathbf{v}^{(1)},\ldots,\mathbf{v}^{(M+1)}\in\mathbb{R}^{D_x}$ such that $\llbracket\mathcal{B}(f^{(\theta,\mathcal{G})});\mathcal{I}\rrbracket$ has an $M\times M$ sub-matrix $\mathbf{S}(\odot^{\rho_{L-1}(\mathcal{C}^*,\mathcal{V})}(\mathbf{Z}\mathbf{Z}^\top))\mathbf{Q}$, where $\mathbf{S},\mathbf{Q}\in\mathbb{R}^{M\times M}$ are full-rank diagonal matrices and $\odot^{\rho_{L-1}(\mathcal{C}^*,\mathcal{V})}(\mathbf{Z}\mathbf{Z}^\top)$ is the $\rho_{L-1}(\mathcal{C}^*,\mathcal{V})$'th Hadamard power of $\mathbf{Z}\mathbf{Z}^\top$.*

*Proof.* Consider the weights $\theta = (\mathbf{W}^{(1)}, \ldots, \mathbf{W}^{(L)}, \mathbf{W}^{(o)})$ given by:

$$\mathbf{W}^{(1)} := \mathbf{I} \in \mathbb{R}^{D_h \times D_x},$$

$$\mathbf{W}^{(2)} := \begin{pmatrix} 1 & 1 & \cdots & 1 \\ 0 & 0 & \cdots & 0 \\ \vdots & \vdots & \cdots & \vdots \\ 0 & 0 & \cdots & 0 \end{pmatrix} \in \mathbb{R}^{D_h \times D_h},$$

$$\forall l \in \{3, \ldots, L\}: \mathbf{W}^{(l)} := \begin{pmatrix} 1 & 0 & \cdots & 0 \\ 0 & 0 & \cdots & 0 \\ \vdots & \vdots & \cdots & \vdots \\ 0 & 0 & \cdots & 0 \end{pmatrix} \in \mathbb{R}^{D_h \times D_h},$$

$$\mathbf{W}^{(o)} := \begin{pmatrix} 1 & 0 & \cdots & 0 \end{pmatrix} \in \mathbb{R}^{1 \times D_h},$$

where $\mathbf{I}$ is a zero padded identity matrix, *i.e.* it holds ones on its diagonal and zeros elsewhere. We define the templates $\mathbf{v}^{(1)}, \ldots, \mathbf{v}^{(M)} \in \mathbb{R}^{D_x}$ to be the vectors holding the respective rows of $\mathbf{Z}$ in their first $D$ coordinates and zeros in the remaining entries (recall $D := \min\{D_x, D_h\}$). That is, denoting the rows of $\mathbf{Z}$ by $\mathbf{z}^{(1)}, \ldots, \mathbf{z}^{(M)} \in \mathbb{R}^D_{>0}$, we let $\mathbf{v}^{(m)}_{:D} := \mathbf{z}^{(m)}$ and $\mathbf{v}^{(m)}_{D+1:} := 0$ for all $m \in [M]$. We set all entries of the last template vector to one, *i.e.* $\mathbf{v}^{(M+1)} := (1, \ldots, 1) \in \mathbb{R}^{D_x}$.

Since $\mathcal{C}^* \in \mathcal{S}(\mathcal{I})$, *i.e.* it is an admissible subset of $\mathcal{C}_{\mathcal{I}}$ (Definition 4), there exist $\mathcal{I}' \subseteq \mathcal{I}, \mathcal{J}' \subseteq \mathcal{I}^c$ with no repeating shared neighbors (Definition 3) such that $\mathcal{C}^* = \mathcal{N}(\mathcal{I}') \cap \mathcal{N}(\mathcal{J}')$. Notice that $\mathcal{I}'$ and $\mathcal{J}'$ are non-empty as $\mathcal{C}^* \neq \emptyset$ (this is implied by $\rho_{L-1}(\mathcal{C}^*, \mathcal{V}) > 0$). We focus on the $M \times M$ sub-matrix of $[\![\mathcal{B}(f^{(\theta, \mathcal{G})}); \mathcal{I}]\!]$ that includes only rows and columns corresponding to evaluations of $f^{(\theta, \mathcal{G})}$ where all variables indexed by $\mathcal{I}'$ are assigned the same template vector from $\mathbf{v}^{(1)}, \ldots, \mathbf{v}^{(M)}$, all variables indexed by $\mathcal{J}'$ are assigned the same template vector from $\mathbf{v}^{(1)}, \ldots, \mathbf{v}^{(M)}$, and all remaining variables are assigned the all-ones template vector $\mathbf{v}^{(M+1)}$. Denoting this sub-matrix by $\mathbf{U} \in \mathbb{R}^{M \times M}$, it therefore upholds:

$$\mathbf{U}_{m,n} = f^{(\theta, \mathcal{G})}\left( \left(\mathbf{x}^{(i)} \leftarrow \mathbf{v}^{(m)}\right)_{i \in \mathcal{I}'}, \left(\mathbf{x}^{(j)} \leftarrow \mathbf{v}^{(n)}\right)_{j \in \mathcal{J}'}, \left(\mathbf{x}^{(k)} \leftarrow \mathbf{v}^{(M+1)}\right)_{k \in \mathcal{V} \setminus (\mathcal{I}' \cup \mathcal{J}')} \right),$$

for all $m, n \in [M]$, where we use $(\mathbf{x}^{(i)} \leftarrow \mathbf{v}^{(m)})_{i \in \mathcal{I}'}$ to denote that input variables indexed by $\mathcal{I}'$ are assigned the value $\mathbf{v}^{(m)}$. To show that $\mathbf{U}$ obeys the form $\mathbf{S}(\odot^{\rho_{L-1}(\mathcal{C}^*, \mathcal{V})}(\mathbf{Z}\mathbf{Z}^\top))\mathbf{Q}$ for full-rank diagonal $\mathbf{S}, \mathbf{Q} \in \mathbb{R}^{M \times M}$, we prove there exist $\phi, \psi : \mathbb{R}^{D_x} \to \mathbb{R}_{>0}$ such that $\mathbf{U}_{m,n} = \phi(\mathbf{v}^{(m)})\langle \mathbf{z}^{(m)}, \mathbf{z}^{(n)} \rangle^{\rho_{L-1}(\mathcal{C}^*, \mathcal{V})} \psi(\mathbf{v}^{(n)})$ for all $m, n \in [M]$. Indeed, defining $\mathbf{S}$ to hold $\phi(\mathbf{v}^{(1)}), \ldots, \phi(\mathbf{v}^{(M)})$ on its diagonal and $\mathbf{Q}$ to hold $\psi(\mathbf{v}^{(1)}), \ldots, \psi(\mathbf{v}^{(M)})$ on its diagonal, we have that $\mathbf{U} = \mathbf{S}(\odot^{\rho_{L-1}(\mathcal{C}^*, \mathcal{V})}(\mathbf{Z}\mathbf{Z}^\top))\mathbf{Q}$. Since $\mathbf{S}$ and $\mathbf{Q}$ are clearly full-rank (diagonal matrices with non-zero entries on their diagonal), the proof concludes.

For $m, n \in [M]$, let $\mathbf{h}^{(l,i)} \in \mathbb{R}^{D_h}$ be the hidden embedding for $i \in \mathcal{V}$ at layer $l \in [L]$ of the GNN inducing $f^{(\theta, \mathcal{G})}$, over the following assignment to its input variables (*i.e.* vertex features):

$$\left(\mathbf{x}^{(i)} \leftarrow \mathbf{v}^{(m)}\right)_{i \in \mathcal{I}'}, \left(\mathbf{x}^{(j)} \leftarrow \mathbf{v}^{(n)}\right)_{j \in \mathcal{J}'}, \left(\mathbf{x}^{(k)} \leftarrow \mathbf{v}^{(M+1)}\right)_{k \in \mathcal{V} \setminus (\mathcal{I}' \cup \mathcal{J}')}.$$

Invoking Lemma 10 with $\mathbf{v}^{(m)}, \mathbf{v}^{(n)}, \mathcal{I}'$, and $\mathcal{J}'$, for all $i \in \mathcal{V}$ it holds that:

$$\mathbf{h}_1^{(L,i)} = \phi^{(L,i)}\left(\mathbf{v}^{(m)}\right)\langle \mathbf{z}^{(m)}, \mathbf{z}^{(n)} \rangle^{\rho_{L-1}(\mathcal{C}^*, \{i\})} \psi^{(L,i)}\left(\mathbf{v}^{(n)}\right) \quad, \quad \forall d \in \{2, \ldots, D_h\}: \mathbf{h}_d^{(L,i)} = 0,$$

for some $\phi^{(L,i)}, \psi^{(L,i)} : \mathbb{R}^{D_x} \to \mathbb{R}_{>0}$. Since

$$\mathbf{U}_{m,n} = f^{(\theta, \mathcal{G})}\left( \left(\mathbf{x}^{(i)} \leftarrow \mathbf{v}^{(m)}\right)_{i \in \mathcal{I}'}, \left(\mathbf{x}^{(j)} \leftarrow \mathbf{v}^{(n)}\right)_{j \in \mathcal{J}'}, \left(\mathbf{x}^{(k)} \leftarrow \mathbf{v}^{(M+1)}\right)_{k \in \mathcal{V} \setminus (\mathcal{I}' \cup \mathcal{J}')} \right)$$

$$= \mathbf{W}^{(o)}\left( \odot_{i \in \mathcal{V}} \mathbf{h}^{(L,i)} \right)$$

and $\mathbf{W}^{(o)} = (1, 0, \ldots, 0)$, this implies that:

$$\mathbf{U}_{m,n} = \prod_{i \in \mathcal{V}} \mathbf{h}_1^{(L,i)}$$

$$= \prod_{i \in \mathcal{V}} \phi^{(L,i)}\left(\mathbf{v}^{(m)}\right)\langle \mathbf{z}^{(m)}, \mathbf{z}^{(n)} \rangle^{\rho_{L-1}(\mathcal{C}^*, \{i\})} \psi^{(L,i)}\left(\mathbf{v}^{(n)}\right).$$

Rearranging the last term leads to:

$$\mathbf{U}_{m,n} = \left(\prod_{i \in \mathcal{V}} \phi^{(L,i)}\big(\mathbf{v}^{(m)}\big)\right) \cdot \big\langle \mathbf{z}^{(m)}, \mathbf{z}^{(n)} \big\rangle^{\sum_{i \in \mathcal{V}} \rho_{L-1}(\mathcal{C}^*, \{i\})} \cdot \left(\prod_{i \in \mathcal{V}} \psi^{(L,i)}\big(\mathbf{v}^{(n)}\big)\right).$$

Let $\phi : \mathbf{v} \mapsto \prod_{i \in \mathcal{V}} \phi^{(L,i)}(\mathbf{v})$ and $\psi : \mathbf{v} \mapsto \prod_{i \in \mathcal{V}} \psi^{(L,i)}(\mathbf{v})$. Noticing that their range is indeed $\mathbb{R}_{>0}$ and that $\sum_{i \in \mathcal{V}} \rho_{L-1}(\mathcal{C}^*, \{i\}) = \rho_{L-1}(\mathcal{C}^*, \mathcal{V})$ yields the sought-after expression for $\mathbf{U}_{m,n}$:

$$\mathbf{U}_{m,n} = \phi\big(\mathbf{v}^{(m)}\big) \big\langle \mathbf{z}^{(m)}, \mathbf{z}^{(n)} \big\rangle^{\rho_{L-1}(\mathcal{C}^*, \mathcal{V})} \psi\big(\mathbf{v}^{(n)}\big).$$

$\square$

### I.3.2 Weights and Template Vectors Assignment for Vertex Prediction (Proof of Equation (9))

This part of the proof follows a line similar to that of Appendix I.3.1, with differences stemming from the distinction between the operation of a GNN over graph and vertex prediction. Namely, we construct weights $\theta$ and template vectors satisfying $\log(\text{rank}\big[\!\big[\boldsymbol{\mathcal{B}}\big(f^{(\theta,\mathcal{G},t)}\big); \mathcal{I}\big]\!\big]) \geq \log(\alpha_{\mathcal{C}_t^*,t}) \cdot \rho_{L-1}(\mathcal{C}_t^*, \{t\})$, where $\mathcal{C}_t^* \in \text{argmax}_{\mathcal{C} \in \mathcal{S}(\mathcal{I})} \log(\alpha_{\mathcal{C},t}) \cdot \rho_{L-1}(\mathcal{C}, \{t\})$.

If $\rho_{L-1}(\mathcal{C}_t^*, \{t\}) = 0$, then the claim is trivial since there exist weights and template vectors for which $\big[\!\big[\boldsymbol{\mathcal{B}}\big(f^{(\theta,\mathcal{G},t)}\big); \mathcal{I}\big]\!\big]$ is not the zero matrix (*e.g.* taking all weight matrices to be zero-padded identity matrices and choosing a single template vector holding one in its first entry and zeros elsewhere).

Now, assuming that $\rho_{L-1}(\mathcal{C}_t^*, \{t\}) > 0$, which in particular implies that $\mathcal{I} \neq \emptyset, \mathcal{I} \neq \mathcal{V}$, and $\mathcal{C}_t^* \neq \emptyset$, we begin with the case of GNN depth $L = 1$, after which we treat the more general $L \geq 2$ case.

**Case of $L = 1$:** Consider the weights $\theta = (\mathbf{W}^{(1)}, \mathbf{W}^{(o)})$ given by $\mathbf{W}^{(1)} := \mathbf{I} \in \mathbb{R}^{D_h \times D_x}$ and $\mathbf{W}^{(o)} := (1, \ldots, 1) \in \mathbb{R}^{1 \times D_h}$, where $\mathbf{I}$ is a zero padded identity matrix, *i.e.* it holds ones on its diagonal and zeros elsewhere. We choose template vectors $\mathbf{v}^{(1)}, \ldots, \mathbf{v}^{(D)} \in \mathbb{R}^{D_x}$ such that $\mathbf{v}^{(m)}$ holds the $m$'th standard basis vector of $\mathbb{R}^D$ in its first $D$ coordinates and zeros in the remaining entries, for $m \in [D]$ (recall $D := \min\{D_x, D_h\}$). Namely, denote by $\mathbf{e}^{(1)}, \ldots, \mathbf{e}^{(D)} \in \mathbb{R}^D$ the standard basis vectors of $\mathbb{R}^D$, *i.e.* $\mathbf{e}_d^{(m)} = 1$ if $d = m$ and $\mathbf{e}_d^{(m)} = 0$ otherwise for all $m, d \in [D]$. We let $\mathbf{v}_{:D}^{(m)} := \mathbf{e}^{(m)}$ and $\mathbf{v}_{D+1:}^{(m)} := 0$ for all $m \in [D]$.

We prove that for this choice of weights and template vectors, for all $d_1, \ldots, d_{|\mathcal{V}|} \in [D]$:

$$f^{(\theta,\mathcal{G},t)}\big(\mathbf{v}^{(d_1)}, \ldots, \mathbf{v}^{(d_{|\mathcal{V}|})}\big) = \begin{cases} 1 & \text{, if } d_j = d_{j'} \text{ for all } j, j' \in \mathcal{N}(t) \\ 0 & \text{, otherwise} \end{cases}. \tag{28}$$

To see it is so, notice that:

$$f^{(\theta,\mathcal{G},t)}\big(\mathbf{v}^{(d_1)}, \ldots, \mathbf{v}^{(d_{|\mathcal{V}|})}\big) = \mathbf{W}^{(o)} \mathbf{h}^{(1,t)} = \sum_{d=1}^{D_h} \mathbf{h}_d^{(1,t)},$$

with $\mathbf{h}^{(1,t)} = \odot_{j \in \mathcal{N}(t)}(\mathbf{W}^{(1)} \mathbf{v}^{(d_j)}) = \odot_{j \in \mathcal{N}(t)}(\mathbf{I}\mathbf{v}^{(d_j)})$. Since $\mathbf{v}_{:D}^{(d_j)} = \mathbf{e}^{(d_j)}$ for all $j \in \mathcal{N}(t)$ and $\mathbf{I}$ is a zero-padded $D \times D$ identity matrix, it holds that:

$$f^{(\theta,\mathcal{G},t)}\big(\mathbf{v}^{(d_1)}, \ldots, \mathbf{v}^{(d_{|\mathcal{V}|})}\big) = \sum_{d=1}^{D} \prod_{j \in \mathcal{N}(t)} \mathbf{e}_d^{(d_j)}.$$

For every $d \in [D]$ we have that $\prod_{j \in \mathcal{N}(t)} \mathbf{e}_d^{(d_j)} = 1$ if $d_j = d$ for all $j \in \mathcal{N}(t)$ and $\prod_{j \in \mathcal{N}(t)} \mathbf{e}_d^{(d_j)} = 0$ otherwise. This implies that $f^{(\theta,\mathcal{G},t)}(\mathbf{v}^{(d_1)}, \ldots, \mathbf{v}^{(d_{|\mathcal{V}|})}) = 1$ if $d_j = d_{j'}$ for all $j, j' \in \mathcal{N}(t)$ and $f^{(\theta,\mathcal{G},t)}(\mathbf{v}^{(d_1)}, \ldots, \mathbf{v}^{(d_{|\mathcal{V}|})}) = 0$ otherwise, for all $d_1, \ldots, d_{|\mathcal{V}|} \in [D]$.

Equation (28) implies that $\big[\!\big[\boldsymbol{\mathcal{B}}\big(f^{(\theta,\mathcal{G},t)}\big); \mathcal{I}\big]\!\big]$ has a sub-matrix of rank $D$. Specifically, such a sub-matrix can be obtained by examining all rows and columns of $\big[\!\big[\boldsymbol{\mathcal{B}}\big(f^{(\theta,\mathcal{G},t)}\big); \mathcal{I}\big]\!\big]$ corresponding to some fixed indices $(d_i \in [D])_{i \in \mathcal{V} \setminus \mathcal{N}(t)}$ for the vertices that are not neighbors of $t$. Thus, $\text{rank}\big[\!\big[\boldsymbol{\mathcal{B}}\big(f^{(\theta,\mathcal{G},t)}\big); \mathcal{I}\big]\!\big] \geq D$. Notice that necessarily $\rho_0(\mathcal{C}_t^*, \{t\}) = 1$, as it is not zero and there can only be one length zero walk to $t$ (the trivial walk that starts and ends at $t$). Recalling that $\alpha_{\mathcal{C}_t^*,t} := D$ for $L = 1$, we therefore conclude:

$$\log\Big(\text{rank}\big[\!\big[\boldsymbol{\mathcal{B}}\big(f^{(\theta,\mathcal{G},t)}\big); \mathcal{I}\big]\!\big]\Big) \geq \log(D) = \log(\alpha_{\mathcal{C}_t^*,t}) \cdot \rho_0(\mathcal{C}_t^*, \{t\}).$$

**Case of $L \geq 2$:** Let $M := \left(\!\!\left(\begin{smallmatrix} D \\ \rho_{L-1}(\mathcal{C}_t^*, \{t\}) \end{smallmatrix}\right)\!\!\right) = \binom{D + \rho_{L-1}(\mathcal{C}_t^*, \{t\}) - 1}{\rho_{L-1}(\mathcal{C}_t^*, \{t\})}$ be the multiset coefficient of $D$ and $\rho_{L-1}(\mathcal{C}_t^*, \{t\})$ (recall $D := \min\{D_x, D_h\}$). By Lemma 7, there exists $\mathbf{Z} \in \mathbb{R}_{>0}^{M \times D}$ for which

$$\mathrm{rank}\left(\odot^{\rho_{L-1}(\mathcal{C}_t^*, \{t\})}(\mathbf{Z}\mathbf{Z}^\top)\right) = \left(\!\!\left(\begin{matrix} D \\ \rho_{L-1}(\mathcal{C}_t^*, \{t\}) \end{matrix}\right)\!\!\right),$$

with $\odot^{\rho_{L-1}(\mathcal{C}_t^*, \{t\})}(\mathbf{Z}\mathbf{Z}^\top)$ standing for the $\rho_{L-1}(\mathcal{C}_t^*, \{t\})$'th Hadamard power of $\mathbf{Z}\mathbf{Z}^\top$. For this $\mathbf{Z}$, by Lemma 5 below we know that there exist weights $\theta$ and template vectors such that $[\![\mathcal{B}(f^{(\theta, \mathcal{G}, t)}); \mathcal{I}]\!]$ has an $M \times M$ sub-matrix of the form $\mathbf{S}(\odot^{\rho_{L-1}(\mathcal{C}_t^*, \{t\})}(\mathbf{Z}\mathbf{Z}^\top))\mathbf{Q}$, where $\mathbf{S}, \mathbf{Q} \in \mathbb{R}^{M \times M}$ are full-rank diagonal matrices. Since the rank of a matrix is at least the rank of any of its sub-matrices:

$$\mathrm{rank}\left([\![\mathcal{B}(f^{(\theta, \mathcal{G}, t)}); \mathcal{I}]\!]\right) \geq \mathrm{rank}\left(\mathbf{S}\left(\odot^{\rho_{L-1}(\mathcal{C}_t^*, \{t\})}(\mathbf{Z}\mathbf{Z}^\top)\right)\mathbf{Q}\right)$$
$$= \mathrm{rank}\left(\odot^{\rho_{L-1}(\mathcal{C}_t^*, \{t\})}(\mathbf{Z}\mathbf{Z}^\top)\right)$$
$$= \left(\!\!\left(\begin{matrix} D \\ \rho_{L-1}(\mathcal{C}_t^*, \{t\}) \end{matrix}\right)\!\!\right),$$

where the second transition is due to $\mathbf{S}$ and $\mathbf{Q}$ being full-rank. Applying Lemma 8 to lower bound the multiset coefficient, we have that:

$$\mathrm{rank}\left([\![\mathcal{B}(f^{(\theta, \mathcal{G}, t)}); \mathcal{I}]\!]\right) \geq \left(\!\!\left(\begin{matrix} D \\ \rho_{L-1}(\mathcal{C}_t^*, \{t\}) \end{matrix}\right)\!\!\right) \geq \left(\frac{D-1}{\rho_{L-1}(\mathcal{C}_t^*, \{t\})} + 1\right)^{\rho_{L-1}(\mathcal{C}_t^*, \{t\})}.$$

Taking the log of both sides while recalling that $\alpha_{\mathcal{C}_t^*, t} := (D-1) \cdot \rho_{L-1}(\mathcal{C}_t^*, \{t\})^{-1} + 1$, we conclude that:

$$\log(\mathrm{rank}[\![\mathcal{B}(f^{(\theta, \mathcal{G}, t)}); \mathcal{I}]\!]) \geq \log(\alpha_{\mathcal{C}_t^*, t}) \cdot \rho_{L-1}(\mathcal{C}_t^*, \{t\}).$$

**Lemma 5.** *Suppose that the GNN inducing $f^{(\theta, \mathcal{G}, t)}$ is of depth $L \geq 2$ and that $\rho_{L-1}(\mathcal{C}_t^*, \{t\}) > 0$. For any $M \in \mathbb{N}$ and matrix with positive entries $\mathbf{Z} \in \mathbb{R}_{>0}^{M \times D}$, there exist weights $\theta$ and $M + 1$ template vectors $\mathbf{v}^{(1)}, \ldots, \mathbf{v}^{(M+1)} \in \mathbb{R}^{D_x}$ such that $[\![\mathcal{B}(f^{(\theta, \mathcal{G}, t)}); \mathcal{I}]\!]$ has an $M \times M$ sub-matrix $\mathbf{S}(\odot^{\rho_{L-1}(\mathcal{C}_t^*, \{t\})}(\mathbf{Z}\mathbf{Z}^\top))\mathbf{Q}$, where $\mathbf{S}, \mathbf{Q} \in \mathbb{R}^{M \times M}$ are full-rank diagonal matrices and $\odot^{\rho_{L-1}(\mathcal{C}_t^*, \{t\})}(\mathbf{Z}\mathbf{Z}^\top)$ is the $\rho_{L-1}(\mathcal{C}_t^*, \{t\})$'th Hadamard power of $\mathbf{Z}\mathbf{Z}^\top$.*

*Proof.* Consider the weights $\theta = (\mathbf{W}^{(1)}, \ldots, \mathbf{W}^{(L)}, \mathbf{W}^{(o)})$ defined by:

$$\mathbf{W}^{(1)} := \mathbf{I} \in \mathbb{R}^{D_h \times D_x},$$

$$\mathbf{W}^{(2)} := \begin{pmatrix} 1 & 1 & \cdots & 1 \\ 0 & 0 & \cdots & 0 \\ \vdots & \vdots & \cdots & \vdots \\ 0 & 0 & \cdots & 0 \end{pmatrix} \in \mathbb{R}^{D_h \times D_h},$$

$$\forall l \in \{3, \ldots, L\}: \ \mathbf{W}^{(l)} := \begin{pmatrix} 1 & 0 & \cdots & 0 \\ 0 & 0 & \cdots & 0 \\ \vdots & \vdots & \cdots & \vdots \\ 0 & 0 & \cdots & 0 \end{pmatrix} \in \mathbb{R}^{D_h \times D_h},$$

$$\mathbf{W}^{(o)} := \begin{pmatrix} 1 & 0 & \cdots & 0 \end{pmatrix} \in \mathbb{R}^{1 \times D_h},$$

where $\mathbf{I}$ is a zero padded identity matrix, *i.e.* it holds ones on its diagonal and zeros elsewhere. We let the templates $\mathbf{v}^{(1)}, \ldots, \mathbf{v}^{(M)} \in \mathbb{R}^{D_x}$ be the vectors holding the respective rows of $\mathbf{Z}$ in their first $D$ coordinates and zeros in the remaining entries (recall $D := \min\{D_x, D_h\}$). That is, denoting the rows of $\mathbf{Z}$ by $\mathbf{z}^{(1)}, \ldots, \mathbf{z}^{(M)} \in \mathbb{R}_{>0}^D$, we let $\mathbf{v}_{:D}^{(m)} := \mathbf{z}^{(m)}$ and $\mathbf{v}_{D+1:}^{(m)} := 0$ for all $m \in [M]$. We set all entries of the last template vector to one, *i.e.* $\mathbf{v}^{(M+1)} := (1, \ldots, 1) \in \mathbb{R}^{D_x}$.

Since $\mathcal{C}_t^* \in \mathcal{S}(\mathcal{I})$, *i.e.* it is an admissible subset of $\mathcal{C}_\mathcal{I}$ (Definition 4), there exist $\mathcal{I}' \subseteq \mathcal{I}, \mathcal{J}' \subseteq \mathcal{I}^c$ with no repeating shared neighbors (Definition 3) such that $\mathcal{C}_t^* = \mathcal{N}(\mathcal{I}') \cap \mathcal{N}(\mathcal{J}')$. Notice that $\mathcal{I}'$ and $\mathcal{J}'$ are non-empty as $\mathcal{C}_t^* \neq \emptyset$ (this is implied by $\rho_{L-1}(\mathcal{C}_t^*, \{t\}) > 0$). We focus on the $M \times M$

sub-matrix of $\big[\!\big[\mathcal{B}(f^{(\theta,\mathcal{G},t)}); \mathcal{I}\big]\!\big]$ that includes only rows and columns corresponding to evaluations of $f^{(\theta,\mathcal{G},t)}$ where all variables indexed by $\mathcal{I}'$ are assigned the same template vector from $\mathbf{v}^{(1)}, \ldots, \mathbf{v}^{(M)}$, all variables indexed by $\mathcal{J}'$ are assigned the same template vector from $\mathbf{v}^{(1)}, \ldots, \mathbf{v}^{(M)}$, and all remaining variables are assigned the all-ones template vector $\mathbf{v}^{(M+1)}$. Denoting this sub-matrix by $\mathbf{U} \in \mathbb{R}^{M \times M}$, it therefore upholds:

$$\mathbf{U}_{m,n} = f^{(\theta,\mathcal{G},t)}\Big(\big(\mathbf{x}^{(i)} \leftarrow \mathbf{v}^{(m)}\big)_{i \in \mathcal{I}'}, \big(\mathbf{x}^{(j)} \leftarrow \mathbf{v}^{(n)}\big)_{j \in \mathcal{J}'}, \big(\mathbf{x}^{(k)} \leftarrow \mathbf{v}^{(M+1)}\big)_{k \in \mathcal{V} \setminus (\mathcal{I}' \cup \mathcal{J}')}\Big),$$

for all $m, n \in [M]$, where we use $(\mathbf{x}^{(i)} \leftarrow \mathbf{v}^{(m)})_{i \in \mathcal{I}'}$ to denote that input variables indexed by $\mathcal{I}'$ are assigned the value $\mathbf{v}^{(m)}$. To show that $\mathbf{U}$ obeys the form $\mathbf{S}(\odot^{\rho_{L-1}(\mathcal{C}_t^*, \{t\})}(\mathbf{Z}\mathbf{Z}^\top))\mathbf{Q}$ for full-rank diagonal $\mathbf{S}, \mathbf{Q} \in \mathbb{R}^{M \times M}$, we prove there exist $\phi, \psi : \mathbb{R}^{D_x} \to \mathbb{R}_{>0}$ such that $\mathbf{U}_{m,n} = \phi(\mathbf{v}^{(m)})\langle \mathbf{z}^{(m)}, \mathbf{z}^{(n)} \rangle^{\rho_{L-1}(\mathcal{C}_t^*, \{t\})}\psi(\mathbf{v}^{(n)})$ for all $m, n \in [M]$. Indeed, defining $\mathbf{S}$ to hold $\phi(\mathbf{v}^{(1)}), \ldots, \phi(\mathbf{v}^{(M)})$ on its diagonal and $\mathbf{Q}$ to hold $\psi(\mathbf{v}^{(1)}), \ldots, \psi(\mathbf{v}^{(M)})$ on its diagonal, we have that $\mathbf{U} = \mathbf{S}(\odot^{\rho_{L-1}(\mathcal{C}_t^*, \{t\})}(\mathbf{Z}\mathbf{Z}^\top))\mathbf{Q}$. Since $\mathbf{S}$ and $\mathbf{Q}$ are clearly full-rank (diagonal matrices with non-zero entries on their diagonal), the proof concludes.

For $m, n \in [M]$, let $\mathbf{h}^{(l,i)} \in \mathbb{R}^{D_h}$ be the hidden embedding for $i \in \mathcal{V}$ at layer $l \in [L]$ of the GNN inducing $f^{(\theta,\mathcal{G},t)}$, over the following assignment to its input variables (*i.e.* vertex features):

$$\big(\mathbf{x}^{(i)} \leftarrow \mathbf{v}^{(m)}\big)_{i \in \mathcal{I}'}, \big(\mathbf{x}^{(j)} \leftarrow \mathbf{v}^{(n)}\big)_{j \in \mathcal{J}'}, \big(\mathbf{x}^{(k)} \leftarrow \mathbf{v}^{(M+1)}\big)_{k \in \mathcal{V} \setminus (\mathcal{I}' \cup \mathcal{J}')}.$$

Invoking Lemma 10 with $\mathbf{v}^{(m)}, \mathbf{v}^{(n)}, \mathcal{I}'$, and $\mathcal{J}'$, it holds that:

$$\mathbf{h}_1^{(L,t)} = \phi^{(L,t)}\big(\mathbf{v}^{(m)}\big)\langle \mathbf{z}^{(m)}, \mathbf{z}^{(n)} \rangle^{\rho_{L-1}(\mathcal{C}_t^*, \{t\})}\psi^{(L,t)}\big(\mathbf{v}^{(n)}\big) \quad , \quad \forall d \in \{2, \ldots, D_h\} : \mathbf{h}_d^{(L,t)} = 0,$$

for some $\phi^{(L,t)}, \psi^{(L,t)} : \mathbb{R}^{D_x} \to \mathbb{R}_{>0}$. Since

$$\begin{aligned}
\mathbf{U}_{m,n} &= f^{(\theta,\mathcal{G},t)}\Big(\big(\mathbf{x}^{(i)} \leftarrow \mathbf{v}^{(m)}\big)_{i \in \mathcal{I}'}, \big(\mathbf{x}^{(j)} \leftarrow \mathbf{v}^{(n)}\big)_{j \in \mathcal{J}'}, \big(\mathbf{x}^{(k)} \leftarrow \mathbf{v}^{(M+1)}\big)_{k \in \mathcal{V} \setminus (\mathcal{I}' \cup \mathcal{J}')}\Big) \\
&= \mathbf{W}^{(o)}\mathbf{h}^{(L,t)}
\end{aligned}$$

and $\mathbf{W}^{(o)} = (1, 0, \ldots, 0)$, this implies that:

$$\mathbf{U}_{m,n} = \mathbf{h}_1^{(L,t)} = \phi^{(L,t)}\big(\mathbf{v}^{(m)}\big)\langle \mathbf{z}^{(m)}, \mathbf{z}^{(n)} \rangle^{\rho_{L-1}(\mathcal{C}_t^*, \{t\})}\psi^{(L,t)}\big(\mathbf{v}^{(n)}\big).$$

Defining $\phi := \phi^{(L,t)}$ and $\psi := \psi^{(L,t)}$ leads to the sought-after expression for $\mathbf{U}_{m,n}$:

$$\mathbf{U}_{m,n} = \phi\big(\mathbf{v}^{(m)}\big)\langle \mathbf{z}^{(m)}, \mathbf{z}^{(n)} \rangle^{\rho_{L-1}(\mathcal{C}_t^*, \{t\})}\psi\big(\mathbf{v}^{(n)}\big).$$

$\square$

### I.3.3 Technical Lemmas

For completeness, we include the *vector rearrangement inequality* from [61], which we employ for proving the subsequent Lemma 7.

**Lemma 6** (Lemma 1 from [61]). *Let $\mathbf{a}^{(1)}, \ldots, \mathbf{a}^{(M)} \in \mathbb{R}_{\geq 0}^D$ be $M \in \mathbb{N}$ different vectors with non-negative entries. Then, for any permutation $\sigma : [M] \to [M]$ besides the identity permutation it holds that:*

$$\sum\nolimits_{m=1}^M \Big\langle \mathbf{a}^{(m)}, \mathbf{a}^{(\sigma(m))} \Big\rangle < \sum\nolimits_{m=1}^M \big\|\mathbf{a}^{(m)}\big\|^2.$$

Taking the $P$'th Hadamard power of a rank at most $D$ matrix results in a matrix whose rank is at most the multiset coefficient $\left(\!\!\binom{D}{P}\!\!\right) := \binom{D+P-1}{P}$ (see, *e.g.*, Theorem 1 in [2]). Lemma 7, adapted from Appendix B.2 in [64], guarantees that we can always find a $\left(\!\!\binom{D}{P}\!\!\right) \times D$ matrix $\mathbf{Z}$ with positive entries such that $\mathrm{rank}(\odot^P(\mathbf{Z}\mathbf{Z}^\top))$ is maximal, *i.e.* equal to $\left(\!\!\binom{D}{P}\!\!\right)$.

**Lemma 7** (adapted from Appendix B.2 in [64]). *For any $D \in \mathbb{N}$ and $P \in \mathbb{N}_{\geq 0}$, there exists a matrix with positive entries $\mathbf{Z} \in \mathbb{R}_{>0}^{\left(\!\!\binom{D}{P}\!\!\right) \times D}$ for which:*

$$\mathrm{rank}\big(\odot^P(\mathbf{Z}\mathbf{Z}^\top)\big) = \left(\!\!\binom{D}{P}\!\!\right),$$

*where $\odot^P(\mathbf{Z}\mathbf{Z}^\top)$ is the P'th Hadamard power of $\mathbf{Z}\mathbf{Z}^\top$.*

*Proof.* We let $M := \left(\!\!\binom{D}{P}\!\!\right)$ for notational convenience. Denote by $\mathbf{z}^{(1)}, \ldots, \mathbf{z}^{(M)} \in \mathbb{R}^D$ the row vectors of $\mathbf{Z} \in \mathbb{R}_{>0}^{M \times D}$. Observing the $(m, n)$'th entry of $\odot^P(\mathbf{Z}\mathbf{Z}^\top)$:

$$\left[\odot^P(\mathbf{Z}\mathbf{Z}^\top)\right]_{m,n} = \left\langle \mathbf{z}^{(m)}, \mathbf{z}^{(n)} \right\rangle^P = \left(\sum\nolimits_{d=1}^D \mathbf{z}_d^{(m)} \cdot \mathbf{z}_d^{(n)}\right)^P,$$

by expanding the power using the multinomial identity we have that:

$$
\begin{aligned}
\left[\odot^P(\mathbf{Z}\mathbf{Z}^\top)\right]_{m,n} &= \sum_{\substack{q_1,\ldots,q_D \in \mathbb{N}_{\geq 0} \\ \text{s.t. } \sum_{d=1}^D q_d = P}} \binom{P}{q_1, \ldots, q_D} \prod_{d=1}^D \left(\mathbf{z}_d^{(m)} \cdot \mathbf{z}_d^{(n)}\right)^{q_d} \\
&= \sum_{\substack{q_1,\ldots,q_D \in \mathbb{N}_{\geq 0} \\ \text{s.t. } \sum_{d=1}^D q_d = P}} \binom{P}{q_1, \ldots, q_D} \left(\prod_{d=1}^D \left(\mathbf{z}_d^{(m)}\right)^{q_d}\right) \cdot \left(\prod_{d=1}^D \left(\mathbf{z}_d^{(n)}\right)^{q_d}\right),
\end{aligned}
\tag{29}
$$

where in the last equality we separated terms depending on $m$ from those depending on $n$.

Let $\left(\mathbf{a}^{(q_1,\ldots,q_D)} \in \mathbb{R}^M\right)_{q_1,\ldots,q_D \in \mathbb{N}_{\geq 0} \text{ s.t } \sum_{d=1}^D q_d = P}$ be $M$ vectors defined by $\mathbf{a}_m^{(q_1,\ldots,q_D)} = \prod_{d=1}^D \left(\mathbf{z}_d^{(m)}\right)^{q_d}$ for all $q_1, \ldots, q_D \in \mathbb{N}_{\geq 0}$ satisfying $\sum_{d=1}^D q_d = P$ and $m \in [M]$. As can be seen from Equation (29), we can write:

$$\odot^P(\mathbf{Z}\mathbf{Z}^\top) = \mathbf{A}\mathbf{S}\mathbf{A}^\top,$$

where $\mathbf{A} \in \mathbb{R}^{M \times M}$ is the matrix whose columns are $\left(\mathbf{a}^{(q_1,\ldots,q_D)}\right)_{q_1,\ldots,q_D \in \mathbb{N}_{\geq 0} \text{ s.t } \sum_{d=1}^D q_d = P}$ and $\mathbf{S} \in \mathbb{R}^{M \times M}$ is the diagonal matrix holding $\binom{P}{q_1,\ldots,q_D}$ for every $q_1, \ldots, q_D \in \mathbb{N}_{\geq 0}$ satisfying $\sum_{d=1}^D q_d = P$ on its diagonal. Since all entries on the diagonal of $\mathbf{S}$ are positive, it is of full-rank, *i.e.* $\mathrm{rank}(\mathbf{S}) = M$. Thus, to prove that there exists $\mathbf{Z} \in \mathbb{R}_{>0}^{M \times D}$ for which $\mathrm{rank}(\odot^P(\mathbf{Z}\mathbf{Z}^\top)) = M$, it suffices to show that we can choose $\mathbf{z}^{(1)}, \ldots, \mathbf{z}^{(M)}$ with positive entries inducing $\mathrm{rank}(\mathbf{A}) = M$, for $\mathbf{A}$ as defined above. Below, we complete the proof by constructing such $\mathbf{z}^{(1)}, \ldots, \mathbf{z}^{(M)}$.

We associate each of $\mathbf{z}^{(1)}, \ldots, \mathbf{z}^{(M)}$ with a different configuration from the set:

$$\left\{ \mathbf{q} = (q_1, \ldots, q_D) : q_1, \ldots, q_D \in \mathbb{N}_{\geq 0} , \ \sum\nolimits_{d=1}^D q_d = P \right\},$$

where note that this set contains $M = \left(\!\!\binom{D}{P}\!\!\right)$ elements. For $m \in [M]$, denote by $\mathbf{q}^{(m)}$ the configuration associated with $\mathbf{z}^{(m)}$. For a variable $\gamma \in \mathbb{R}$, to be determined later on, and every $m \in [M]$ and $d \in [D]$, we set:

$$\mathbf{z}_d^{(m)} = \gamma^{\mathbf{q}_d^{(m)}}.$$

Given these $\mathbf{z}^{(1)}, \ldots, \mathbf{z}^{(M)}$, the entries of $\mathbf{A}$ have the following form:

$$\mathbf{A}_{m,n} = \prod\nolimits_{d=1}^D \left(\mathbf{z}_d^{(m)}\right)^{\mathbf{q}_d^{(n)}} = \prod\nolimits_{d=1}^D \left(\gamma^{\mathbf{q}_d^{(m)}}\right)^{\mathbf{q}_d^{(n)}} = \gamma^{\sum_{d=1}^D \mathbf{q}_d^{(m)} \cdot \mathbf{q}_d^{(n)}} = \gamma^{\left\langle \mathbf{q}^{(m)}, \mathbf{q}^{(n)} \right\rangle},$$

for all $m, n \in [M]$. Thus, $\det(\mathbf{A}) = \sum_{\text{permutation } \sigma:[M] \to [M]} \mathrm{sign}(\sigma) \cdot \gamma^{\sum_{m=1}^M \left\langle \mathbf{q}^{(m)}, \mathbf{q}^{(\sigma(m))} \right\rangle}$ is polynomial in $\gamma$. By Lemma 6, $\sum_{m=1}^M \left\langle \mathbf{q}^{(m)}, \mathbf{q}^{(\sigma(m))} \right\rangle < \sum_{m=1}^M \|\mathbf{q}^{(m)}\|^2$ for all $\sigma$ which is not the identity permutation. This implies that $\sum_{m=1}^M \|\mathbf{q}^{(m)}\|^2$ is the maximal degree of a monomial in $\det(\mathbf{A})$, and it is attained by a single element in $\sum_{\text{permutation } \sigma:[M] \to [M]} \mathrm{sign}(\sigma) \cdot \gamma^{\sum_{m=1}^M \left\langle \mathbf{q}^{(m)}, \mathbf{q}^{(\sigma(m))} \right\rangle}$ — that corresponding to the identity permutation. Consequently, $\det(\mathbf{A})$ cannot be the zero polynomial with respect to $\gamma$, and so it vanishes only on a finite set of values for $\gamma$. In particular, there exists $\gamma > 0$ such that $\det(\mathbf{A}) \neq 0$, meaning $\mathrm{rank}(\mathbf{A}) = M$. The proof concludes by noticing that for a positive $\gamma$ the entries of the chosen $\mathbf{z}^{(1)}, \ldots, \mathbf{z}^{(M)}$ are positive as well. $\qquad\square$

Additionally, we make use of the following lemmas.

**Lemma 8.** *For any $D, P \in \mathbb{N}$, let $\left(\!\binom{D}{P}\!\right) := \binom{D+P-1}{P}$ be the multiset coefficient. Then:*

$$\left(\!\binom{D}{P}\!\right) \geq \left(\frac{D-1}{P} + 1\right)^P.$$

*Proof.* For any $N \geq K \in \mathbb{N}$, a known lower bound on the binomial coefficient is $\binom{N}{K} \geq \left(\frac{N}{K}\right)^K$. Hence:

$$\left(\!\binom{D}{P}\!\right) = \binom{D+P-1}{P} \geq \left(\frac{D+P-1}{P}\right)^P = \left(\frac{D-1}{P} + 1\right)^P.$$

$\square$

**Lemma 9.** *For $D_1, D_2, K \in \mathbb{N}$, consider a polynomial function mapping variables $\theta \in \mathbb{R}^K$ to matrices $\mathbf{A}(\theta) \in \mathbb{R}^{D_1 \times D_2}$, i.e. the entries of $\mathbf{A}(\theta)$ are polynomial in $\theta$. If there exists a point $\theta^* \in \mathbb{R}^K$ such that $\mathrm{rank}(\mathbf{A}(\theta^*)) \geq R$, for $R \in [\min\{D_1, D_2\}]$, then the set $\{\theta \in \mathbb{R}^K : \mathrm{rank}(\mathbf{A}(\theta)) < R\}$ has Lebesgue measure zero.*

*Proof.* A matrix is of rank at least $R$ if and only if it has a $R \times R$ sub-matrix whose determinant is non-zero. The determinant of any sub-matrix of $\mathbf{A}(\theta)$ is polynomial in the entries of $\mathbf{A}(\theta)$, and so it is polynomial in $\theta$ as well. Since the zero set of a polynomial is either the entire space or a set of Lebesgue measure zero [19], the fact that $\mathrm{rank}(\mathbf{A}(\theta^*)) \geq R$ implies that $\{\theta \in \mathbb{R}^K : \mathrm{rank}(\mathbf{A}(\theta)) < R\}$ has Lebesgue measure zero. $\square$

**Lemma 10.** *Let $\mathbf{v}, \mathbf{v}' \in \mathbb{R}^{D_x}_{\geq 0}$ whose first $D := \min\{D_x, D_h\}$ entries are positive, and disjoint $\mathcal{I}', \mathcal{J}' \subseteq \mathcal{V}$ with no repeating shared neighbors (Definition 3). Denote by $\mathbf{h}^{(l,i)} \in \mathbb{R}^{D_h}$ the hidden embedding for $i \in \mathcal{V}$ at layer $l \in [L]$ of a GNN with depth $L \geq 2$ and product aggregation (Equations (2) and (5)), given the following assignment to its input variables (i.e. vertex features):*

$$\left(\mathbf{x}^{(i)} \leftarrow \mathbf{v}\right)_{i \in \mathcal{I}'}, \left(\mathbf{x}^{(j)} \leftarrow \mathbf{v}'\right)_{j \in \mathcal{J}'}, \left(\mathbf{x}^{(k)} \leftarrow \mathbf{1}\right)_{k \in \mathcal{V} \backslash (\mathcal{I}' \cup \mathcal{J}')},$$

*where $\mathbf{1} \in \mathbb{R}^{D_x}$ is the vector holding one in all entries. Suppose that the weights $\mathbf{W}^{(1)}, \ldots, \mathbf{W}^{(L)}$ of the GNN are given by:*

$$\mathbf{W}^{(1)} := \mathbf{I} \in \mathbb{R}^{D_h \times D_x},$$

$$\mathbf{W}^{(2)} := \begin{pmatrix} 1 & 1 & \cdots & 1 \\ 0 & 0 & \cdots & 0 \\ \vdots & \vdots & \cdots & \vdots \\ 0 & 0 & \cdots & 0 \end{pmatrix} \in \mathbb{R}^{D_h \times D_h},$$

$$\forall l \in \{3, \ldots, L\} : \mathbf{W}^{(l)} := \begin{pmatrix} 1 & 0 & \cdots & 0 \\ 0 & 0 & \cdots & 0 \\ \vdots & \vdots & \cdots & \vdots \\ 0 & 0 & \cdots & 0 \end{pmatrix} \in \mathbb{R}^{D_h \times D_h},$$

*where $\mathbf{I}$ is a zero padded identity matrix, i.e. it holds ones on its diagonal and zeros elsewhere. Then, for all $l \in \{2, \ldots, L\}$ and $i \in \mathcal{V}$, there exist $\phi^{(l,i)}, \psi^{(l,i)} : \mathbb{R}^{D_x} \to \mathbb{R}_{>0}$ such that:*

$$\mathbf{h}^{(l,i)}_1 = \phi^{(l,i)}(\mathbf{v}) \langle \mathbf{v}_{:D}, \mathbf{v}'_{:D} \rangle^{\rho_{l-1}(\mathcal{C}, \{i\})} \psi^{(l,i)}(\mathbf{v}') \quad, \quad \forall d \in \{2, \ldots, D_h\} : \mathbf{h}^{(l,i)}_d = 0,$$

*where $\mathcal{C} := \mathcal{N}(\mathcal{I}') \cap \mathcal{N}(\mathcal{J}')$.*

*Proof.* The proof is by induction over the layer $l \in \{2, \ldots, L\}$. For $l = 2$, fix $i \in \mathcal{V}$. By the update rule of a GNN with product aggregation:

$$\mathbf{h}^{(2,i)} = \odot_{j \in \mathcal{N}(i)} \left(\mathbf{W}^{(2)} \mathbf{h}^{(1,j)}\right).$$

Plugging in the value of $\mathbf{W}^{(2)}$ we get:

$$\mathbf{h}^{(2,i)}_1 = \prod_{j \in \mathcal{N}(i)} \left(\sum_{d=1}^{D_h} \mathbf{h}^{(1,j)}_d\right) \quad, \quad \forall d \in \{2, \ldots, D_h\} : \mathbf{h}^{(2,i)}_d = 0. \tag{30}$$

Let $\bar{\mathbf{v}}, \bar{\mathbf{v}}' \in \mathbb{R}^{D_h}$ be the vectors holding $\mathbf{v}_{:D}$ and $\mathbf{v}'_{:D}$ in their first $D$ coordinates and zero in the remaining entries, respectively. Similarly, we use $\bar{\mathbf{1}} \in \mathbb{R}^{D_h}$ to denote the vector whose first $D$ entries are one and the remaining are zero. Examining $\mathbf{h}^{(1,j)}$ for $j \in \mathcal{N}(i)$, by the assignment of input variables and the fact that $\mathbf{W}^{(1)}$ is a zero padded identity matrix we have that:

$$\mathbf{h}^{(1,j)} = \odot_{k \in \mathcal{N}(j)}\big(\mathbf{W}^{(1)}\mathbf{x}^{(k)}\big) = \big(\odot^{|\mathcal{N}(j) \cap \mathcal{I}'|}\bar{\mathbf{v}}\big) \odot \big(\odot^{|\mathcal{N}(j) \cap \mathcal{J}'|}\bar{\mathbf{v}}'\big) \odot \big(\odot^{|\mathcal{N}(j) \setminus (\mathcal{I}' \cup \mathcal{J}')|}\bar{\mathbf{1}}\big)$$
$$= \big(\odot^{|\mathcal{N}(j) \cap \mathcal{I}'|}\bar{\mathbf{v}}\big) \odot \big(\odot^{|\mathcal{N}(j) \cap \mathcal{J}'|}\bar{\mathbf{v}}'\big)\,.$$

Since the first $D$ entries of $\bar{\mathbf{v}}$ and $\bar{\mathbf{v}}'$ are positive while the rest are zero, the same holds for $\mathbf{h}^{(1,j)}$. Additionally, recall that $\mathcal{I}'$ and $\mathcal{J}'$ have no repeating shared neighbors. Thus, if $j \in \mathcal{N}(\mathcal{I}') \cap \mathcal{N}(\mathcal{J}') = \mathcal{C}$, then $j$ has a single neighbor in $\mathcal{I}'$ and a single neighbor in $\mathcal{J}'$, implying $\mathbf{h}^{(1,j)} = \bar{\mathbf{v}} \odot \bar{\mathbf{v}}'$. Otherwise, if $j \notin \mathcal{C}$, then $\mathcal{N}(j) \cap \mathcal{I}' = \emptyset$ or $\mathcal{N}(j) \cap \mathcal{J}' = \emptyset$ must hold. In the former $\mathbf{h}^{(1,j)}$ does not depend on $\mathbf{v}$, whereas in the latter $\mathbf{h}^{(1,j)}$ does not depend on $\mathbf{v}'$.

Going back to Equation (30), while noticing that $|\mathcal{N}(i) \cap \mathcal{C}| = \rho_1(\mathcal{C}, \{i\})$, we arrive at:

$$\mathbf{h}_1^{(2,i)} = \prod_{j \in \mathcal{N}(i) \cap \mathcal{C}}\Big(\sum_{d=1}^{D_h} \mathbf{h}_d^{(1,j)}\Big) \cdot \prod_{j \in \mathcal{N}(i) \setminus \mathcal{C}}\Big(\sum_{d=1}^{D_h} \mathbf{h}_d^{(1,j)}\Big)$$
$$= \prod_{j \in \mathcal{N}(i) \cap \mathcal{C}}\Big(\sum_{d=1}^{D_h} [\bar{\mathbf{v}} \odot \bar{\mathbf{v}}']_d\Big) \cdot \prod_{j \in \mathcal{N}(i) \setminus \mathcal{C}}\Big(\sum_{d=1}^{D_h} \mathbf{h}_d^{(1,j)}\Big)$$
$$= \langle \mathbf{v}_{:D}, \mathbf{v}'_{:D}\rangle^{\rho_1(\mathcal{C}, \{i\})} \cdot \prod_{j \in \mathcal{N}(i) \setminus \mathcal{C}}\Big(\sum_{d=1}^{D_h} \mathbf{h}_d^{(1,j)}\Big)\,.$$

As discussed above, for each $j \in \mathcal{N}(i) \setminus \mathcal{C}$ the hidden embedding $\mathbf{h}^{(1,j)}$ does not depend on $\mathbf{v}$ or it does not depend on $\mathbf{v}'$. Furthermore, $\sum_{d=1}^{D_h} \mathbf{h}_d^{(1,j)} > 0$ for all $j \in \mathcal{N}(i)$. Hence, there exist $\phi^{(2,i)}, \psi^{(2,i)} : \mathbb{R}^{D_x} \to \mathbb{R}_{>0}$ such that:

$$\mathbf{h}_1^{(2,i)} = \phi^{(2,i)}(\mathbf{v}) \langle \mathbf{v}_{:D}, \mathbf{v}'_{:D}\rangle^{\rho_1(\mathcal{C}, \{i\})} \psi^{(2,i)}(\mathbf{v}')\,,$$

completing the base case.

Now, assuming that the inductive claim holds for $l - 1 \geq 2$, we prove that it holds for $l$. Let $i \in \mathcal{V}$. By the update rule of a GNN with product aggregation $\mathbf{h}^{(l,i)} = \odot_{j \in \mathcal{N}(i)}(\mathbf{W}^{(l)}\mathbf{h}^{(l-1,j)})$. Plugging in the value of $\mathbf{W}^{(l)}$ we get:

$$\mathbf{h}_1^{(l,i)} = \prod_{j \in \mathcal{N}(i)} \mathbf{h}_1^{(l-1,j)} \quad , \quad \forall d \in \{2, \ldots, D_h\} : \mathbf{h}_d^{(l,i)} = 0\,.$$

By the inductive assumption $\mathbf{h}_1^{(l-1,j)} = \phi^{(l-1,j)}(\mathbf{v}) \langle \mathbf{v}_{:D}, \mathbf{v}'_{:D}\rangle^{\rho_{l-2}(\mathcal{C}, \{j\})} \psi^{(l-1,j)}(\mathbf{v}')$ for all $j \in \mathcal{N}(i)$, where $\phi^{(l-1,j)}, \psi^{(l-1,j)} : \mathbb{R}^{D_x} \to \mathbb{R}_{>0}$. Thus:

$$\mathbf{h}_1^{(l,i)} = \prod_{j \in \mathcal{N}(i)} \mathbf{h}_1^{(l-1,j)}$$
$$= \prod_{j \in \mathcal{N}(i)} \phi^{(l-1,j)}(\mathbf{v}) \langle \mathbf{v}_{:D}, \mathbf{v}'_{:D}\rangle^{\rho_{l-2}(\mathcal{C}, \{j\})} \psi^{(l-1,j)}(\mathbf{v}')$$
$$= \Big(\prod_{j \in \mathcal{N}(i)} \phi^{(l-1,j)}(\mathbf{v})\Big) \cdot \langle \mathbf{v}_{:D}, \mathbf{v}'_{:D}\rangle^{\sum_{j \in \mathcal{N}(i)} \rho_{l-2}(\mathcal{C}, \{j\})} \cdot \Big(\prod_{j \in \mathcal{N}(i)} \psi^{(l-1,j)}(\mathbf{v}')\Big)\,.$$

Define $\phi^{(l,i)} : \mathbf{v} \mapsto \prod_{j \in \mathcal{N}(i)} \phi^{(l-1,j)}(\mathbf{v})$ and $\psi^{(l,i)} : \mathbf{v}' \mapsto \prod_{j \in \mathcal{N}(i)} \psi^{(l-1,j)}(\mathbf{v}')$. Since the range of $\phi^{(l-1,j)}$ and $\psi^{(l-1,j)}$ is $\mathbb{R}_{>0}$ for all $j \in \mathcal{N}(i)$, so is the range of $\phi^{(l,i)}$ and $\psi^{(l,i)}$. The desired result thus readily follows by noticing that $\sum_{j \in \mathcal{N}(i)} \rho_{l-2}(\mathcal{C}, \{j\}) = \rho_{l-1}(\mathcal{C}, \{i\})$:

$$\mathbf{h}_1^{(l,i)} = \phi^{(l,i)}(\mathbf{v}) \langle \mathbf{v}_{:D}, \mathbf{v}'_{:D}\rangle^{\rho_{l-1}(\mathcal{C}, \{i\})} \psi^{(l,i)}(\mathbf{v}')\,.$$

$\square$

## I.4 Proof of Theorem 4

The proof follows a line identical to that of Theorem 2 (Appendix I.2), requiring only slight adjustments. We outline the necessary changes.

Extending the tensor network representations of GNNs with product aggregation to directed graphs and multiple edge types is straightforward. Nodes, legs, and leg weights are as described in Appendix E for undirected graphs with a single edge type, except that:

- Legs connecting $\delta$-tensors with weight matrices in the same layer are adapted such that only incoming neighbors are considered. Formally, in Equations (15) to (20), $\mathcal{N}(i)$ is replaced by $\mathcal{N}_{in}(i)$ in the leg definitions, for $i \in \mathcal{V}$.

- Weight matrices $(\mathbf{W}^{(l,q)})_{l \in [L], q \in [Q]}$ are assigned to nodes in accordance with edge types. Namely, if at layer $l \in [L]$ a $\delta$-tensor associated with $i \in \mathcal{V}$ is connected to a weight matrix associated with $j \in \mathcal{N}_{in}(i)$, then $\mathbf{W}^{(l,\tau(j,i))}$ is assigned to the weight matrix node, as opposed to $\mathbf{W}^{(l)}$ in the single edge type setting. Formally, let $\mathbf{W}^{(l,j,\gamma)}$ be a node at layer $l \in [L]$ connected to $\boldsymbol{\delta}^{(l,i,\gamma')}$, for $i \in \mathcal{V}, j \in \mathcal{N}_{in}(i)$, and some $\gamma, \gamma' \in \mathbb{N}$. Then, $\mathbf{W}^{(l,j,\gamma)}$ stands for a copy of $\mathbf{W}^{(l,\tau(j,i))}$.

For $\mathbf{X} = (\mathbf{x}^{(1)}, \ldots, \mathbf{x}^{(|\mathcal{V}|)}) \in \mathbb{R}^{D_x \times |\mathcal{V}|}$, let $\mathcal{T}(\mathbf{X})$ and $\mathcal{T}^{(t)}(\mathbf{X})$ be the tensor networks corresponding to $f^{(\theta,\mathcal{G})}(\mathbf{X})$ and $f^{(\theta,\mathcal{G},t)}(\mathbf{X})$, respectively, whose construction is outlined above. Then, Lemma 1 (from Appendix I.2.1) and its proof apply as stated. Meaning, $\operatorname{sep}(f^{(\theta,\mathcal{G})}; \mathcal{I})$ and $\operatorname{sep}(f^{(\theta,\mathcal{G},t)}; \mathcal{I})$ are upper bounded by the minimal modified multiplicative cut weights in $\mathcal{T}(\mathbf{X})$ and $\mathcal{T}^{(t)}(\mathbf{X})$, respectively, among cuts separating leaves associated with vertices of the input graph in $\mathcal{I}$ from leaves associated with vertices of the input graph in $\mathcal{I}^c$. Therefore, to establish Equations (11) and (12), it suffices to find cuts in the respective tensor networks with sufficiently low modified multiplicative weights. As is the case for undirected graphs with a single edge type (see Appendices I.2.2 and I.2.3), the cuts separating nodes corresponding to vertices in $\mathcal{I}$ from all other nodes yield the desired upper bounds. $\qquad\square$

## I.5   Proof of Theorem 5

The proof follows a line identical to that of Theorem 3 (Appendix I.3), requiring only slight adjustments. We outline the necessary changes.

In the context of graph prediction, let $\mathcal{C}^* \in \operatorname{argmax}_{\mathcal{C} \in \mathcal{S}^{\rightarrow}(\mathcal{I})} \log(\alpha_{\mathcal{C}}) \cdot \rho_{L-1}(\mathcal{C}, \mathcal{V})$. By Lemma 3 (from Appendix I.3), to prove that Equation (13) holds for weights $\theta$, it suffices to find template vectors for which $\log(\operatorname{rank}[\![\boldsymbol{\mathcal{B}}(f^{(\theta,\mathcal{G})}); \mathcal{I}]\!]) \geq \log(\alpha_{\mathcal{C}^*}) \cdot \rho_{L-1}(\mathcal{C}^*, \mathcal{V})$. Notice that, since the outputs of $f^{(\theta,\mathcal{G})}$ vary polynomially with $\theta$, so do the entries of $[\![\boldsymbol{\mathcal{B}}(f^{(\theta,\mathcal{G})}); \mathcal{I}]\!]$ for any choice of template vectors. Thus, according to Lemma 9 (from Appendix I.3.3), by constructing weights $\theta$ and template vectors satisfying $\log(\operatorname{rank}[\![\boldsymbol{\mathcal{B}}(f^{(\theta,\mathcal{G})}); \mathcal{I}]\!]) \geq \log(\alpha_{\mathcal{C}^*}) \cdot \rho_{L-1}(\mathcal{C}^*, \mathcal{V})$, we may conclude that this is the case for almost all assignments of weights, meaning Equation (13) holds for almost all assignments of weights. For undirected graphs with a single edge type, Appendix I.3.1 provides such weights $\mathbf{W}^{(1)}, \ldots, \mathbf{W}^{(L)}, \mathbf{W}^{(o)}$ and template vectors. The proof in the case of directed graphs with multiple edge types is analogous, requiring only a couple adaptations: *(i)* weight matrices of all edge types at layer $l \in [L]$ are set to the $\mathbf{W}^{(l)}$ chosen in Appendix I.3.1; and *(ii)* $\mathcal{C}_{\mathcal{I}}$ and $\mathcal{S}(\mathcal{I})$ are replaced with their directed counterparts $\mathcal{C}_{\mathcal{I}}^{\rightarrow}$ and $\mathcal{S}^{\rightarrow}(\mathcal{I})$, respectively.

In the context of vertex prediction, let $\mathcal{C}_t^* \in \operatorname{argmax}_{\mathcal{C} \in \mathcal{S}^{\rightarrow}(\mathcal{I})} \log(\alpha_{\mathcal{C},t}) \cdot \rho_{L-1}(\mathcal{C}, \{t\})$. Due to arguments similar to those above, to prove that Equation (14) holds for almost all assignments of weights, we need only find weights $\theta$ and template vectors satisfying $\log(\operatorname{rank}[\![\boldsymbol{\mathcal{B}}(f^{(\theta,\mathcal{G},t)}); \mathcal{I}]\!]) \geq \log(\alpha_{\mathcal{C}_t^*, t}) \cdot \rho_{L-1}(\mathcal{C}_t^*, \{t\})$. For undirected graphs with a single edge type, Appendix I.3.2 provides such weights and template vectors. The adaptations necessary to extend Appendix I.3.2 to directed graphs with multiple edge types are identical to those specified above for extending Appendix I.3.1 in the context of graph prediction.

Lastly, recalling that a finite union of measure zero sets has measure zero as well establishes that Equations (13) and (14) jointly hold for almost all assignments of weights. $\qquad\square$

## I.6   Proof of Proposition 1

We first prove that the contractions described by $\mathcal{T}(\mathbf{X})$ produce $f^{(\theta,\mathcal{G})}(\mathbf{X})$. Through an induction over the layer $l \in [L]$, for all $i \in \mathcal{V}$ and $\gamma \in [\rho_{L-l}(\{i\}, \mathcal{V})]$ we show that contracting the sub-tree whose root is $\boldsymbol{\delta}^{(l,i,\gamma)}$ yields $\mathbf{h}^{(l,i)}$ — the hidden embedding for $i$ at layer $l$ of the GNN inducing $f^{(\theta,\mathcal{G})}$, given vertex features $\mathbf{x}^{(1)}, \ldots, \mathbf{x}^{(|\mathcal{V}|)}$.

For $l = 1$, fix some $i \in \mathcal{V}$ and $\gamma \in [\rho_{L-1}(\{i\}, \mathcal{V})]$. The sub-tree whose root is $\boldsymbol{\delta}^{(1,i,\gamma)}$ comprises $|\mathcal{N}(i)|$ copies of $\mathbf{W}^{(1)}$, each associated with some $j \in \mathcal{N}(i)$ and contracted in its second mode with a copy of $\mathbf{x}^{(j)}$. Additionally, $\boldsymbol{\delta}^{(1,i,\gamma)}$, which is a copy of $\boldsymbol{\delta}^{(|\mathcal{N}(i)|+1)}$, is contracted with the copies of $\mathbf{W}^{(1)}$ in their first mode. Overall, the execution of all contractions in the sub-tree can be written as $\boldsymbol{\delta}^{(|\mathcal{N}(i)|+1)} *_{j \in [|\mathcal{N}(i)|]} (\mathbf{W}^{(1)} \mathbf{x}^{(\mathcal{N}(i)_j)})$, where $\mathcal{N}(i)_j$, for $j \in [|\mathcal{N}(i)|]$, denotes the $j$'th neighbor of $i$ according to an ascending order (recall vertices are represented by indices from 1 to $|\mathcal{V}|$). The base case concludes by Lemma 11:

$$\boldsymbol{\delta}^{(|\mathcal{N}(i)|+1)} *_{j \in [|\mathcal{N}(i)|]} \left( \mathbf{W}^{(1)} \mathbf{x}^{(\mathcal{N}(i)_j)} \right) = \odot_{j \in [|\mathcal{N}(i)|]} \left( \mathbf{W}^{(1)} \mathbf{x}^{(\mathcal{N}(i)_j)} \right) = \mathbf{h}^{(1,i)} \,.$$

Assuming that the inductive claim holds for $l - 1 \geq 1$, we prove that it holds for $l$. Let $i \in \mathcal{V}$ and $\gamma \in [\rho_{L-l}(\{i\}, \mathcal{V})]$. The children of $\boldsymbol{\delta}^{(l,i,\gamma)}$ in the tensor network are of the form $\mathbf{W}^{(l,\mathcal{N}(i)_j,\phi_{l,i,j}(\gamma))}$, for $j \in [|\mathcal{N}(i)|]$, and each $\mathbf{W}^{(l,\mathcal{N}(i)_j,\phi_{l,i,j}(\gamma))}$ is connected in its other mode to $\boldsymbol{\delta}^{(l-1,\mathcal{N}(i)_j,\phi_{l,i,j}(\gamma))}$. By the inductive assumption for $l-1$, we know that performing all contractions in the sub-tree whose root is $\boldsymbol{\delta}^{(l-1,\mathcal{N}(i)_j,\phi_{l,i,j}(\gamma))}$ produces $\mathbf{h}^{(l-1,\mathcal{N}(i)_j)}$, for all $j \in [|\mathcal{N}(i)|]$. Since $\boldsymbol{\delta}^{(l,i,\gamma)}$ is a copy of $\boldsymbol{\delta}^{(|\mathcal{N}(i)|+1)}$, and each $\mathbf{W}^{(l,\mathcal{N}(i)_j,\phi_{l,i,j}(\gamma))}$ is a copy of $\mathbf{W}^{(l)}$, the remaining contractions in the sub-tree of $\boldsymbol{\delta}^{(l,i,\gamma)}$ thus give:

$$\boldsymbol{\delta}^{(|\mathcal{N}(i)|+1)} *_{j \in [|\mathcal{N}(i)|]} \left( \mathbf{W}^{(l)} \mathbf{h}^{(l-1,\mathcal{N}(i)_j)} \right),$$

which according to Lemma 11 amounts to:

$$\boldsymbol{\delta}^{(|\mathcal{N}(i)|+1)} *_{j \in [|\mathcal{N}(i)|]} \left( \mathbf{W}^{(l)} \mathbf{h}^{(l-1,\mathcal{N}(i)_j)} \right) = \odot_{j \in [|\mathcal{N}(i)|]} \left( \mathbf{W}^{(l)} \mathbf{h}^{(l-1,\mathcal{N}(i)_j)} \right) = \mathbf{h}^{(l,i)} \,,$$

establishing the induction step.

With the inductive claim at hand, we show that contracting $\mathcal{T}(\mathbf{X})$ produces $f^{(\theta,\mathcal{G})}(\mathbf{X})$. Applying the inductive claim for $l = L$, we have that $\mathbf{h}^{(L,1)}, \ldots, \mathbf{h}^{(L,|\mathcal{V}|)}$ are the vectors produced by executing all contractions in the sub-trees whose roots are $\boldsymbol{\delta}^{(L,1,1)}, \ldots, \boldsymbol{\delta}^{(L,|\mathcal{V}|,1)}$, respectively. Performing the remaining contractions, defined by the legs of $\boldsymbol{\delta}^{(|\mathcal{V}|+1)}$, therefore yields $\mathbf{W}^{(o)} \big( \boldsymbol{\delta}^{(|\mathcal{V}|+1)} *_{i \in [|\mathcal{V}|]} \mathbf{h}^{(L,i)} \big)$. By Lemma 11:

$$\boldsymbol{\delta}^{(|\mathcal{V}|+1)} *_{i \in [|\mathcal{V}|]} \mathbf{h}^{(L,i)} = \odot_{i \in [|\mathcal{V}|]} \mathbf{h}^{(L,i)} \,.$$

Hence, $\mathbf{W}^{(o)} \big( \boldsymbol{\delta}^{(|\mathcal{V}|+1)} *_{i \in [|\mathcal{V}|]} \mathbf{h}^{(L,i)} \big) = \mathbf{W}^{(o)} \big( \odot_{i \in [|\mathcal{V}|]} \mathbf{h}^{(L,i)} \big) = f^{(\theta,\mathcal{G})}(\mathbf{X})$, meaning contracting $\mathcal{T}(\mathbf{X})$ results in $f^{(\theta,\mathcal{G})}(\mathbf{X})$.

An analogous proof establishes that the contractions described by $\mathcal{T}^{(t)}(\mathbf{X})$ yield $f^{(\theta,\mathcal{G},t)}(\mathbf{X})$. Specifically, the inductive claim and its proof are the same, up to $\gamma$ taking values in $[\rho_{L-l}(\{i\}, \{t\})]$ instead of $[\rho_{L-l}(\{i\}, \mathcal{V})]$, for $l \in [L]$. This implies that $\mathbf{h}^{(L,t)}$ is the vector produced by contracting the sub-tree whose root is $\boldsymbol{\delta}^{(L,t,1)}$. Performing the only remaining contraction, defined by the leg connecting $\boldsymbol{\delta}^{(L,t,1)}$ with $\mathbf{W}^{(o)}$, thus results in $\mathbf{W}^{(o)} \mathbf{h}^{(L,t)} = f^{(\theta,\mathcal{G},t)}(\mathbf{X})$. $\qquad\square$

### I.6.1 Technical Lemma

**Lemma 11.** *Let $\boldsymbol{\delta}^{(N+1)} \in \mathbb{R}^{D \times \cdots \times D}$ be an order $N + 1 \in \mathbb{N}$ tensor that has ones on its hyperdiagonal and zeros elsewhere, i.e. $\delta^{(N+1)}_{d_1,\ldots,d_{N+1}} = 1$ if $d_1 = \cdots = d_{N+1}$ and $\delta^{(N+1)}_{d_1,\ldots,d_{N+1}} = 0$ otherwise, for all $d_1, \ldots, d_{N+1} \in [D]$. Then, for any $\mathbf{x}^{(1)}, \ldots, \mathbf{x}^{(N)} \in \mathbb{R}^D$ it holds that $\boldsymbol{\delta}^{(N+1)} *_{i \in [N]} \mathbf{x}^{(i)} = \odot_{i \in [N]} \mathbf{x}^{(i)} \in \mathbb{R}^D$.*

*Proof.* By the definition of tensor contraction (Definition 7), for all $d \in [D]$ we have that:

$$\left( \boldsymbol{\delta}^{(N+1)} *_{i \in [N]} \mathbf{x}^{(i)} \right)_d = \sum_{d_1,\ldots,d_N=1}^{D} \delta^{(N+1)}_{d_1,\ldots,d_N,d} \cdot \prod_{i \in [N]} \mathbf{x}^{(i)}_{d_i} = \prod_{i \in [N]} \mathbf{x}^{(i)}_d = \left( \odot_{i \in [N]} \mathbf{x}^{(i)} \right)_d \,.$$

$\square$

