# OpenReview forum: "On the Ability of Graph Neural Networks to Model Interactions Between Vertices"
_NeurIPS.cc/2023/Conference — NeurIPS 2023 poster_

### Official Review · Reviewer_uCaB · 2023-07-04

**Soundness:** 3 good
**Presentation:** 3 good
**Contribution:** 2 fair
**Rating:** 6
**Confidence:** 3

**Summary:**

The paper studies how to characterize the ability of Graph Neural Networks (GNNs) to model the interaction given a partition of vertices. They quanlify the interaction strength with separation rank, and prove that it is governed by walk index (number of walks starting from the partition boundary). This relation is empirically demonstrated in their experiments, where a GNN with higher walk index beats the one with lower Walk Index. Lastly, Walk Index Sparsification is proposed to perform edge deletion based on Walk Index, and it markedly outperforms other baselines in terms of preserving prediction accuracy.

**Strengths:**

- An overall interesting theoretical work that supports the intuition: the interaction between two parts in a graph depends on how much interconnected their boundary is.
- The orginization is good, the theoretical analysis looks sound and the experiments are solid. The application to edge sparsification is impressive.

**Weaknesses:**

- Walk Index Sparsification is unable to scale up to large graphs with deep GNNs, given its $O(|E||V|^3\log(L))$ complexity. Correspondingly, the experiments only covered 2-WIS and 1-WIS. To further improve, the authors could discuss some potentially efficient implementations.

**Questions:**

See Weaknesses.

**Limitations:**

As stated in Weaknesses, the computational overhead is a concern.

---

> ### Author Rebuttal · Authors · 2023-08-05
>
> Thank you for your time and feedback. We are glad that you found our theory interesting, supported by solid experiments, and with an impressive application to edge sparsification. We address your concern below, and would greatly appreciate it if you would consider increasing your score.
>
> > ​​Walk Index Sparsification is unable to scale up to large graphs with deep GNNs, given its $O(|E||V|^3 \log (L))$ complexity. Correspondingly, the experiments only covered 2-WIS and 1-WIS. To further improve, the authors could discuss some potentially efficient implementations.
>
> As discussed in lines 284 to 293 of the paper, a naive implementation of (L - 1)-WIS (i.e. WIS for depth L GNNs) indeed requires runtime cubic in the number of vertices. While exploration of more efficient exact implementations (including ones involving parallelization) is left for future work, we propose and evaluate an extremely efficient approximation, namely 1-WIS (i.e. WIS for depth 2 GNNs), which requires only linear runtime and memory. For example, we demonstrated the applicability of 1-WIS to depth 10 GNNs (see Appendix H) over the OGBN-ArXiv dataset, which has more than a hundred thousand vertices and a million edges. Running 1-WIS over OGBN-ArXiv takes only ~20 minutes on a single V100 GPU.

---

### Official Review · Reviewer_2FCe · 2023-07-05

**Soundness:** 3 good
**Presentation:** 3 good
**Contribution:** 3 good
**Rating:** 7
**Confidence:** 3

**Summary:**

The paper proposes a new way to analyze the capacity of GNNs based on the complexity of interactions they can model across a partitioning of the nodes. In particular, this complexity is quantified via *separation rank*, and, for a certain kind of GNN with product aggregation, this rank is proved to scale with *walk index*, the number of random walks originating from a vertex on the partition's boundary. Experiments indicate that this finding generalizes beyond the particular GNN in the proofs. First, three different GNNs are shown to perform better at an image classification task when the data are converted to a graph so as to maximize walk index across the relevant partition. Second, the paper proposes WIS, a way to reduce the number of edges in the graph while maintaining GNN performance, based on maintaining high walk indices. WIS is shown to outperform prior methods on several real-world networks.

**Strengths:**

- Conceptual and theoretical explanations on the capacity of GNNs are an important topic, and this paper takes a fresh approach as compared to much of the WL-isomorphism-based work.
- The paper is well-organized, and the writing is grammatical and easy-to-read.
- Experiments show the superior efficacy of the proposed WIS method for graph sparsification relative to prior methods.

**Weaknesses:**

- While the machinery of the proof appears to be quite complex, I would appreciate some description of the proof concept in the main paper, even if it is at a very high level.
- Related to the prior point on the proof concept, I am not totally convinced that walk index is the fundamental quantity here. Particularly the results in Figure 3 for Cora, where 2-WIS's performance is essentially matched by 1-WIS (which simply relates to node degrees), suggest that there may be a simpler explanation. To convince the reader, perhaps there could be more conceptual description of walk index and separation rank, and why the two are fundamentally linked. Or perhaps there could be more experiments showing that walk index is more predictive of performance as compared to other notions of connectivity.

**Questions:**

- Do you see this work as totally orthogonal to WL-isomorphism-based limitation work? As I see it, those results are all-or-nothing, in the sense that they show GNNs either can or cannot distinguish two kinds of graphs. The bound here instead is somehow softer and more nuanced, showing that the complexity at which the GNN can model some interaction (as quantified by separation rank) scales with walk index. Is there some sense in which, or some specific cases for which, the WL results are subsumed by your new results?

- Related to the second 'weakness' above, do you see walk index as the fundamental notion here? If so, why?

**Limitations:**

Some limitations are noted throughout the work -- for example, it is noted that the theoretical results only formally apply to GNNs with product aggregation. A more collected discussion of limitations of the current analysis/understanding, and possible future directions, would be helpful.

---

> ### Author Rebuttal · Authors · 2023-08-05
>
> Thank you for the positive feedback and support!
>
> > While the machinery of the proof appears to be quite complex, I would appreciate some description of the proof concept in the main paper, even if it is at a very high level.
>
> Due to lack of space we deferred a high-level description of the proof concepts to Appendix A. For the camera-ready version, we will use the additional space to incorporate this content in the main paper. Thank you for the suggestion!
>
> > Related to the prior point on the proof concept, I am not totally convinced that walk index is the fundamental quantity here. Particularly the results in Figure 3 for Cora, where 2-WIS's performance is essentially matched by 1-WIS (which simply relates to node degrees), suggest that there may be a simpler explanation. To convince the reader, perhaps there could be more conceptual description of walk index and separation rank, and why the two are fundamentally linked. Or perhaps there could be more experiments showing that walk index is more predictive of performance as compared to other notions of connectivity.
>
> Our theory implies that the walk index is a fundamental quantity in the sense that it governs the strength of interactions that the analyzed GNNs can model across a partition of vertices. Intuitively, it characterizes the amount of computation a GNN performs to aggregate information across the partition. With regards to experimentation, it is challenging to judge this way whether the walk index is more fundamental than alternative quantities, the reason being that factors beyond expressivity (e.g. ones relating to optimization and generalization) may confound results. We did however show that WIS, which is explicitly based on the walk index, sparsifies edges better than known alternatives. 1-WIS is also based on the walk index (despite being expressible in simpler terms) so we do not view its strong performance as an indication that the walk index may not be the central quantity. We believe further exploration of WIS, and in particular regimes where higher-order WIS performs better than 1-WIS, is a promising direction for future work.
>
> > Do you see this work as totally orthogonal to WL-isomorphism-based limitation work? As I see it, those results are all-or-nothing, in the sense that they show GNNs either can or cannot distinguish two kinds of graphs. The bound here instead is somehow softer and more nuanced, showing that the complexity at which the GNN can model some interaction (as quantified by separation rank) scales with walk index. Is there some sense in which, or some specific cases for which, the WL results are subsumed by your new results?
>
> This work can be viewed as orthogonal to WL-based analyses of expressive power. Namely, WL-based analyses characterize which graphs a GNN can distinguish between. In contrast, for a given graph topology and fixed-size GNN, our work characterizes the type of functions the GNN can represent with respect to the vertex features, as measured by separation rank. We believe both approaches bear significance, yet while studies of the ability of GNNs to distinguish non-isomorphic graphs are abundant, much less is known about the type of functions GNNs can efficiently represent with respect to the vertex features. We hope our work can lead to further progress along this line, whether through the notion of separation rank or through other notions of complexity that take into account the vertex features.
>
> > Some limitations are noted throughout the work -- for example, it is noted that the theoretical results only formally apply to GNNs with product aggregation. A more collected discussion of limitations of the current analysis/understanding, and possible future directions, would be helpful.
>
> As you noted, we mentioned possible limitations of our work throughout. In the revised manuscript we will also include a concentrated account of them and future directions in the summary section. Thank you for the suggestion!

---

> > ### Comment · Reviewer_2FCe · 2023-08-14
> >
> > Thank you for your response. I will keep my score to support the paper, though I hope that a few limitations will be noted clearly:
> > - The theoretical results only apply to product aggregation, which is not commonly used to my knowledge, and, as Reviewer CGQv also noted, the aggregation function is important.
> > - Empirically, k-WIS has not been shown to be more useful than 1-WIS (degree information). I think some demo to this effect would help corroborate the theory and overcome the point above about the aggregation function, even if the k-WIS variant is not scalable.

---

> > > ### Author Response · Authors · 2023-08-15
> > >
> > > We are adding to the text a concentrated account of limitations and directions for future work. The points you raised (which currently are mentioned throughout) will be noted clearly in this account. Thank you for the support!

---

### Official Review · Reviewer_CGQv · 2023-07-22

**Soundness:** 3 good
**Presentation:** 3 good
**Contribution:** 2 fair
**Rating:** 5
**Confidence:** 4

**Summary:**

This paper proposes to analyze the expressivity of graph neural networks (GNNs) by their ability to model interactions. The authors consider GNNs with product aggregation scheme, and analyze their ability to model interactions for a given partition of the graph through the notion of separation rank. The authors show quantitatively that these GNNs can model interactions better for partitions with higher walk index (Theorem 1). Motivated from their theoretical analysis, the authors propose an edge sparsification algorithm --- Walk Index Sparsification (WIS, Algorithm 1, 2) --- to drop edges that have least amount of impact on GNN's ability to model interations for a given partition in a given graph. The utility of WIS is demonstrated on both graph-level prediction and vertex-level prediction tasks.

**Strengths:**

Strengths:

1. The proposal to study the expressivity of GNNs via their ability to model vertex interactions is interesting.

2. The theoretical analysis is novel: by focusing on GNNs with product aggregation, the authors connect them to tensor networks, of which the notion of separation rank can be used to study the network's ability to model interactions.

3. The paper is overall well-written and easy to follow, where important technical details and illustrative examples are explained clearly in the supplementary materials.


**Weaknesses:**

Weaknesses:

1. The theoretical analysis focus on a special kind of GNNs with product aggregation, whereas the empirical evaluations are all done using standard GNNs with mean aggregation with ReLU nonlinearity. Although the authors meant to show that theory results can carry over to standard GNNs, they do not explain explicitly how such connections can be drawn. If the main contribution of the paper is...
(a) to show that using separation rank to understand GNNs is relevant, then the authors should outline how the analysis carries forward to standard GNNs
(b) to show that walk index of partitions in a graph determines GNN performance, then the authors should examine the case where the original graph is replaced with a fully-connected graph, and discuss GNNs with Graph Transformers
(c) to propose an edge sparsification algorithm, then comparisons with stronger baselines are needed, such as DropEdge [1] and curvature-based graph rewriting methods [2].


2. The conclusion that GNNs can model interactions better on partitions that have high walk index (i.e. more interconnected) seems to crucially depend on the choice of aggregation function is tied to the graph topology. However, it seems possible to use different aggregation/pooling function that delineates from the graph topology to capture interactions among partitions with low walk index. More explanations on how to interpret the results are needed.


References:

[1] Rong, Yu, et al. "Dropedge: Towards deep graph convolutional networks on node classification." arXiv preprint arXiv:1907.10903 (2019).

[2] Topping, Jake, et al. "Understanding over-squashing and bottlenecks on graphs via curvature." arXiv preprint arXiv:2111.14522 (2021).


**Questions:**


1. The analysis focus on GNNs with product aggregation, i.e. multiplication of the neighborhood node's embedding vectors $\mathbf{h}^{(l, i)}=\odot_{j \in \mathcal{N}(i)}\left(\mathbf{W}^{(l)} \mathbf{h}^{(l-1, j)}\right)$, where $\mathcal{N}(i)$ depends on the graph topology. In this set-up, the authors show that such GNNs can model interactions among two partitions in a graph better if they are more inter-connected. Yet, earlier work by Cohen and Shashuwa [3] show that for the case of convolutional neural networks (CNNs), their ability to model interactions depend on the choice of pooling functions. In light of this result, it seems plausible for GNNs to model partitions that are less-connected (e.g., Fig 2 -left) if one delineates the aggregation function from the graph topology, which is also commonly used in the literature of graph rewriting (e.g. [2]). Can the authors provide more explicit interpretations of their results and comparison with related work using separation rank for CNNs/RNNs?

2. As a thought example: for GNNs with product aggregation, do they achieve maximally separation rank for a fully-connected graph? How would the WIS algorithm compare to a simple baseline: replacing the original graph with a fully-connected graph, followed by randomly dropping edges (as opposed to (i) edges are removed randomly from the original graph)? This baseline is motivated from [4], where the authors show that replacing the original graph with a fully-connected graph at the output layer can boost GNN performance.

3. In line 264-266, the authors explain that the edge sparsification algorithm should first choose the partitions of interest, whereas in Algorithm 1 and 2, such choice is default to partitions induced by singletons and the rest of the nodes. It seems possible that the optimal partition choice depends on the downstream tasks (e.g., graph-level clustering versus node-level prediction). Can the authors provide more details and recommendations?

References:

[3] Cohen, Nadav, and Amnon Shashua. "Inductive bias of deep convolutional networks through pooling geometry." arXiv preprint arXiv:1605.06743 (2016).

[4] Alon, Uri, and Eran Yahav. "On the bottleneck of graph neural networks and its practical implications." arXiv preprint arXiv:2006.05205 (2020).

**Limitations:**

The limitations were discussed, but can be explained more thoroughly through (1) detailed interpretations of the theoretical results (assumptions and applicability); (2) comparison with prior works that make use of separation rank to analyze expressivity of deep learning architectures.

---

> ### Author Rebuttal · Authors · 2023-08-05
>
> Thank you for your feedback, for highlighting the interest and novelty of our theory, and for noting that the paper is well-written. We treat your comments and questions below. If our response is satisfactory, we would greatly appreciate it if you would consider raising your score.
> > (a) to show that using separation rank to understand GNNs is relevant, then the authors should outline how the analysis carries forward to standard GNNs
>
> Analyzing neural network architectures identical to those popular in practice is often notoriously difficult. Accordingly, it is customary for works in deep learning theory to analyze different variations of these architectures. Following numerous past works (see lines 144 to 152 of the paper), we analyze neural networks with product aggregation, and demonstrate the applicability of our conclusions to architectures with other aggregation types via experimentation, regarding formal proof of the applicability as an avenue for future work. As stated in lines 163 to 166, some of the past analyses of neural networks with product aggregation were later extended to account for additional aggregations. We believe our theory may be similarly extended, and intend to pursue this route in future work.
>
> > (b) to show that walk index of partitions in a graph determines GNN performance, then the authors should examine the case where the original graph is replaced with a fully-connected graph, and discuss GNNs with Graph Transformers
>
> Increasing connectivity improves expressivity in terms of separation rank, and there is evidence for it improving performance in practice. However, the performance in practice depends not only on expressivity but also on optimization and generalization, and from that perspective excess connectivity may lead to problems, for example over-smoothing and over-squashing. This is discussed in lines 203 to 213. Regarding comparison to Graph Transformers, our work focuses on message-passing GNNs. Analyzing the strength of interactions that Graph Transformers can model and comparing it against our results for message-passing GNNs is a direction we view as appropriate (and interesting!) for future research.
>
> > (c) to propose an edge sparsification algorithm, then comparisons with stronger baselines are needed, such as DropEdge [1] and curvature-based graph rewriting methods [2].
>
> Edge sparsification algorithms, such as the one we propose, attempt to maintain the prediction accuracy as the number of removed edges increases, with the goal of reducing computational and/or memory costs. The sparsification is done once, as a preprocessing step, and is fixed during training. Thus, in their original form, DropEdge and graph rewiring algorithms such as that from [2], whose goal is improving prediction accuracy (rather than efficiency), are incomparable to edge sparsification methods. Specifically, DropEdge requires storing the whole graph throughout training, while graph rewiring algorithms typically add or remove only a few edges, with the resulting graph having roughly the same number of edges as the original one (or even more edges). One may try to adapt curvature-based rewiring algorithms to edge sparsification. We believe doing so and comparing them to WIS is an interesting topic for future work.
>
> > ...it seems plausible for GNNs to model partitions that are less-connected (e.g., Fig 2 -left) if one delineates the aggregation function from the graph topology, which is also commonly used in the literature of graph rewriting (e.g. [2]).
>
> The aggregation in the standard message-passing GNN formulation is done according to the graph structure. Hence we focus on this case in the paper. Indeed, it is possible to detach the pooling geometry from the input graph structure, in which case our results apply as is with the walk index over the input graph replaced by that of the graph defining the pooling geometry. We hope the above has addressed your inquiry; if not please let us know.
>
> > Can the authors provide more explicit interpretations of their results and comparison with related work using separation rank for CNNs/RNNs?
>
> CNNs and RNNs can be cast as GNNs with certain input graph structures – grid graph for CNNs and chain graph for RNNs. Accordingly, our separation rank bounds extend those established for CNNs and RNNs to GNNs with arbitrary input graph structures.  We will mention this point in the text. Thank you for bringing it up!
>
> > How would the WIS algorithm compare to a simple baseline: replacing the original graph with a fully-connected graph, followed by randomly dropping edges (as opposed to (i) edges are removed randomly from the original graph)?
>
> Since the baseline ignores all information about the original graph structure, it is expected to yield poor prediction accuracy. To affirm this expectation, we ran experiments over the Cora dataset using GCN. Indeed, **it led to substantially worse test accuracy compared to the rest of the baselines and WIS**. Specifically, over the graphs obtained by the proposed baseline, the test accuracy was roughly 73% on average for each of the edge sparsity levels – 0%, 5%, 10%, …, 100%. This is around the test accuracy one gets when removing all edges, and in particular significantly lower than that achieved by WIS (or the other baselines) — cf. Figure 3 (left) in the paper.
>
> > “...It seems possible that the optimal partition choice depends on the downstream tasks (e.g., graph-level clustering versus node-level prediction). Can the authors provide more details and recommendations?”
>
> Excellent point. Indeed, the optimal choice of partitions can vary depending on the task at hand. As stated in lines 272 to 275, we focus on a specific instantiation of the general WIS scheme which is tailored for vertex prediction tasks (these are particularly relevant with large-scale graphs). Exploring other instantiations, and methods for automatically choosing partitions, are regarded as promising avenues for future work.

---

> > ### Comment · Reviewer_CGQv · 2023-08-12
> > **Explicit comparison of stable rank bounds obtained from this paper versus prior works**
> >
> > Thank you very much for your detailed response. I have the follow-up question on your point of:
> >
> > *CNNs and RNNs can be cast as GNNs with certain input graph structures – grid graph for CNNs and chain graph for RNNs. Accordingly, our separation rank bounds extend those established for CNNs and RNNs to GNNs with arbitrary input graph structures.*
> >
> > Would it be possible to provide an explicit comparison, showing the separation rank bound obtained from your work for GNN, its applications for CNNs and RNNs (viewed as special instances of graphs), along side with previous bounds obtained solely for CNNs and RNNs? I believe this will increase the technical strength of the paper.

---

> > > ### Author Response · Authors · 2023-08-15
> > >
> > > Certainly! We provide such a comparison below. We will elaborate upon this point in the revised manuscript; thank you for the question!
> > >
> > > For a two-dimensional grid graph with $N$ vertices, a message-passing GNN can be viewed as a CNN. Given $I \\subseteq [N]$, recall that $C_I$ denotes the boundary of the partition induced by $I$, i.e. the set of vertices with an edge crossing the partition. The $L - 1$ walk index of $I$ in the grid graph satisfies $| C_I | \\cdot 3^{L - 1} \\leq WI_{L - 1} (I) \\leq |C_I| \\cdot 5^{L - 1}$ (this is because  each vertex has degree $5$ including self-loops, except those on the edge or corner of the grid which are of degree $3$ or $4$ respectively). Hence, Theorem 1 implies that the separation rank of the analyzed GNN with respect to $I$ is at most $D_h^{O (|C_I| \cdot 5^{L - 1})}$, where $D_h$ is the network’s width.
> > >
> > > Similarly, for a chain graph with $N$ vertices, a message-passing GNN can be viewed as a bidirectional RNN. Given a partition induced by $I \\subseteq [N]$, the $L - 1$ walk index of $I$ in the chain graph satisfies $| C_I | \\cdot 2^{L - 1} \\leq WI_{L - 1} (I) \\leq |C_I| \\cdot 3^{L - 1}$ (this is because each vertex has degree $3$ including self-loops, except those on the endpoints which are of degree $2$). Thus, Theorem 1 implies that the separation rank of the analyzed GNN with respect to $I$ is at most $D_h^{O( |C_I| \cdot 3^{L - 1} )}$.
> > >
> > > The above results — special cases of our theory — extend those of [1,2,3,4], which study the separation rank of certain CNNs and RNNs. In particular, the results extend the CNN framework of [1,2] by introducing overlaps to convolution windows,$^\dagger$ and the RNN framework of [3,4] by introducing bidirectionality. Given that the models analyzed in [1,2,3,4] are weaker, the corresponding bounds on separation rank developed in these papers are lower.
> > >
> > > Consider for example the “odd-even” and “low-high” partitions from Figure 1(c) of [1]. Denote the subsets yielding these partitions by $I_{odd}$ and $I_{low}$ respectively. Applying our CNN result to $I_{odd}$ and $I_{low}$ yields upper bounds of $D_h^{O (N \cdot 5^{L - 1})}$ and $D_h^{O( \sqrt{N} \cdot 5^{L - 1} )}$ respectively. In contrast, the bounds in [1,2] (which were derived for a weaker model) are $D_h^{O(N)}$ and $D_h$ respectively.
> > >
> > > Similarly, for a chain graph, consider the “odd-even” partition induced by $I_{odd} := \\{ 1, 3, …, N \\}$ and the “low-high” partition induced by $I_{low} := \\{1, \ldots, N / 2 \\}$. Applying our RNN result to $I_{odd}$ and $I_{low}$ yields upper bounds of $D_h^{O (N \\cdot 3^{L - 1})}$ and $D_h^{O( 3^{L - 1} )}$ respectively. In contrast, the bounds in [3,4] (which were derived for a weaker model) are $D_h^{O(N)}$ and $D_h$ respectively.
> > >
> > > We hope the above fully addresses your question. If not please let us know and we will happily expand!
> > >
> > > ---
> > >
> > > $\dagger$ Note that CNNs with overlapping windows were studied in [5], but this paper did not provide upper bounds on the separation rank.
> > >
> > > [1] Cohen, Nadav, and Amnon Shashua. "Inductive bias of deep convolutional networks through pooling geometry." ICLR (2017)
> > >
> > > [2] Levine, Yoav, et al. "Deep learning and quantum entanglement: Fundamental connections with implications to network design." ICLR (2018)
> > >
> > > [3] Khrulkov, Valentin, Alexander Novikov, and Ivan Oseledets. "Expressive power of recurrent neural networks." ICLR (2018)
> > >
> > > [4] Levine, Yoav, Or Sharir, and Amnon Shashua. "Benefits of depth for long-term memory of recurrent networks." (2018)
> > >
> > > [5] Sharir, Or, and Amnon Shashua. "On the expressive power of overlapping architectures of deep learning." ICLR (2018).

---

> > > > ### Comment · Reviewer_CGQv · 2023-08-15
> > > > **Clarification on the proof of "stronger" bound**
> > > >
> > > > Thank you for your further explanation. I follow the first half but want to clarify the comparison at the end:
> > > > Since $D_h$ represents the network width, then $D_h > 1$.
> > > > Given $D_h > 1$, and $L > 1$,  shouldn't it be
> > > > $$
> > > > D_h^{O(N)} < D_h^{O(N  5^{L-1})}; D_h^{O(N)} < D_h^{O(N  3^{L-1})}
> > > > $$
> > > > Namely, the upper bounds in [1,2], [3,4] (LHS) seem to be sharper than yours (RHS) for the odd-even partition? What do I miss here?

---

> > > > > ### Author Response · Authors · 2023-08-15
> > > > >
> > > > > The bounds from [1,2] and [3,4] are not sharper but rather apply to weaker architectures that can attain only a smaller separation rank. In particular, the CNNs considered in [1,2] have non-overlapping convolution windows while the analyzed GNN over a grid graph is equivalent to a CNN with overlapping convolution windows, as is typically the case in practice. Furthermore, the RNNs considered in [3,4] are unidirectional while the analyzed GNN over a chain graph is equivalent to a bidirectional RNN.
> > > > >
> > > > > We hope this has clarified why the bounds from [1,2,3,4] are lower (but not sharper). Please let us know if you have any further questions, we will happily elaborate.

---

> > > > > > ### Author Response · Authors · 2023-08-17
> > > > > >
> > > > > > Dear Reviewer,
> > > > > >
> > > > > > We hope that our latest response has fully addressed your question. Please let us know if you have any additional feedback; there is still some time left in the discussion period and we will gladly address any question or comment you may have.
> > > > > >
> > > > > > Authors

---

> > > > > > > ### Comment · Reviewer_CGQv · 2023-08-18
> > > > > > > **Thank you - this clarifies!**
> > > > > > >
> > > > > > > Thank you again for the clarification. This makes sense to me and I was confused with the wording "lower" versus "sharper". Maybe consider rephrasing as "their weaker model....tighter bound" to ease the burden of the readers.
> > > > > > >
> > > > > > > I encourage the authors to add these bound comparison and discussion to the paper, which will be a good contribution.

---

> > > > > > > > ### Author Response · Authors · 2023-08-18
> > > > > > > >
> > > > > > > > We will indeed include the comparison and discussion above in the paper; thank you for the suggestion and feedback! If by this we have addressed all of your questions, we would greatly appreciate it if you would consider increasing your score. Thank you again for the time and for the thoughtful engagement during the discussion period.

---

### Decision · Program_Chairs · 2023-09-21

**Decision:**

Accept (poster)

**Comment:**

This paper proposes to characterize how GNNs can model the interaction strength between a partitioning of nodes via separation rank, and proves the separation rank is positively related to a new defined walk index (the number of random walks starting from the partition boundary nodes). Reviewers agree that the theory is new and interesting, orthogonal to the traditional WL-based GNN expressivity analysis. One downside of the paper lies in its strong assumption on the element-wise product based neighbor aggregation function, which is rarely used in practical GNNs. The authors are suggested to extend their theory to standard sum/mean based aggregation functions (if possible to any extent) in addition to the empirical verification. Overall, a recommendation of acceptance is given.